

# Optimizing Four Years of CO$_2$ Biospheric Fluxes from OCO-2 and in situ data in TM5: Fire Emissions from GFED and Inferred from MOPITT CO data

Hélène Peiro [1], Sean Crowell [1], and Berrien Moore III [1]

[1]University of Oklahoma, Norman, OK, USA

**Correspondence:** Helene Peiro (helene.peiro@ou.edu)

**Abstract.** Column mixing ratio of carbon dioxide (CO$_2$) data alone do not provide enough information for source attribution. Carbon monoxide (CO) is a product of inefficient combustion often used as a tracer of CO$_2$. CO data can then provide a powerful constraint on fire emissions, supporting more accurate estimation of biospheric CO$_2$ fluxes. In this framework and using the chemistry transport model TM5, a CO inversion using MOPITT v8 data is performed to estimate fire emissions

which are then converted in CO$_2$ fire emissions through the use of emission ratio. These CO$_2$ fire emissions allow us, then, to estimate adjusted CO$_2$ Net Ecosystem Exchange (NEE) and respiration which are then used as priors for CO$_2$ inversions constrained either by the Orbiting Carbon Observatory 2 (OCO-2) v9 or by in situ data. For comparison purpose, we also balanced the respiration using fire emissions from the Global Fire Database Emissions (GFED) version 3 (GFED3) and version 4.1s (GFED4.1s). We hence study the impact of CO fire emissions in our CO$_2$ inversions at global, latitudinal and regional

scales over the period 2015 - 2018 and compare our results to the two other similar approaches using GFED3 and GFED4.1s, as well as with an inversion using CASA-GFED3 fire and NEE priors. After comparison at the different scales, the inversions are evaluated against TCCON data. Results show that variations in posterior flux are much smaller across different prior mean fluxes when compared with the data assimilated. However, at global scale and for most of the regions, while the net fluxes remain robust, we can observe differences in fire emissions among the priors, resulting in large adjustments in the Net

Ecosystem Exchange (NEE) to match the fires and observations. Tropical flux estimates from in situ inversions are highly sensitive to the prior flux assumed, of which fires are a significant component. Slightly larger CO$_2$ net sources are observed when using GFED4.1s and MOPITT CO prior in CO$_2$ OCO-2 inversions than compared with the other priors, particularly during the 2015 El Niño event for most Tropical regions. Larger CO$_2$ net sources with MOPITT CO and GFED4.1s priors are also observed in Tropical Asia in CO$_2$ in situ inversions than compared with the other priors during the 2015-2016 El

Niño period and shows large net emissions than compared to OCO-2 inversions. Evaluation with TCCON suggests that the re-balanced posterior simulated give biases and accuracy very close each other where biases have decreased and variability matches better the validation data than with the CASA-GFED3. Further work is needed to improve prior fluxes in Tropical regions where fires are a significant component.



# 1 Introduction

Carbon dioxide ($CO_2$) is the most important greenhouse gas contributing to global climate change (IPCC, 2014). Gaps in our understanding of the processes that control land-sea-atmosphere exchange of $CO_2$ are a leading order uncertainty in future projections of the global climate (Friedlingstein et al., 2014). The global net flux, and hence the airborne fraction, can be deduced from the atmospheric growth rate (Ballantyne et al., 2012), and historically different efforts, such as the Global Carbon Project (Le Quéré et al., 2009) have divided the total global net flux into its constituent components, consisting of

fluxes from the ocean, terrestrial biosphere, fossil fuel combustion and other anthropogenic activities, and biomass burning.

$CO_2$ emissions from fires are well-characterized at the largest space and time scales, but the uncertainties increase rapidly as we look to finer space and time scales. Two approaches are currently employed to estimate global emissions from fires. The first uses total fuel consumption per product of the burned area and the fuel consumption per unit area deduced from the burned area and active fires products of the Moderate Resolution Imaging Spectroradiometer (MODIS). The Global Fire Emissions

Database (GFED) products (van der Werf et al., 2010) and the Fire INventory from NCAR (FINN) (Wiedinmyer et al., 2011), for instance, use this approach. GFED was developed for understanding monthly contribution of fires to global carbon cycling (van der Werf et al., 2004), while FINN was developed for near real-time estimation (Wiedinmyer et al., 2011). The second technique deduces fuel consumption from Fire Radiative Power (FRP) determined from infrared thermal measurements. Two emission inventories use this approach, the Global Fire Assimilation System (GFAS) (Kaiser et al., 2012) and the Quick Fire

Emissions Database (QFED) (Darmenov and Silva, 2015). Several studies used and compared these fire emissions inventories and found several differences in capturing wildfire activity over different areas as well as sources of uncertainties from the cloud gap adjustments, small fires estimations and land use and land cover estimation (Liu et al., 2020). While these fire emission inventories all use the MODIS thermal anomalies (Giglio et al., 2006), they use different methods of translating emission factors and land cover to estimate fire emissions. Although the quantification of emissions from biomass burning

from space-based instruments has increased significantly, uncertainties regarding input data and methodologies can still lead to errors up to an order of magnitude for the total trace gases emissions (Vermote et al., 2009; Baldassarre et al., 2015).

Moving from global annual fluxes to finer scales in space and time complicates the inference a great deal. Interpreting atmospheric measurements at these scales requires the use of an atmospheric chemistry transport model (CTM) and optimal estimation machinery, frequently referred to in the literature as "atmospheric inversions", or "top-down inversions". However,

even using the same set of observations such as the Orbiting Carbon Observatory 2 (OCO-2) data in different inverse modeling systems can induced a large range of $CO_2$ fluxes estimation at regional scales (Crowell et al., 2019; Peiro et al., 2022). Flux estimates from top-down inversions have been shown to be sensitive to the choice of transport model (Schuh et al., 2019), and observational coverage (Byrne et al., 2017). Even more importantly, atmospheric measurements of $CO_2$ dry air mole fractions represent the combined influence of all upstream emissions and transport, and so individual tracer measurements cannot be used

to differentiate between different source or sink processes without more information. Additionally, prior estimate of the fluxes and their associated uncertainties can impact posterior $CO_2$ estimations (Lauvaux et al., 2012b, a; Byrne et al., 2017; Gurney et al., 2003; Wang et al., 2018; Chevallier et al., 2005; Baker et al., 2006, 2010). A few studies (Liu et al., 2017a; Palmer et al.,





2019; Crowell et al., 2019; Peiro et al., 2022) utilized $XCO_2$ from OCO-2 to constrain top-down surface fluxes of $CO_2$. All of the mentioned studies found the Tropics to be a large source region for 2015-2016, though the explanations varied. Crowell
et al. (2019) showed that an ensemble of inversion models delivered robust results for Tropical regions when OCO-2 data was assimilated. The ensemble employed included different atmospheric transport models, prior ocean and terrestrial biosphere and fire fluxes, and assimilation techniques. All of the participating models did not optimize fire and fossil fuel emissions. As such, only the non-fossil land (net biosphere exchange; NBE) and ocean flux at regional scales was examined in the study, with no attempt to attribute ensemble spread to different sources of uncertainty, such as the assumed fire emissions, which neglected to
include some of the global inventories, such as FINN, QFED, and GFED4.1s (earlier versions of GFED were included).

Most inversion models do not explicitly constrain fire emissions with $CO_2$ observations. Rather, it is assumed that fire emissions have much lower uncertainty (generally believed to be less than 10% (Quéré et al., 2018; Quilcaille et al., 2018)) than the ocean and terrestrial biosphere fluxes (Quéré et al., 2018; Khatiwala et al., 2009, 2013), and so are held fixed, with the net ecosystem exchange (NEE) being assumed to be the residual between the posterior total net land flux and the assumed
fire and fossil fuel emissions. This inference is problematic, not least due to the aforementioned fire emissions uncertainties in time and space, which could alias into inferred biospheric fluxes at continental or regional scales (Wiedinmyer and Neff, 2007; Peylin et al., 2013). To reduce the uncertainties associated with fires and consequently with $CO_2$ biospheric emissions, we can examine gas species that are co-emitted with $CO_2$ from fires, such as carbon monoxide (CO).

CO is an air pollutant that affects the oxidation capacity of the atmosphere through its reaction with the hydroxyl radical
(OH), leading to a relatively short atmospheric lifetime of one to three months because of its fast oxidation with OH. Reactions between CO and OH impact atmospheric composition on hemispheric (mainly in the Tropics) or even global scales (Logan et al., 1981). CO also leads to the formation of tropospheric ozone ($O_3$), an important short-lived greenhouse gas, and $CO_2$. CO is produced by incomplete combustion, i.e. when there is not enough oxygen to make $CO_2$ (van der Werf et al., 2010), such as in the case of smoldering fires. In this way, $CO_2$ is strongly co-emitted with CO in the presence of combustion
(Bakwin et al., 1997; Potosnak et al., 1999; Turnbull et al., 2006). Previous studies used trace gases such as CO to improve the $CO_2$ flux estimation or to separate $CO_2$ emissions sources. Wang et al. (2010) used the $CO_2$/CO correlation slope to differentiate the source signature of $CO_2$ and separate the different characteristics of $CO_2$ emissions between rural and urban sites in China. Basu et al. (2014) estimated $CO_2$ emissions with Greenhouse gases Observing SATellite (GOSAT) data and the Comprehensive Observation Network for TRace gases by AIrLiner (CONTRAIL) project and studied seasonal variations of
$CO_2$ fluxes during the 2009 and 2011 period over Tropical Asia. By using the Infrared Atmospheric Sounding Interferometer (IASI) CO measurements, their study showed an increased source of $CO_2$ in 2010 that was caused not by rising of biomass burning emissions but by biosphere response to above-average temperatures. In addition to CO, some studies worked on the correlation between additional species and $CO_2$ to constrain $CO_2$ emission from biomass burning. Konovalov et al. (2014) used satellite CO and aerosol optical depth data to constrain $CO_2$ emissions from wildfires in Siberia by estimating FRP to
biomass burning rate conversion factors. Using this approach, they found that global emissions inventories underestimated $CO_2$ emissions from Siberia from 2007 to 2011.





As biomass burning emissions estimates are necessary for constraining top-down $CO_2$ emissions, we want to provide our $CO_2$ inversion model with fire emissions that contain as much realism as possible. Fires that incorporate information from both traditional bottom-up estimation techniques and atmospheric CO data may provide a better estimate than the global inventories alone. The corresponding top-down $CO_2$ fluxes imposing these optimized fire emissions should have more fidelity, particularly in the Tropics, where fires and the biosphere strongly interact with one another, and especially during severe drought conditions associated with the 2015 El Niño. The objective of this paper is to assess the improvement in $CO_2$ biogenic emissions estimates when CO-informed fire emissions are imposed, particularly during the 2015 El Niño event and the post-event (2017 and 2018). First, we constrain CO emissions using data from the Measurements of Pollution in The Troposphere (MOPITT). We use these optimized CO emissions together with key vegetation parameters from GFED to create an updated estimate of fire $CO_2$ emissions that incorporates both sets of information. Finally, these updated fire emissions are imposed in an atmospheric $CO_2$ inversion that constrains $CO_2$ fluxes, using either OCO-2 $XCO_2$ retrievals or in situ data, with different assumed fire emissions and appropriately rebalanced prior biogenic fluxes.

This paper is ordered as follows. The assimilation and evaluation data sets and the inversion modeling framework are described in Section 2. The results for CO and $CO_2$ flux estimates and evaluation against independent data are presented in Section 3. The importance of these results for conclusions about the terrestrial biosphere using top-down inversion models is discussed in Section 4. Conclusions and proposed future work are presented in Section 5.

## 2 Data and methodology

Our experiments focus on estimation of top-down fluxes using the TM5-4DVAR system (e.g. Meirink et al. (2008); Basu et al. (2013); Crowell et al. (2018)). Our inversions are performed in sequence, first assimilating total column CO retrievals from the MOPITT v8 products to produce optimized CO fluxes which are used to update the assumed $CO_2$ fire emissions, and then we optimized $CO_2$ fluxes using total column $CO_2$ from OCO-2 version 9 retrievals or in situ data. We introduce hereafter the observations used in the inversions, the inversion system and the observations used for validations.

### 2.1 Data sets

#### 2.1.1 MOPITT data

Space-based CO data are available from a large variety of instruments: IASI (Infrared Atmospheric Sounding Interferometer, Turquety et al. (2004); Clerbaux et al. (2009)) on-board Metop satellite, MOPITT (Measurements of Pollution in the Troposphere, Drummond et al. (2010, 2016)) on-board the Terra satellite, the Tropospheric Emission Spectrometer (TES, Beer (1999)) on-board EOS-Aura and the Atmospheric InfraRed Sounder (AIRS, Aumann et al. (2003)) on-board EOS-Aqua. These satellite data can be used to monitor fire emissions from an atmospheric point of view. So far, MOPITT has been the only space-based instrument deriving CO from near-infrared (NIR), thermal infrared (TIR) and multispectral radiances (TIR + NIR). Recently, TROPOspheric Monitoring Instrument (TROPOMI, Landgraf et al. (2016)) and GOSAT-2 TANSO-FTS-2





(http://www.gosat-2.nies.go.jp/) are also retrieving CO from NIR radiances. However, MOPITT products have been consistently validated against airborne vertical profiles and ground based measurements, allowing a well-understood of its continuity
and consistency (Worden et al., 2010; Deeter et al., 2019).

MOPITT (Drummond, 1993) was launched in 1999 on board the Terra satellite. Terra flies in a sun-synchronous polar orbit at an altitude of 705 km, crossing the equator at approximately 10:30 local time each morning and evening. It has a nadir view with spatial resolution of 22 x 22 km. Its swath is 650 km wide, with 116 cross-track footprints. MOPITT achieves a global coverage in about 4 days.

MOPITT uses gas filter correlation radiometry to retrieve CO mixing ratios from radiances in the 4.7 $\mu$m (TIR) and 2.3 $\mu$m (NIR) spectral bands. TIR-only retrievals of MOPITT have been shown to be mostly sensitive to CO in the mid-upper troposphere (excluding regions with strong thermal gradients such as deserts, Deeter et al. (2007)). NIR-only retrievals depend on reflected solar radiation, and are also used for retrievals of CO total column, though the vertical sensitivity is stronger near the surface than the TIR-only retrievals (Deeter et al., 2009; Worden et al., 2010). MOPITT TIR + NIR retrievals can
provide improved estimates of CO near source locations and has enhanced land surface sensitivity compared to the TIR only product (Deeter et al., 2015). In this study, we consequently use the level 2 TIR-NIR profiles product in order to have better sensitivity of CO on the total column with greatest sensitivity in the lower troposphere (Deeter et al., 2013). With the observing limitations of NIR data, this product is limited to daytime observations over land. In addition, because retrievals with surface pressures less than 900 hPa might be of lower quality, they are removed for the assimilation (Fortems-Cheiney et al., 2011; Yin
et al., 2015). MOPITT retrieval products are generated with an optimal estimation-based retrieval algorithm and a fast radiative transfer model involving both MOPITT calibrated radiances and a priori knowledge of CO variability (Deeter et al., 2003). The MOPITT operational fast forward model (MOPFAS) is a radiative transfer model based on HITRAN2012 (Rothman et al., 2013) database with CO parameters in log(VMR) used to simulate the MOPITT measured radiances (Edwards et al., 1999). For this retrieval method, cloud-free observations are required. The MOPITT v8 products consist of CO profile with 10 pressure
levels. In our assimilation system, simulated values of log XCO using the MOPITT v8 averaging kernel are compared to the retrievals, and the difference is then propagated into flux adjustments using the TM5 adjoint.

Several studies have used inverse modelling with MOPITT data to estimate CO emissions (Huijnen et al., 2016; Yin et al., 2016; Nechita-Banda et al., 2018) and they showed that MOPITT v7 data have poor performance at detecting extreme events. However, MOPITT v8 implemented a bias correction in the radiance which demonstrated improved retrievals relative to v7
(Deeter et al., 2019). In particular, MOPITT v8 does not exhibit a latitudinal dependence in partial CO column biases observed in v7 (Deeter et al., 2019). MOPITT v8 TIR-NIR product biases are within 5% at all levels when compared to NOAA aircraft profiles. In addition, apparent long-term trends in v7 biases have been decreased to 0.1%/yr or less at all retrievals levels for v8 products (Deeter et al., 2019). We thus expect to have better performance in the detection of extreme events by assimilating MOPITT v8 and less bias in the inferred CO emissions overall.


### 2.1.2 OCO-2 data

The OCO-2 (Crisp et al. (2017); Eldering et al. (2017)) satellite was launched in July 2014 as the first NASA mission dedicated to observing $CO_2$ from space. The satellite flies in a sun-synchronous orbit with an altitude of 705 km and a 16 day revisit time. OCO-2 passes each location at approximately 13:30 local time (Crisp and Johnson, 2005). OCO-2 observes 8 footprints across a 10 km ground track, each of which is less than 1.29 km by 2.25 km (Eldering et al., 2017). Smaller spatial footprints increase the number of cloud-free scenes allowing for more successful retrievals with lower errors (O'Dell et al., 2018), e.g. relative to the Greenhouse Gases Observing Satellite (GOSAT; Kuze et al. (2009)).

OCO-2 measures the absorption of solar reflectance spectra within $CO_2$ (1.6 $\mu$m and 2.0 $\mu$m) and molecular oxygen ($O_2$) bands (0.76 $\mu$m). Retrievals from OCO-2 have sensitivity throughout the entire troposphere with highest sensitivity close to the surface (Eldering et al., 2017). As with CO, retrievals of $CO_2$ from TIR observations such as those from TES or AIRS typically have lower sensitivity in the atmospheric boundary layer (Eldering et al., 2017).

$CO_2$ retrieval products come from the Atmospheric Carbon Observations from Space (ACOS) retrieval algorithm (O'Dell et al., 2012; Crisp et al., 2012; O'Dell et al., 2018; Kiel et al., 2019). OCO-2 radiance measurements are analyzed with remote sensing retrieval algorithms to spatially estimate column-averaged $CO_2$ dry air mole fraction, $XCO_2$. This quantity represents the average concentration of $CO_2$ in a column of dry air from the surface to the top of the atmosphere. ACOS $XCO_2$ product have been largely validated against ground-based observations from the Total Column Carbon Observing Network (TCCON; Wunch et al. (2017)). Our study uses the OCO-2 version 9 data product, as it contains all of the improvements as well as a bug fix that was found after the release of the version 8 (v8). Being a nonlinear optimal estimation product, retrievals contain residual errors that must be removed through the use of a bias correction (O'Dell et al., 2018; Kiel et al., 2019). Residual biases in $XCO_2$ were reduced especially over rough topography, which were found to be caused by relative pointing offsets between the three bands. Even after the bias correction is applied, errors on regional scales likely remain (O'Dell et al., 2018). Despite these shortcomings, data coverage from satellites is dense in the Tropics relative to the global in situ network, which has very few sites there. Despite biases with satellite data but thanks to their large spatial coverage, several studies prefer to use satellite data over the Tropics. For instance, Liu et al. (2017a) and Palmer et al. (2019) have discussed the impacts of the 2015-2016 El Niño event on the carbon cycle, particularly in the Tropics using OCO-2 v7. In addition, OCO-2 retrievals have been used in several inversion models. For example, Crowell et al. (2019) showed that with different assumptions (such as a large ensemble of atmospheric inversions using different CTM, data assimilation algorithms, and prior flux), OCO-2 posterior inferred fluxes globally agree with in-situ data, but that this agreement breaks down quickly at smaller space and time scales.

To finish regarding the data we are using in our study, Huijnen et al. (2016) and Patra et al. (2017) have shown that pyrogenic $CO_2$ emissions estimates from CO MOPITT data (through the use of emission factors) are consistent with OCO-2 measurements using a forward simulation with a CTM. With this in mind, and also that OCO-2 and MOPITT have similar vertical sensitivity for their retrievals of $CO_2$ and CO, we use these two data sets to constrain surface fluxes for these two tracers. Using $CO_2$ and CO together in this way is an important proof of concept for upcoming missions such as GeoCarb (Moore et al., 2018), which will measure both tracers from geostationary orbit over the Americas.





### 2.1.3   In situ data

The in situ $CO_2$ data used for assimilation come from 5 collections in ObsPack format (Masarie et al., 2014). These collections include :

- the obspack_co2_1_GLOBALVIEWplus_v5.0_2019-08-12 (Cooperative Global Atmospheric Data Integration Project, 2019) which contribute to 93% of all data.

- obspack_co2_1_NRT_v5.0_2019-08-13 (NOAA Carbon Cycle Group ObsPack Team, 2019) which provides near-real time
provisional observation and so the data did not get final quality control.

-obspack_co2_1_AirCore_v2.0_2018-11-13 which is provided by the balloon-borne AirCore instrument. This dataset includes almost the entire atmospheric column.

-obspack_co2_1_INPE_RESTRICTED_v2.0_2018-11-13 (NOAA Carbon Cycle Group ObsPack Team, 2018). This collection of data only comes from aircraft profiles at fives sites in Brazil.

-obspack_co2_1_NIES_Shipboard_v2.1_2019-06-12. The data come from 9 volunteer ships of opportunity operated by the Japanese National Institute for Environmental Studies (Tohjima et al., 2005; Nara et al., 2017).

These 5 collections provide around 540 assimilable observations per day. These $CO_2$ measurements are collected in flasks or by continuous analyzers at surface, tower, and aircraft sites (see Fig. 1) and are an important anchor for this exercise because their error characteristics are generally well-known, being directly established via calibration traceable to World Meteorological
Organization standards. Additionally, these measurements provide traceability to a long history of flux estimates derived from these data as an atmospheric constraint.

### 2.1.4   Observations for validation : TCCON data

We evaluate our posterior model mole fractions against retrievals from TCCON, which is a ground-based network of Fourier transform spectrometers established in 2004 and designed to retrieve atmospheric gases from NIR spectra (Wunch et al., 2011).
The global monthly means of the total column $CO_2$ measurements have accuracy and precision better than 0.25% (less than 1 ppm) relative to validation with aircraft measurements (Wunch et al., 2010, 2011). TCCON measurements have been used in several papers for validation of satellite measurements (e.g. Kulawik et al. (2016); Wunch et al. (2017); O'Dell et al. (2018); Kiel et al. (2019)). Our evaluation uses data from 23 operational instruments of TCCON globally. Table 1 lists all TCCON sites used for the evaluation and Fig. 2 represents the location of the sites over the globe.

### 2.2   Chemistry transport model TM5

We employ TM5 (Krol et al., 2005) and the Four-dimensional Variational (4DVAR, Meirink et al. (2008)) framework to link trace gas emissions to atmospheric tracer mixing ratios. Several inverse modelling studies have estimated CO emissions or $CO_2$ emissions using TM5-4DVAR (Hooghiemstra et al., 2011; Van Leeuwen et al., 2013; van der Laan-Luijkx et al., 2015; Nechita-banda et al., 2018; Basu et al., 2018; Crowell et al., 2018, 2019). TM5 is driven by 3-hourly offline meteorological fields from
the ERA-Interim (Dee et al., 2011) reanalysis of the European Centre for Medium range Weather Forecasts (ECMWF). We



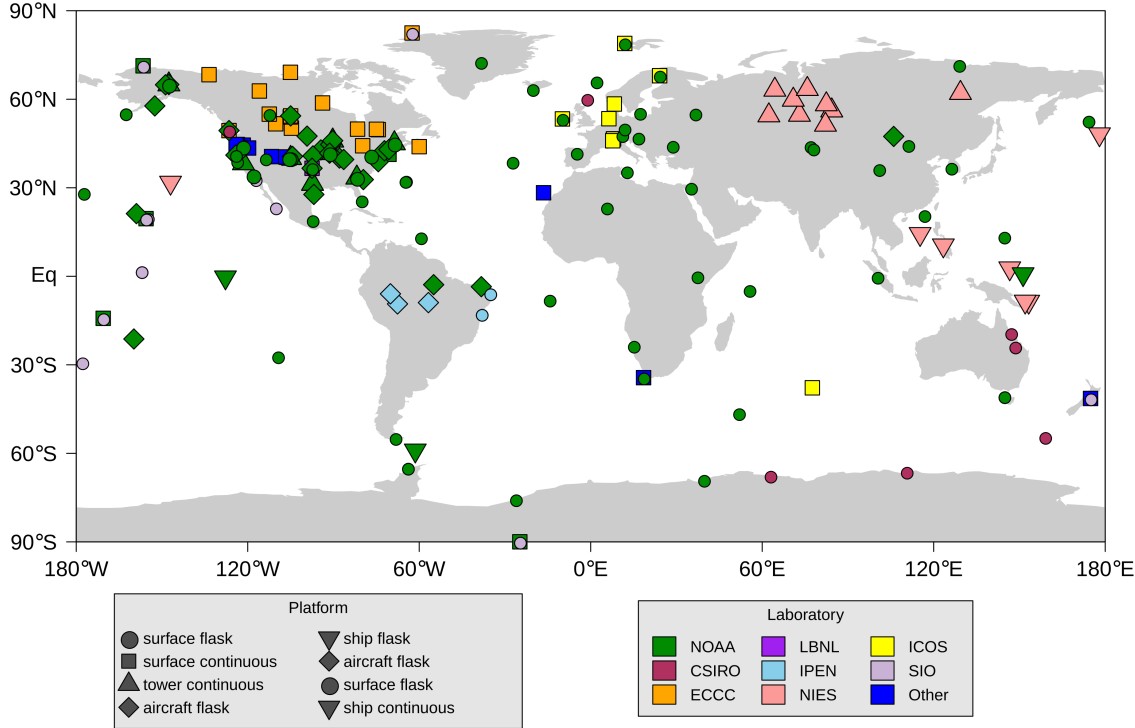

**Figure 1.** Distribution of assimilated in situ measurements around the world. The instrument platform is indicated by marker shape, whereas the color represents the laboratory collecting the data. NOAA is the United States National Oceanic and Atmospheric Administration, CSIRO is the Australian Commonwealth Scientific and Industrial Research Organisation, ECCC is Environment and Climate Change Canada, LBNL is the Lawrence Berkeley National Laboratory, IPEN is the Brazilian Instituto de Pesquisas Energeticas e Nucleares, NIES in the Japanese National Institute for Environmental Studies, ICOS is the European Union Integrated Carbon Observation System, and SIO is the Scripps Institute of Oceanography. Mobile shipboard programs are shown with a single marker at the mean location of the measurements. Figure from Jacobson et al. (2020a).

run TM5 on a 3°x2° horizontal resolution grid for the CO inversion and on a 6°x4° horizontal resolution grid for the $CO_2$ inversions with 25 vertical hybrid sigma-pressure levels. The initial condition for CO is globally constant to 80ppb, which is then combined with a 6 month spin-up to account for discrepancies from the real atmospheric distribution of CO. The initial global distribution of $CO_2$ is taken from the CarbonTracker (Peters et al. (2007) version CT2017, with updates documented at http://carbontracker.noaa.gov) posterior mole fractions. The CT2017 fields are constrained over the period 2000-2016 with data from the global in situ. Both inversions are run from July 1, 2014 until March 1, 2019, i.e. with six months of spinup and two months of spindown to avoid so-called "edge effects" affecting the period of interest from 2015-2018.

The CO sink from OH is represented in TM5 by a monthly OH climatology from Spivakovsky et al. (2000). This OH climatology is scaled by a factor 0.92 based on methyl chloroform simulations (Huijnen et al., 2010).



**Table 1.** Geolocation and reference of each TCCON station used for the evaluation section.

| TCCON sites | Country | Latitude | Longitude | Data revision | Reference |
|---|---|---|---|---|---|
| Eureka | Canada | 80.05N | 86.42W | R3 | Strong et al. (2019) |
| Ny-Ålesund | Spitsbergen | 78.9N | 11.9E | R0 | Notholt et al. (2014b) |
| Sodankylä | Finland | 67.4N | 26.6E | R0 | Kivi et al. (2014) |
| Białystok | Poland | 53.2N | 23.0E | R2 | Deutscher et al. (2019) |
| Bremen | Germany | 53.10N | 8.85E | R0 | Notholt et al. (2014a) |
| Karlsruhe | Germany | 49.1N | 8.4E | R1 | Hase et al. (2015) |
| Paris | France | 48.8N | 2.4E | R0 | Té et al. (2014) |
| Orléans | France | 47.9N | 2.1E | R1 | Warneke et al. (2019) |
| Garmisch | Germany | 47.5N | 11.1E | R2 | Sussmann and Rettinger (2018) |
| Park Falls | Wisconsin (USA) | 45.9N | 90.3W | R1 | Wennberg et al. (2017) |
| Rikubetsu | Japan | 43.5N | 143.8E | R2 | Morino et al. (2018b) |
| Lamont | Oklahoma (USA) | 36.6N | 97.5W | R1 | Wennberg et al. (2016) |
| Anmeyondo | Korea | 36.5N | 126.3E | R0 | Goo et al. (2014) |
| Tsukuba | Japan | 36.1N | 140.1E | R2 | Morino et al. (2018a) |
| Edwards | California (USA) | 34.2N | 118.2W | R1 | Iraci et al. (2016) |
| Caltech | California (USA) | 34.1N | 118.1W | R0 | Wennberg et al. (2014) |
| Saga | Japan | 33.2N | 130.3E | R0 | Kawakami et al. (2014) |
| Izaña | Tenerife | 28.3N | 16.5W | R1 | Blumenstock et al. (2017) |
| Ascension Island | UK | 7.9S | 14.3W | R0 | Feist et al. (2014) |
| Darwin | Australia | 12.4S | 130.9E | R0 | Griffith et al. (2014a) |
| Réunion Island | France | 20.9S | 55.5E | R1 | De Mazière et al. (2017) |
| Wollongong | Australia | 34.4S | 150.9E | R0 | Griffith et al. (2014b) |
| Lauder 125HR | New Zealand | 45.0S | 169.7E | R0 | Sherlock et al. (2014) |

## 2.3 Inversion system and analyses

We use TM5-4DVAR to infer fluxes as the long window ensures a long term spatio-temporal distribution of the trace gas in the atmosphere that is consistent with multi-year flux distributions. The TM5-4DVAR model is used in this study to estimate CO and $CO_2$ emissions with the corresponding satellite and in situ. TM5-4DVAR utilizes optimal estimation to minimize a Bayesian cost function (Rodgers, 2000) in order to find the state vector corresponding to surface emissions of CO or $CO_2$ that best match the observations within their relative uncertainties. The a posteriori flux is found by minimizing the mismatch between the forward model and the observations weighted by the inverse of the observation error covariance matrix **R** while staying close to a set of a prior fluxes weighted by the inverse of the a priori error covariance matrix **B**. These matrices are discussed in more detail in Section 2.3.1. Although the CTM is quasi-linear, the observation operator for CO is not. Since we use log(VMR) for the MOPITT retrievals as the CO observable, the non-linear optimizer M1QN3 from Gilbert and Lemaréchal



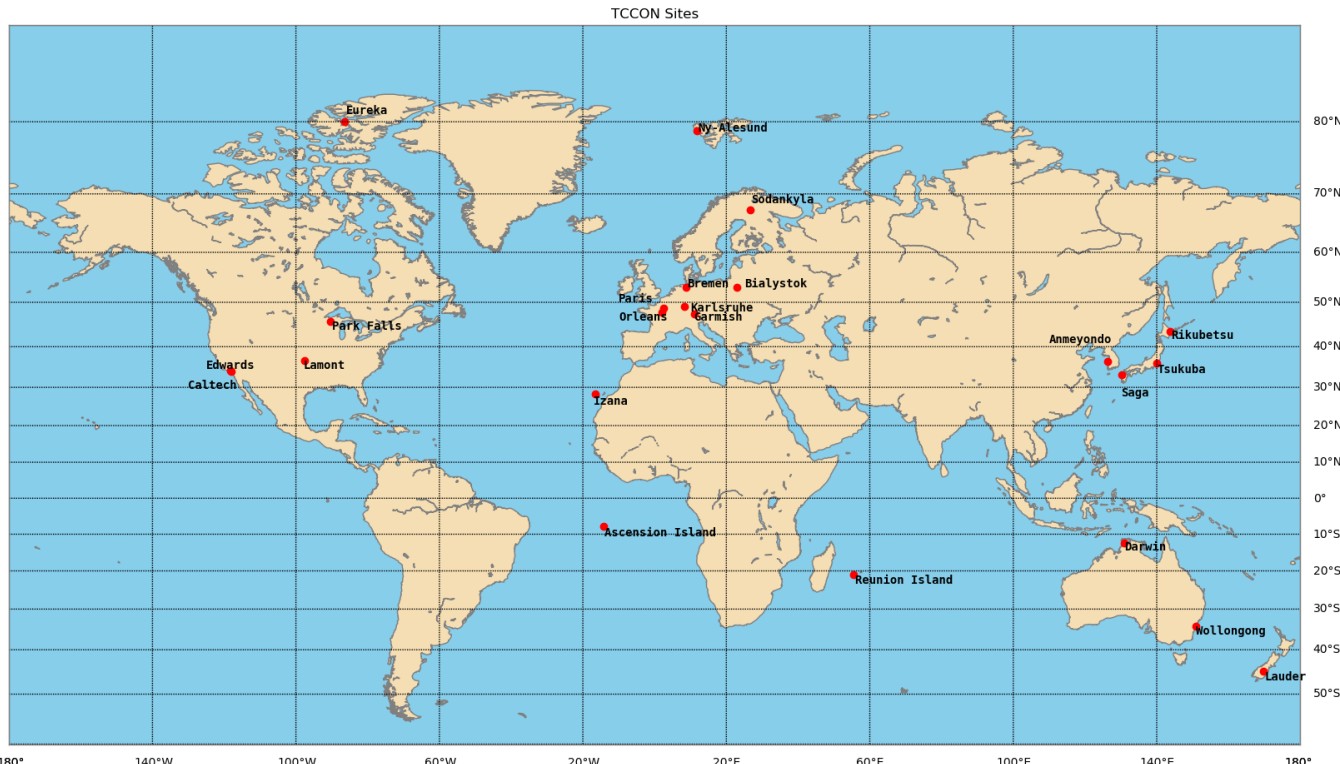

**Figure 2.** Localisation of the TCCON site used in this study over the globe.

(1989) is employed. Both the transport and observation operators for $CO_2$ are linear, and so we employ the conjugate gradient method to estimate the optimal $CO_2$ emissions, the implementation of which is described in great detail in Basu et al. (2013).

### 2.3.1 A priori information

#### a) CO parameterizations

Injection heights, in the CO inversion, are computed using IS4FIRES (Integrated System for Wild-Land Fires, http://is4fires.
fmi.fi/, Sofiev et al. (2013)). This emission database is driven by re-analysis FRP obtained from MODIS (Giglio et al., 2006)) instrument on board Aqua and Terra satellites.

Three emissions categories are used for the CO inversion : anthropogenic (which represents the combustion of fossil fuels and biofuels), natural sources (direct CO emissions from vegetation and oceans) and biomass burning (vegetation fires). In our configuration, we only optimize biomass burning emissions.

Anthropogenic emissions come from MACCity inventory (Granier et al., 2011). This inventory provides projected inter-annual trends in the anthropogenic CO emissions.


The oxidation of $CH_4$ and non-methane volatile organic compounds (NMVOCs) such as isoprene ($C_5H_8$) and monoterpene ($C_{10}H_{16}$) leads through photolysis and reaction with OH to the formation of formaldehyde, the major chemical source of CO (Atkinson, 2000). Isoprene is a member of the group of hydrocarbons known as terpenes. It is explicitly taken into account in

TM5 as it represents the dominant biogenic NMVOC emission (Guenther et al., 2012). Isoprene and monoterpene oxidation schemes are based on the mechanisms developed by Yarwood et al. (2005). Isoprene contributes to 9-16 % of the global CO burden (Pfister et al., 2008). They account for 68% in TM5 of the biogenic NMVOC emissions that react to produce CO. By contrast, monoterpene accounts for 15% (Tsigaridis et al., 2014). The chemical production of CO coming from the oxidation of methane and NMVOCs requires monthly 3-D CO fields produced by oxidation of biogenic and anthropogenic hydrocarbons

including $CH_4$. We use chemical production of CO from the oxidation of $CH_4$ and from NMVOCs by using a 2010 simulation with the full chemistry version of TM5 (Huijnen et al., 2010).

A priori biomass burning CO emissions are taken from the GFED4.1s inventory (van der Werf et al., 2010) and incorporate a daily cycle.

The first version of GFED was released in 2004. Since then, several improvements have been incorporated into GFED.

Improvement on the mapping of burned area from active fire data in GFED2 (Giglio et al., 2006) was no longer necessary when the MODIS product became available for GFED3 (Giglio et al., 2009). Burned area particularly affects the spatiotemporal variability of carbon emissions during fires. This spatiotemporal impact has been implemented in GFED with biogeochemical modeling framework providing estimation of biomass combustion over different vegetation types (Giglio et al., 2013). All GFED versions are then based on the Carnegie-Ames-Standford Approach (CASA) model adjusted to account for fires (see

van der Werf et al. (2004) and van der Werf et al. (2017) for more details). The most recent versions (GFED4 and GFED4.1s which includes small fire burned area) modified the burned-area-to-burned fraction conversion, which have been shown to increase burned area and fire carbon emissions of 11% in GFED4.1s compared to GFED3 (van der Werf et al., 2017) at the global scale. Liu et al. (2017b) found that with the omission of small fires in GFED3, global fire emissions are underestimated. Accounting for small fires increased global burned area and carbon emissions by 35% (Randerson et al., 2012), and improved

the agreement of spatial distribution between active fires and burned area over regions with large fires such as savanna fires and boreal forests. Including small fires in GFED amplifies emissions over regions where drought stress and burned area varied considerably from year to year in response to, for instance, the El Niño Southern Oscillation (ENSO). The GFED4.1s burned area are based on fire observations from the MODIS instrument with a 500 m horizontal resolution. The MODIS burned area data have been combined with active fire data from Tropical Rainfall Measuring Mission (TRMM), the Visible and Infrared

Scanner (VIS), and the Along-Track Scanning Radiometer (ATSR), three other instruments on board with MODIS. GFED4.1s has a spatial resolution of 0.25°x0.25° and includes estimates of burned area, carbon emissions, monthly biospheric carbon fluxes based on the CASA-GFED4s framework and the information from small fire fraction. Additionally, monthly carbon emissions of GFED4.1s distinguish between different vegetation types such as boreal forest, agricultural waste, temperate forest, deforestation, peat-land, and savanna.

The prior uncertainty covariance matrix **B** is described by a product of uncertainty variance and correlations in space and time. Spatially, a Gaussian correlation length scale of 1000 km is used, while we assume the prior errors have a temporal





correlation scale of 4 days. As in Hooghiemstra et al. (2011, 2012) and Nechita-Banda et al. (2018), an uncertainty standard deviation of 250% has been applied for the grid-scale prior of biomass burning emission. This large uncertainty is assumed since these inventories support large uncertainties. As mentioned by Hooghiemstra et al. (2011), this yields between 40-100%

of prior continental emissions uncertainty, depending on the region.

**b) Computation of an optimized CO$_2$ Fire Prior**

In this section, we describe the computation of our optimized prior fire emission we will use for observing impact of CO fire emissions in the posterior CO$_2$ biospheric fluxes. For each pixel of CO posterior biomass burning emissions constrained by MOPITT retrievals, we break down the global CO fire emissions using the GFED4.1s partitioning in order to obtain posterior

simulated CO fire emissions for each monthly vegetation type (savanna, boreal forests, peat, temperate forests, deforestation and agriculture waste). Figure 3 shows for instance the GFED vegetation type for each year over the lands. Each color represents pixel with one or several vegetation types.

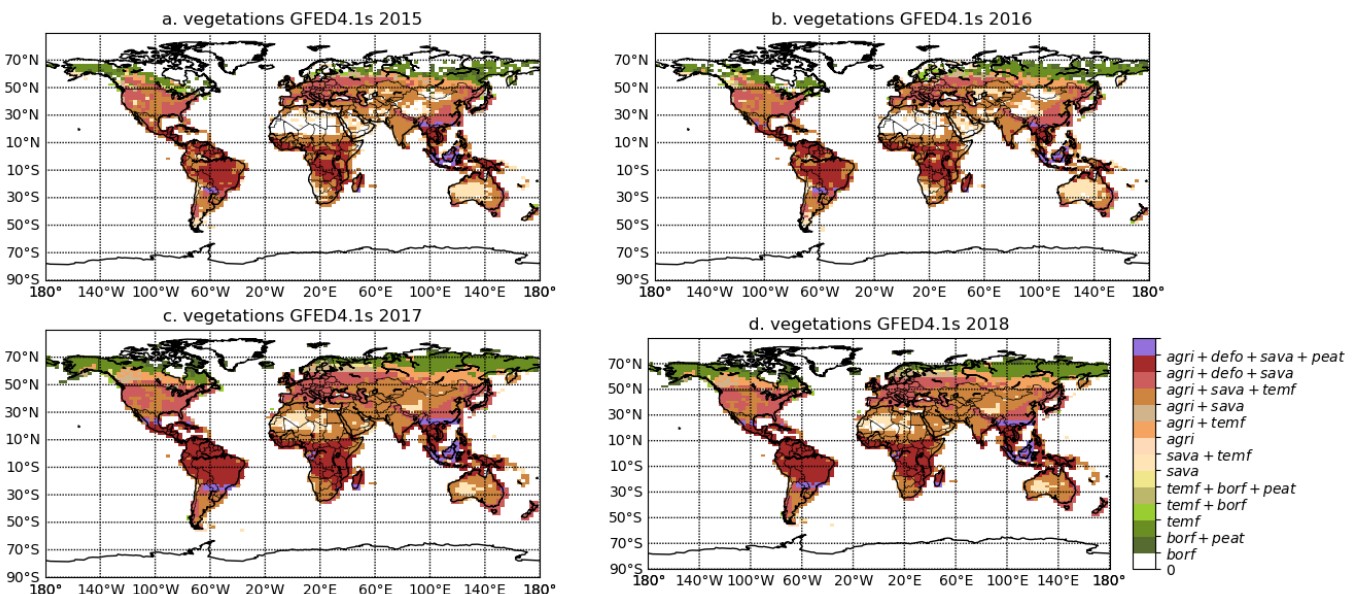

**Figure 3.** GFED vegetation types by pixel on a 3°x2° grid for a) 2015, b) 2016, c) 2017, and d) 2018

We first calculated the emission ratios $ER_{(CO/CO_2)}$ which allowed us to convert CO fire emissions to CO$_2$ fire emissions. The emission ratios are computed using the GFED emission factor for each vegetation type (annotated $i$ in the equation 1).

Following the equation of Andreae and Merlet (2001) :

$$ER_{(CO/CO_2)_i} = \frac{EF_{CO_i}}{EF_{CO_2{}_i}} \cdot \frac{M_{CO_2}}{M_{CO}} \qquad (1)$$





with $M_{CO} = 28$ g.mol$^{-1}$ and $M_{CO_2} = 44$ g.mol$^{-1}$ the molecular weights of CO and CO$_2$; $EF$ are the emission factors for each vegetation types describes in table 2. Emission factors allow us to estimate trace gases emissions from carbon losses during fires (Andreae and Merlet, 2001).

**Table 2.** Emission Factors in $g.kg^{-1}.DM^{-1}$ for CO and CO$_2$, and emission ratios $ER_{(CO/CO_2)}$ available from GFED4.1s by vegetation types

|  | Savanna | Boreal forests | Temperate forests | Deforestation | Peat | Agriculture waste |
|---|---|---|---|---|---|---|
| EF$_{CO}$ | 63 | 127 | 88 | 93 | 210 | 102 |
| EF$_{CO_2}$ | 1686 | 1489 | 1647 | 1643 | 1703 | 1585 |
| ER$_{(CO/CO_2)}$ | 0.059 | 0.134 | 0.084 | 0.089 | 0.194 | 0.101 |

We then aggregated the 0.25°x0.25° vegetation fraction partitioning of GFED to create vegetation masks at a 3°x2° grid. We applied this aggregated mask to the posterior simulated CO fires, which partitioned the posterior CO fires by vegetation types (we took care to divide the emissions by number of vegetation found by pixel). Finally, the emission ratio for each vegetation type was divided to the posterior CO fire partitioned as used in Christian et al. (2003) and Basu et al. (2014). This results in monthly CO$_2$ emission per vegetation type per grid box. Finally, we sum up these emissions across all surface types in order

to get monthly total optimized prior CO$_2$ biomass burning emissions that we called "FIREMo". We used this FIREMo as the fire emissions in CO$_2$ inversions along with a re-balanced respiration and NEE in order to have the net fluxes in balance with fire estimate, using the parameterization described in the following section (2.3.1.c).

### c) CO$_2$ parameterizations

CO$_2$ emissions are separated into four categories: anthropogenic sources, ocean fluxes, terrestrial biosphere fluxes and fires.

The anthropogenic emissions are taken from the Open-source Data Inventory for Anthropogenic CO$_2$ 2018 (ODIAC2018; Oda and Maksyutov (2011)). A diurnal cycle is imposed by TIMES product with weekly scaling as suggested by Nassar et al. (2013). Fossil fuel emissions are not optimized in the CO$_2$ inversions, as is typical of global tracer transport inversions (e.g. Peylin et al. (2013); Crowell et al. (2019)). Ocean fluxes are taken from the climatological fluxes described in Takahashi et al. (2009). They are assumed to have an uncertainty variance of 50%.

Terrestrial biosphere fluxes and fire emissions are difficult to disentangle a priori, and some inverse modeling studies (e.g. Crowell et al. (2019)) choose instead to report the net land fluxes. Likewise, some global land flux estimates such as GEOS-Carb (Ott, 2020) use fire estimates to revise the terrestrial biosphere flux estimates through modification of ecosystem respiration. We take a similar approach, starting with the gross primary production and respiration estimates from the GEOS-Carb CASA-GFED 3-hourly 0.5°×0.625° (Ott, 2020). We then modify the net flux in concert with each fire emissions estimated as follows.





Net ecosystem exchange (NEE) in the GEOS-Carb product is expressed as the sum of heterotrophic respiration (Rh) and gross ecosystem exchange (GEE) :

$$NEE3 = Rh3 + GEE3 \tag{2}$$

We modified the respiration from GEOS-Carb (*Rh3*) to create respiration estimates for GFED4.1s (*Rh4*) and MOPITTOpt (*RMo* linked with the FIREMo we calculated previously) which balance with the updated $CO_2$ fire estimate so that estimated

respiration increases (decreases) in the places where each fire estimate is smaller (larger) than FIRE3 (GFED3):

$$Rhx = Rh3 + max(FIRE3 - FIREx, 0) \tag{3}$$

where $x$ is either "4" or "Mo". The resulting net ecosystem exchange, i.e. NEE4 or NEEMo, is then computed using (2), with *GEE3* used for both NEE4 or NEEMo equations. We then apply a simple rebalancing scheme to match the yearly NOAA global mean growth rate ($AGR_{NOAA}$) for 2015-2018 (see table 3), since

$$AGR = \overline{NEE} + \overline{FIRE} + \overline{FOSSIL} + \overline{BIOFUEL} + \overline{OCEAN} \tag{4}$$

where $\overline{X}$ represents the global total annual flux for category $X$. We use ODIACv2018 (with 2018 repeated for 2019) to compute the global fossil fuel totals (values in the table 3), BIOFUEL from the CASA land biosphere model (van der Werf et al., 2004), and a fixed annual value of -2.6 PgC/yr for the oceans for simplicity, and we use FIRE from each source described above.

**Table 3.** Global total fossil fuel emissions, fire and biofuel emissions and AGR from NOAA in PgC/yr.

|  | 2014 | 2015 | 2016 | 2017 | 2018 | 2019 |
|---|---|---|---|---|---|---|
| BIOFUEL | 0.478 | 0.479 | 0.476 | 0.486 | 0.486 | 0.486 |
| FOSSIL FUEL | 9.85 | 9.89 | 9.91 | 10.07 | 10.28 | 10.28 |
| FIRE3 | 1.83 | 2.03 | 1.63 | 1.97 | 1.97 | 1.49 |
| FIRE4 | 1.88 | 2.09 | 1.73 | 1.69 | 1.64 | 0.34 |
| FIREMo | 1.62 | 1.82 | 1.47 | 1.58 | 1.56 | 1.44 |
| $AGR_{NOAA}$ | 4.3 | 6.3 | 6.06 | 4.54 | 5.05 | 5.55 |

Any mismatch between the AGR derived from our prior flux estimates ($AGR_x$) and $AGR_{NOAA}$ is assumed to be due to an

incorrect estimate of global NEE. We adjust NEE at each gridpoint with a simple scaling on global total respiration (i.e. Rhx) and GEE:

$$AGR_{NOAA} - AGR_x = (1+k)\overline{Rhx} + (1-k)\overline{GEE}. \tag{5}$$

where $x$ is either 3, 4, or Mo, depending on whether we use FIRE3 (GFED3), FIRE4 (GFED4.1s), or FIREMo. This equation is easily solved for $k$ using each annual global total, and the resulting corrections are applied to each 3-hourly gridded value

of GEE and respiration for each choice of fire emissions. In this way, the a priori global $CO_2$ emissions are ensured to match



**Table 4.** Experimental Configurations

|  | CMS-GFED3 | GFED3re | GFED4re | MOre |
|---|---|---|---|---|
| ODIAC Fossil | X | X | X | X |
| FIRE3 (GFED3 Fires) | X | X |  |  |
| FIRE4 (GFED4.1s Fires) |  |  | X |  |
| FIREMo (MOPITT Fires) |  |  |  | X |
| Takahashi Ocean Flux | X | X | X | X |
| Annual Total Matches AGR$_{NOAA}$ |  | X | X | X |

the annual global growth rate as measured by NOAA regardless of the fire emissions assumed, as well as a spatial pattern and seasonality that aligns with bottom up models' GEE and Rh estimates as closely as possible.

We run the $CO_2$ inversions with the re-balanced terrestrial biosphere net flux NEEre$x$ corresponding to either GFED3, GFED4 or FIREMo priors. In order to assess the impacts of the rebalancing procedure, we perform a fourth experiment that assumes the GEOS-Carb NEE as the prior biosphere flux with GFED3 fires, and the results are labeled in what follows as CMS-GFED3. All $CO_2$ FIRE priors include both biomass and biofuel burning. The details of each of the 4 priors and the experimental configurations are detailed in table 4.

## 3 Results

In this study, several inversions were performed with the TM5-4DVAR inversion framework. MOPITT v8 L2 CO data were assimilated to constrain fire emissions of CO. Separately, OCO-2 v9 XCO$_2$ and in situ $CO_2$ are used to constrain net fluxes of $CO_2$ (see table 5).

**Table 5.** $CO_2$ inversions used in this study with the observations assimilated, NEE and fire emissions associated.

| Inversions name | NEE used | Fire used | data assimilated |
|---|---|---|---|
| GFED4re | NEEre4 | FIRE4 | OCO-2 |
| GFED3re | NEEre3 | FIRE3 | OCO-2 |
| MOre | NEEreMo | FIREMo | OCO-2 |
| CMS-GFED3 | GEOS-Carb | FIRE3 | OCO-2 |
| IS4re | NEEre4 | FIRE4 | in situ |
| IS3re | NEEre3 | FIRE3 | in situ |
| ISMOre | NEEreMo | FIREMo | in situ |
| ISCMS | GEOS-Carb | FIRE3 | in situ |

We optimized CO biomass burning emissions and $CO_2$ biospheric and oceanic emissions on a weekly basis. For the OCO-2 and in situ $CO_2$ inversions, we use four different sets of prior biosphere and fire emissions (see section 2.3.1).





In section 3.1, we examine the impacts of assimilating MOPITT v8 XCO observations on inferred fire CO emissions after

vegetation partition and the comparison with the prior GFED4.1s. CO emissions categorized by vegetation combustion types
are the one used for developing FIREMo.

In section 3.2, we focus on the $CO_2$ inversions. As fire emissions are not optimized in $CO_2$ inversions, we will examine how
posterior NEE varies according to observation constraint and the imposed fire fluxes. We first compare (in 3.2.1) the variability
and magnitude between the biospheric priors used in the $CO_2$ inversions over the globe. zonal bands and over the same regions

as in Crowell et al. (2019), which are Transcom (Gurney et al., 2002) regions that are further subdivided at the equator (which
we will called OCO-2 MIP regions). The regions are defined in Fig. 4 and are composed of 16 land regions and 11 ocean
regions. We will focus on regions over land, as we are mostly interested in the interplay between assumed fire emissions and
inferred NEE. We then investigate the covariation of imposed $CO_2$ fire emissions and optimized NEE with OCO-2 data and
in-situ data (3.2.2). Finally, posterior simulated $CO_2$ mixing ratio are validated against TCCON data over the globe in section

370    3.2.3.

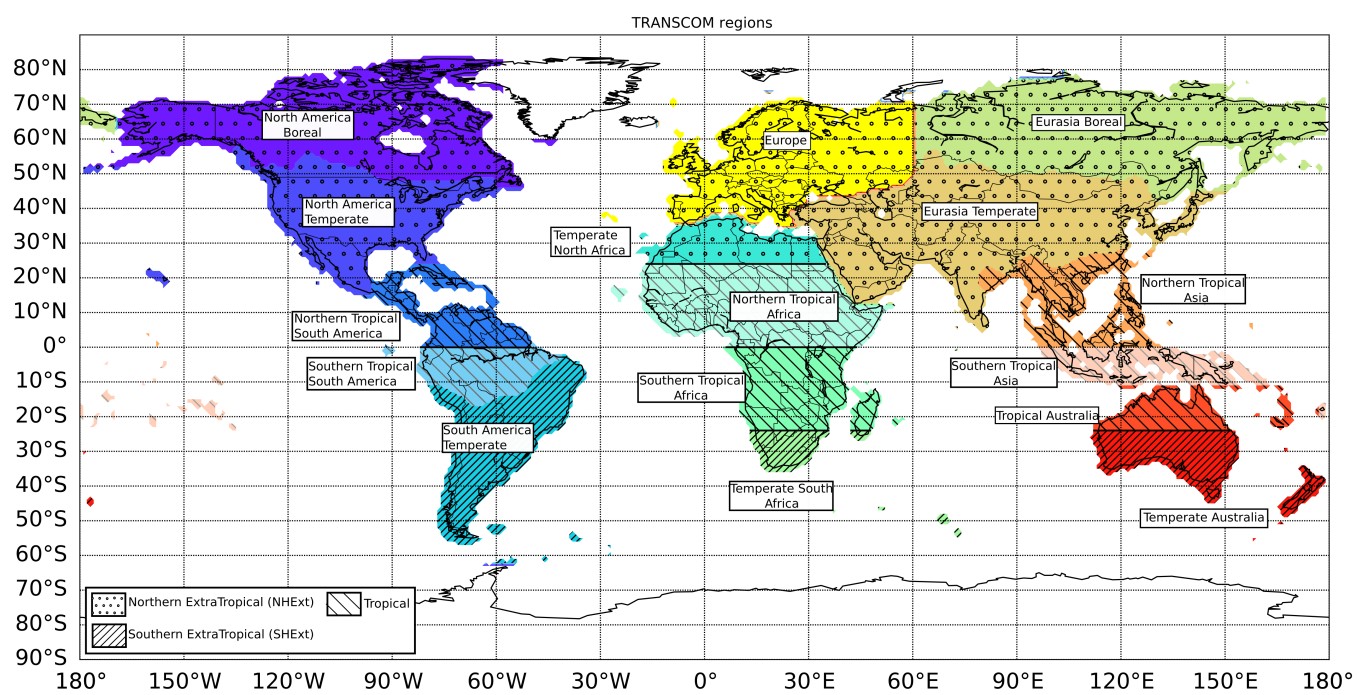

**Figure 4.** OCO-2 MIP regions for which prior and posteriors gridded fluxes are aggregated for comparison





**Figure 5.** CO fire emissions by vegetation combustion types over the OCO-2 MIP regions between fire priors (hatch bars) and fire posterior for a) 2015, b) 2016, c) 2017 and d) 2018. Vegetation vegetation types are representing by colors : agriculture in gray, deforestation in yellow, savanna in dark-red, temperate forest in blue, peat land in red and boreal forests in green. Emissions are annually in TgCO/yr.



## 3.1 Fire CO emissions partitioned by vegetation combustion : MOPITT optimized emissions versus GFED4.1s emissions

Figure 5 shows the annual CO fire emissions partitioned by vegetation combustion between MOPITT CO optimized fire emissions and fire prior (GFED4.1s) over the globe and by OCO-2 MIP regions. We can see globally, that depending on the regions, the assimilation of MOPITT data implies less or more CO fire emission than the prior. For instance, in 2015, the Northern Tropical South America shows more CO fire emissions for all types of vegetation (agricultural, savanna and deforestation) with the posterior while the Southern Tropical South America gives more CO with the prior for the same types of vegetation. For some other regions, such as Northern Temperate America and Temperate North Africa, MOPITT and GFED4.1s give almost the same amount of CO fire emissions for all types of vegetation. Over Temperate Northern Africa, this can be a result of bias satellite data due to cloud coverage, giving the CO posterior emissions closer to the prior GFED4.1s. However, for Northern Temperate America, the prior might be well enough constrained and validated over this region, to give similar CO fire emissions than the posterior CO. For Temperate North Africa, MOPITT posterior fires remain close to the prior GFED4.1s estimates, meaning that the inferred emissions are consistent with GFED4.1s. This region is known to have a lot of Saharan dust transported across the Atlantic ocean and towards Europe most of the year and to be largely cloudy during the wet season of the African monsoon (from May to August), which could explain the posterior emissions being close to the prior. This is also the case for Northern Tropical Africa, however, MOPITT posterior fires has lower emissions than the prior GFED4.1s estimate. But, we still need further investigation over Northern Tropical Africa to understand why GFED4.1s and MOPITT are different each other. Tropical South America is also known to have a cloudy coverage limiting satellite observations, but over this region we only observe similar emissions between the prior and the posterior for Northern Tropical America, even if MOPITT has slightly higher emissions, while for Southern Tropical America, differences between the prior and the posterior are strong.

Differences between MOPITT posterior emissions and GFED4.1s are particularly large for Tropical Africa and for the boreal forests of North America and Eurasia with difference of more than 15 TgCO/yr. On average for the 2015-2018 period of study and for the regions Europe, North Tropical South America, Temperate South America, Temperate Australia and North Tropical Asia, MOPITT gives higher emissions for the deforestation types, savanna and agricultural waste. This is the case for the Northern Tropical Southern America and Australia, which are dominated by trees and grass savanna. This characteristic has been also observed by the previous study of Pechony et al. (2013) who compared the older version of GFED with MOPITT and TES emissions for the period 2005-2008.

For Eurasia Temperate, discrepancies appear between MOPITT and GFED4.1s for all type of vegetation and for all years. These regions are characterized by agricultural waste and savanna, as well as temperate forests, regarding the GFED4.1s vegetation types. The discrepancies observed between MOPITT and GFED4.1s could then be that the vegetation type is not well represented for these regions. As mentioned in Pechony et al. (2013), agriculture and savanna vegetation types might not be the dominant burning vegetation type over North Africa and the Middle East, since these regions have seen an increase in croplands area well control by human activities and so burn rarely. However, Kazakhstan is a region of temperate forest often





dominated by fires (Venevsky et al., 2019), a characteristic that MOPITT posterior emissions seem to observe as mush as the prior GFED4.1s. We can also observe that over Northern Tropical Asia, MOPITT fire emissions are higher than GFED4.1s (see Fig. 5 and Fig.A2). This is observed for all years, where MOPITT emissions are almost 5 TgCO/yr (2 TgCO/yr) for savanna (for the other vegetation types) higher than from GFED4.1s. As mentioned in Pétron et al. (2002) and Arellano et al. (2004), CO emissions in Northern Tropical Asia are significantly underestimated in current inventories.

Previous studies have shown that peat area and depth, producing large amount of carbon ($\sim 0.60$ PgC/yr which represents 26% of the total carbon fire emissions, Nechita-Banda et al. (2018)), were found to have significant uncertainties in Indonesia in the emissions inventories (Lohberger et al., 2017; Hooijer and Vernimmen, 2013). The added value of MOPITT CO observations is especially important for peat lands situated over Indonesia and Tropical Asia (see Fig. 3). MOPITT tends to capture smaller fires due to the large field of view (Pechony et al., 2013). In addition, MOPITT can capture the seasonality of peat fires

over Indonesia in comparison to GFED4.1s. Figure A1 shows for Southern Tropical Asia (mainly visible in 2015 due to the large emissions) that GFED4.1s have a fire peak earlier than MOPITT. van der Laan-Luijkx et al. (2015) and Nechita-banda et al. (2018) hypothesized that GFED4.1s might not capture the timing of emissions over area with peat fires due to the use of burned area, which may be more sensitive to the initial stages of the fire than to the continued burning. Knowing this fact, and the large uncertainties attributed to fire emissions inventories in the Tropics, will be important for the following of this study

in the computation of FIREMo and its uses in the $CO_2$ inversions.

Overall, GFED4.1s gives higher CO fire emissions than MOPITT with some exception where MOPITT gives higher CO fire emissions particularly during the 2015-2016 El Nino period such as Northern Tropical South America and for agricultural waste, savanna and deforestation of Northern Tropical Asia.

## 3.2   OCO-2 and in situ $CO_2$ inversions with different fire and NEE priors

We performed inversions with different $CO_2$ fire and NEE priors assimilating: i) OCO-2 $XCO_2$ retrievals and ii) $CO_2$ in-situ data. See table 5 for details of the eight $CO_2$ inversions.

To investigate the uncertainty in inferred $CO_2$ emissions arising from the selection of fires, we perform $CO_2$ inversions with three different global gridded fire estimates. The first is taken from the GEOS-Carb GFED3 product (Ott, 2020), which we label "FIRE3"; for the second we use GFED4.1s, denoted "FIRE4". The third set, described in Section 2.3.1.b and denoted

"FIREMo", is created by first optimizing CO emissions with MOPITT observations and then converting them to $CO_2$ emissions using the landcover ratios and parameters in GFED4.1s. The methodological differences between FIRE3 (GFED3) and FIRE4(GFED4.1s) are described in section 2.3.1.a. Figure 6 shows annual differences between FIRE3 and FIRE4 from 2015 through 2018 over the OCO-2 MIP regions. We note that regional differences are as large as 140 TgC per year, or roughly $\sim 10\%$ of the annual global fire emissions budget which has been estimated to $1.6 \pm 0.7$ GtC/yr (Friedlingstein et al., 2020).

Additionally, the size and sign of the differences varies by year and by region. For instance, FIRE3 (GFED3) generally predicts higher $CO_2$ emissions over the Boreal regions, while FIRE4 (GFED4.1s) largely predicts more fire emissions from the Northern midlatitudes. This is consistent with differences between the two models, i.e. GFED4.1s uses a different set of emission factors separating trace gas emissions and aerosol from boreal forest to Temperate forests (Akagi et al., 2011; van der Werf



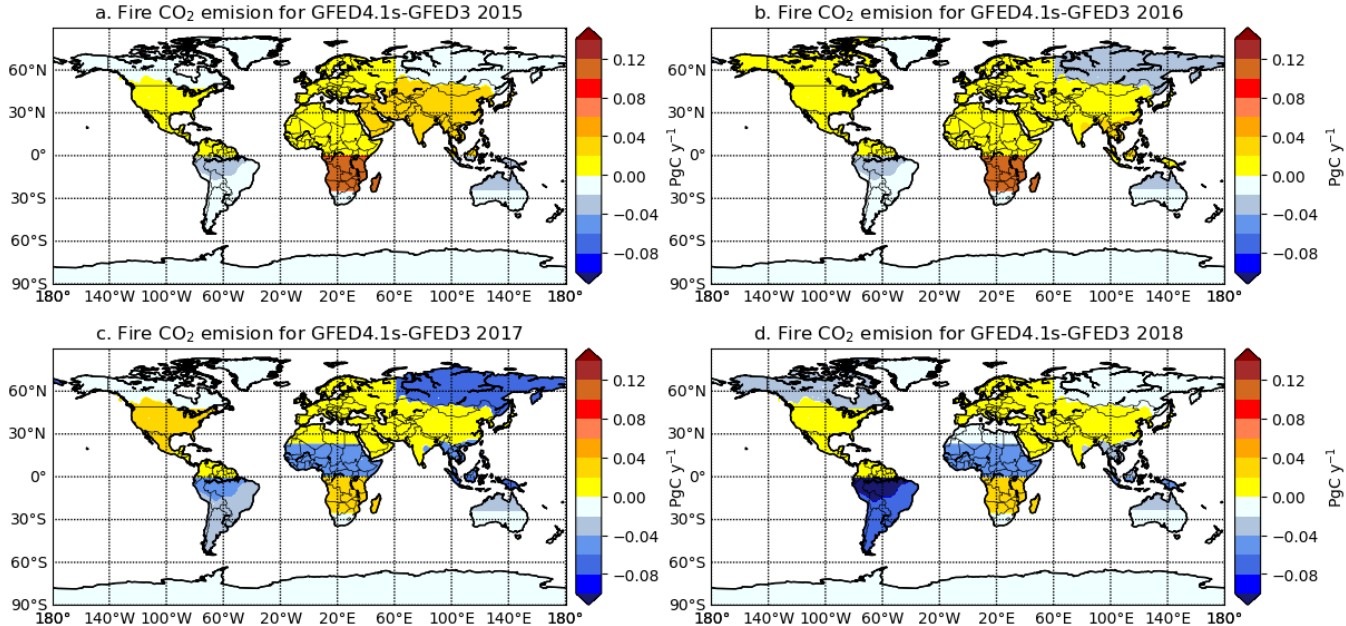

**Figure 6.** Annual differences between FIRE4 and FIRE3 in PgC/yr over the regions of Fig. 4 for a) 2015, b) 2016, c) 2017 and d) 2018.

et al., 2017). van der Werf et al. (2017) have shown that GFED3 does not capture the different patterns of fire severity between the boreal regions of North America and Eurasia and the differences between Boreal and Temperate forests fires (which could explain the large difference between FIRE4 and FIRE3 in Fig.6). In addition, van der Werf et al. (2017) found that including small fire burned area in GFED4 doubled the burned area in Temperate North America and Europe compared to GFED3. Interestingly, the differences in the Tropics have a pronounced meridional structure, where GFED4.1s predicts smaller emissions in South America, Tropical Asia, and North Africa (after 2016), and larger emissions in Southern Tropical Africa. The addition of small fire burned area included in GFED4.1s has a strong impact in the Southern Tropical Africa regions where agricultural waste burning and shifting cultivation are important drivers of fire activity. van der Werf et al. (2017) have shown that the increase of burned area in these regions were associated with small fire burned area from the last GFED version. Small fires linked with deforestation and agricultural waste are also important over the Indonesia, however deforestation activity decreased of almost 50% in 2017 and 2018 thanks to several Indonesian policies in order to prevent forest fires and land clearing with particularly the new law avoiding to clear forest for oil palm plantations (Global Forest Watch, 2020). This might explain the decrease in fire emissions over Southern Tropical Asia in 2017 and 2018 with GFED4.1s, in addition that 2017 and 2018 were not impacted by the 2015 El Niño event where large fires burned in Indonesia.

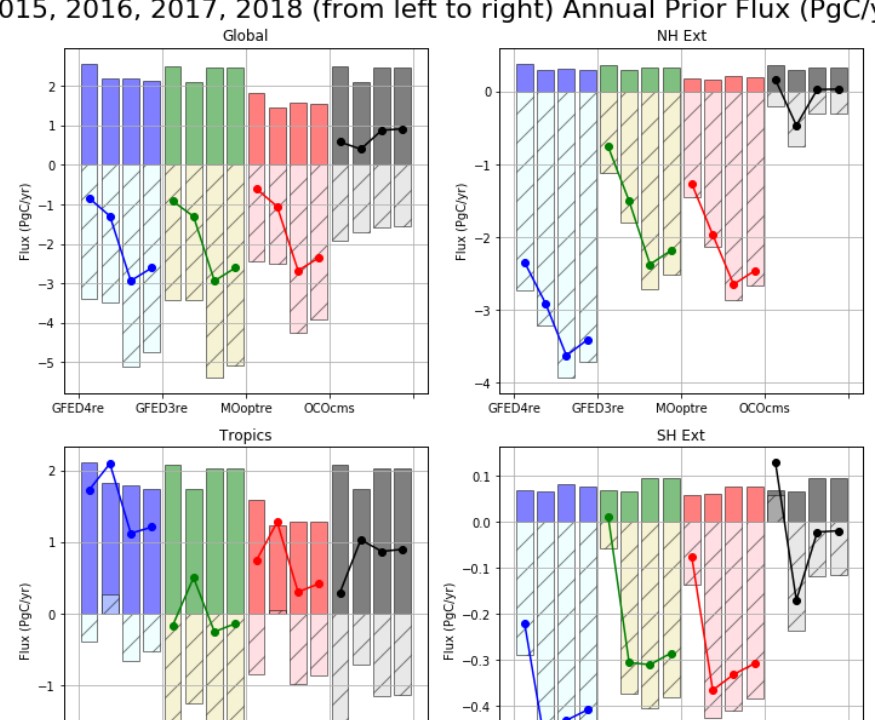

**Figure 7.** Annual prior $CO_2$ emissions, in global and by latitude bands, used later in top-down inversions. Annual net flux (lines), NEE (bars with hatches) and FIRE (bars with darker colors) prior emissions are represented from 2015 through 2018 (left to right) between GFED4.1s (blue), GFED3 (green), MOPITTopt (red) and CMS (black).

### 3.2.1 Prior NEE and fires $CO_2$ fluxes

Figure 7 shows annual $CO_2$ emissions for the prior estimates in global and by latitude bands from 2015 through 2018. The
prior categories shown are fire, NEE and net fluxes for the prior GFED4, GFED3, MOPITTopt and CMS-GFED3. At the global scale, the three non-CMS priors (GFED3, GFED4, MOPITTopt) give the same net sink of carbon for the whole period (matching the NOAA AGR with the same assumed fossil and ocean fluxes), decreasing from 2015 through 2018, while the CMS-GFED3 prior gives sources of carbon increasing in time. Global fire emissions as well as net carbon fluxes, of the non-CMS priors, are within the spread of estimation of the Global Carbon Budget estimated by Le Quéré et al. (2018) and Bastos
et al. (2018). The decrease (increase) in NEE sinks (net sources) for CMS-GFED3 prior during the period of study is driven by the fact that the product imposes a long term balance between fire and NEE and is not constrained to match the measured growth rate of $CO_2$ in the atmosphere. The discrepancy shows up particularly in the Northern Hemisphere Extra-Tropics (NH Ext) and Southern Hemisphere Extra-Tropics (SH Ext) where sinks of CMS are generally smaller than the others.





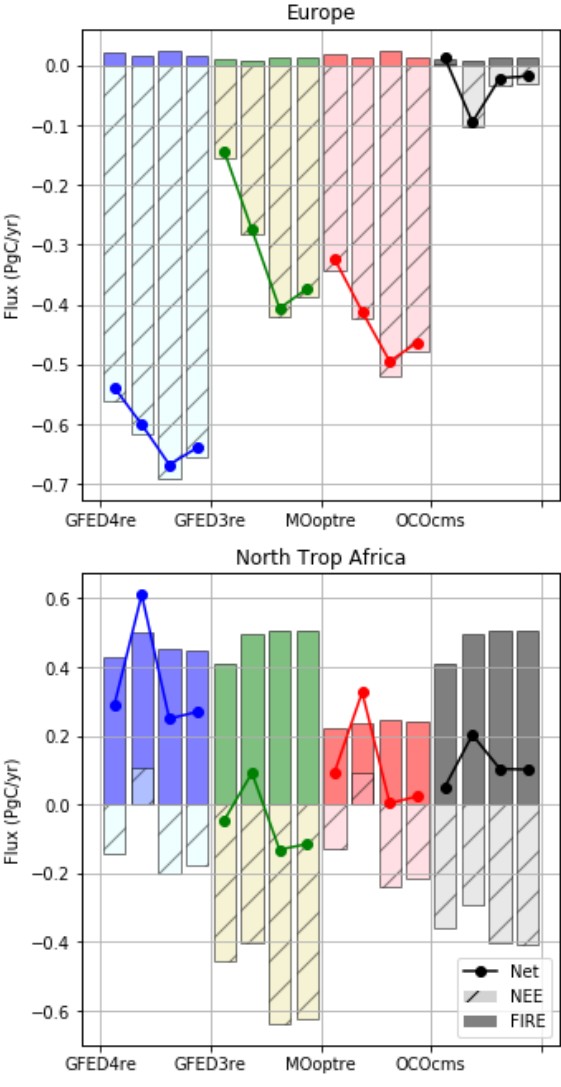

**Figure 8.** Same as Fig. 7 but for Europe and Northern Tropical Africa regions.





At latitude scale, we can see that the non-CMS priors give more net sinks in the Northern (between -1 PgC/yr and -4 PgC/yr)
and Southern Hemispheres Extra-Tropics, while CMS has lower values and even net sources in 2015 for both hemispheres
and in 2018 as well for the Northern Hemisphere. These differences between the priors is also observed for the Northern and
Southern Hemisphere OCO-2 MIP regions (Fig. A5). In addition, we can observe that GFED4 and MOPITTopt priors have
a deeper Northern Hemisphere sinks than GFED3 (particularly observed for Europe and Northern Asia, Fig. A5 and Fig. 8
and Fig. 9), which is balanced by stronger net sources over the Tropics (coming mainly from Southern Tropical Africa and
Southern Tropical Asia respectively, Fig. A5). For Tropical South Africa, we can see a discrepancy in the zonal flux distribution
between GFED3 and, GFED4 and MOPITTopt which could result due to the adjustment and modification of the respiration
with the fires for GFED4 and MOPITTopt which is not considered for GFED3 as we took the fire and respiration directly from
CMS (GEOS-Carb). The scaling of GFED3 GPP and respiration to match the global AGR yields deeper biogenic sinks over
the Tropics than with all the other priors. We can also observed for Southern Tropical Africa that FIRE4 has larger fires than
FIREMo, consistent with the fact that GFFED4.1s CO was observing higher CO fire emissions than MOPITT (Fig. 5 and Fig.
A1)

In addition, the global fires emissions indicate that FIREMo observes less emissions compared to all other priors, a difference
coming from tropical regions. These lower fire emissions observed by FIREMo in the Tropics come mainly from Tropical
Australia (with values in 2015 of ∼0.05 PgC/yr for FIREMo compared to ∼0.07 PgC/yr for FIRE4 and ∼0.095 PgC/yr
for FIRE3), Tropical Africa (∼0.35 PgC/yr, ∼0.50 PgC/yr, ∼0.55 PgC/yr respectively for FIREMo, FIRE4 and FIRE3) and
Southern Tropical South America (∼0.1 PgC/yr, ∼0.17 PgC/yr, ∼0.2 PgC/yr respectively for FIREMo, FIRE4 and FIRE3).
Characteristics that came with the differences between GFED4.1s CO and MOPITT CO fire emissions (see Fig.5 and Fig.A2).
But we observe a larger fire emissions with FIREMo in Southern Tropical Asia (fire emissions of ∼0.37 PgC/yr for FIREMo
compared to ∼0.35 PgC/yr for FIRE4) and in Northern Tropical Asia (∼0.22 PgC/yr for FIREMo, ∼0.21 PgC/yr for FIRE4,
and ∼0.19 PgC/yr for FIRE3).

We can then see the impact of including MOPITT CO data in the $CO_2$ fire priors compared to the emission inventories
known to have biased carbon emissions in Tropical Asia. The larger emissions with FIREMo compared to FIRE4 over tropical
Asia comes mainly from some specific vegetation. The main vegetation type in this region is savanna and we can observe that
for the $CO_2$ prior emissions, FIREMo has the higher flux for Northern tropical Asia (Southern tropical Asia) compared to
FIRE3 and FIRE4 (FIRE4 respectively) for savanna but also for agriculture and deforestation (see Fig. A3). Another impact of
including CO MOPITT data in the $CO_2$ fire priors appears in variability. As already observed with the CO emissions (Fig. A1)
and discussed in van der Laan-Luijkx et al. (2015) and Nechita-banda et al. (2018), the seasonality of fires over tropical Asia
is better capture with MOPITT than with the emission inventories for peat lands. However, this is not only true for peat but
also for other vegetation types. For savanna, agriculture and peat lands, FIREMo observed a peak in fire seasonality after the
peaks observed with the GFED (Fig. A4). This is particularly true for the 2015 El Niño fires but less for the fires that occurred
in 2017 and 2018. In this period, FIREMO does not observed as much fire emissions as the GFED with a similar seasonality.
The difference in seasonality for 2015 could be a result of the large and intense fires during the El Niño event burning larger





**Figure 9.** Same as Fig. 7 but for Northern Asia and Tropical Asia regions.





regions and releasing more smokes which could have impacted the MODIS burned area data used in GFED but probably also the MOPITT retrievals. Further investigation are then needed to study this region.

### 500  3.2.2  Comparison between the $CO_2$ posterior fire and biospheric emissions : impact of re-balanced NEE and fire prior in $CO_2$ posterior emissions

We assimilated OCO-2 and in situ data separately in order to assess the impact of these data in conjunction with different fire emissions and corresponding land flux priors. In all inversions, only NEE and ocean fluxes have been optimized.

**a) Global and latitudinal flux**

**Figure 10.** Global and latitudinal $CO_2$ posterior emissions between OCO-2 and in situ inversions as GFED4re (in blue), GFED3re (in green), CMS (in black) and MOre (in red). Annual fluxes are displayed from 2015 (left) through 2018 (right). FIRE emissions are darker colored bars, NEE fluxes are hatched bars and lines depict the net land fluxes.





Figure 10 shows global and latitudinal annual net fluxes, FIRE and NEE fluxes for both OCO-2 and in situ inversions. We can see that globally, net fluxes for OCO-2 posterior emissions across the different priors are consistent. The sinks seem to adjust the different fire contributions. This is also observed for the in situ inversions. The in situ inversions give, compared to OCO-2 inversions, a $\sim 1$ PgC/yr deeper net sink in 2015 and 2016, and a $\sim 2$PgC/yr deeper net sink for 2017 and 2018, which originates from a weaker source (or sink) in the Tropical regions than the OCO-2 informed inversions. These differences are in agreement with the differences between OCO-2 and IS inversions the larger ensemble of inversions detailed in Crowell et al. (2019) and Peiro et al. (2022).

The Northern Hemisphere Extra-Tropics (NH Ext) posterior fluxes are consistent across the different priors for both observation constraints, which is not surprising given the good coverage of the in situ observations in this region. For OCO-2 inversions, we can see small variations from year to year (going to -2.5 PgC/yr in 2015 through -2.75 PgC/yr in 2016) except for 2018 where the net sink drops to -2 PgC/yr. The net sinks observed with the in situ inversions are weaker than OCO-2 for 2016, 2017, and 2018, and the year-to-year variations are significantly larger than the OCO-2 results. Similar behavior is observed for the SH Ext and opposite behavior for the Tropics with larger sources for the OCO-2 inversions, which could be related to cross-talk between the zonal bands given the sparse coverage of in situ data in the Tropics. However the net fluxes are closer between in situ and OCO-2 inversions over the NH Ext than for the Tropics, where the in situ fluxes are generally 1-1.5 PgC less than the OCO-2 fluxes. The consistency across the priors for the Northern latitude bands are also observed in the simulation study of Philip et al. (2019) where they used different NEE priors to observe the impact on the OCO-2 posteriors.

Comparing the posterior emissions (Fig. 10) with the prior emissions (Fig. 7), we can see an impact from the observations assimilated on the CMS prior which shift significantly toward the other posterior fluxes at global and latitudinal scales. CMS posterior emissions seem to have slightly weaker sinks than the posterior using the rebalanced priors. The imposed AGR seems then to have an impact at latitudinal scales. For the Tropics, we can again observe a consistency in OCO-2 across the priors. MOre and ISMOre have a smaller sink in 2015 (with sources for OCO-2 inversion) compared to the other inversions in order to balance the 0.5 PgC/yr smaller fires that FIREMo gives. This balance was also observed for the priors (see Fig. A5). The range of net flux observed with all OCO-2 inversions are consistent with other studies (Palmer et al., 2019; Crowell et al., 2019). Further, the intense fires and $CO_2$ sources related to the 2015 El Niño Oscillation over the Tropics and mainly Indonesia might not be seen with in situ data due to their weak coverage in these regions. This could then explain why we observe stronger sinks with in situ than OCO-2 posterior NEE emissions. Comparing the OCO-2 inversions with the in-situ ones shows that the in situ net fluxes are totally different with a Tropical sink in all years except in 2015 and 2016 which has a net source. 2017 and 2018 are particularly strong sinks of $\sim$ -0.5 PgC/yr. These deeper sinks would explain part of the larger global sink from the in situ inversions. We will see in the next subsection that these sinks with in situ are coming from the Northern Tropical regions such as Northern Tropical Africa and Northern Tropical Asia (see Fig. 11 and Fig. 13). As observed for the previous Extra-Tropical band, SH Ext shows similar fluxes across the priors for each data constraint. However 2016 is adjusted downward significantly for the OCO-2 fluxes (between -0.4 PgC/yr and -0.6 PgC/yr) compared to the in situ fluxes (between -0.2 PgC/yr and 0.1 PgC/yr). In addition, while OCO-2 net fluxes have stronger sources in 2016 over the Tropics, they have a deeper sinks over the SH Ext than with the in situ fluxes. This result suggest a transport connection between the Tropics and SH Ext fluxes with the





OCO-2 inversions, where land coverage is limited and hence retrievals are sparser than the other regions. On the other hand, this does not seem to be the case in the in situ results, but we know that there are a few in situ data present in the SH Ext and so they have a different data constraint.

The MOPITT results look like the GFED4.1s results for the Tropical regions, while the GFED3 and CMS results look alike, suggesting the sensitivity in these regions to the fire prior, not only for IS but also for OCO-2 data constraint.

**b) Regional fluxes**

When we compare the posterior regional fluxes, we can see difference in the carbon balance.

If we first look the Northern Extra-Tropical regions (North America, North Asia and Europe), we can see that the OCO-2 inversions have deeper net sinks over the Boreal regions than with OCO-2 (see Fig. A6). The in situ data are placing almost all of the NH Ext sink over Northern Asia, while there are sources of carbon over North America for 2015. OCO-2 only sees

net sinks of carbon for all NH Ext. In-situ data do not have an homogenize coverage over the NH Ext band : large number of observations are situated over Temperate North America and Europe but are very sparse over the Boreal regions and Temperate Eurasia (see Fig. 1). The large differences in net sinks occur then over the regions with few data (Boreal and Eurasia regions). It is interesting also to see balance between the regions in Northern Hemisphere with Southern Hemisphere. For instance, it seems that the 2018 drop off sinks (starting in 2017 and so representing the "recovery" period) observed with both IS and OCO-

2 over North Asia is balanced by the Tropical Asia (North and South) where net fluxes go from sources to sinks. 2015 was a large net sources of carbon (due to intense fires) while 2016, 2017 and 2018 are deeper sinks with IS over Northern Tropical Asia and sinks with OCO-2 over Southern Tropical Asia. At the same time, posterior fluxes in Europe are anti-correlated with posterior fluxes over Northern Tropical Africa. The deeper sinks observed with OCO-2 in Europe are anti-correlated with the net sources observed in Northern Tropical Africa, where the post ENSO period has smaller sources in Northern Tropical Africa

linked with smaller sinks in Europe (Fig. 11). Emissions estimated observed with OCO-2 are more in line with previous studies even though the years of study were before 2015. Reuter et al. (2014) found, using GOSAT data, uptake of carbon per year around 1 PgC/yr which was 0.5 PgC/yr higher than expected from in situ inversions. However, as mentioned in Reuter et al. (2017), there is a lack of carbon budget information over Europe and there is hence no estimate that can be refuted at present.

We then can see that this deeper sinks in the NH Ext with OCO-2 are in fact balanced by the Tropics. The weaker net

sink observed with IS compared to satellite data was also observed with Houweling et al. (2015) who were using GOSAT retrievals instead of OCO-2 for the 2009-2010 period. Additionally, what is observed here between IS and OCO-2 inversions was also observed in the study of Peiro et al. (2022) who found that by using OCO-2 v9 the inversions showed weaker sinks over Northern Asia but a deeper sink over Europe and Northern America than with the in situ inversions. We can see that our inversions here are within the estimates observed in the study of Peiro et al. (2022). However, for Europe, we can see that the

variability in our priors is different than the ones used in Peiro et al. (2022). Our re-balanced priors give the deepest sink in 2017 (in 2016 for CMS prior) which is observed as well in the posteriors net fluxes using OCO-2 and it is in opposition of the OCO-2 inversions of Peiro et al. (2022) which have deeper sinks in 2016. This is due to stronger fire emissions in 2017





**Figure 11.** Same as fig.10 but for North Tropical Africa and Europe.





compared to the other years balanced with the respiration, and the differences between the two studies could be due to the re-balanced respiration.

In average for all regions, we can see a disagreement between IS and OCO-2 inversions but an agreement across priors within each observational constraint. Additionally, comparing the different priors, we can see that inversions using the FIREMo are closer to the ones using FIRE4 while CMS is closer to the inversions using FIRE3, i.e. the fires seem to have more impact than the rebalancing to match the global AGR. We can observe that for almost all regions, the sinks with NEE4 and NEEMo are deeper than with NEE3 and CMS but they are balanced with higher sources for the other regions that have net sources, regions
mainly over the Tropics.

        Focusing on the Tropical regions, we can first see that for Northern Tropical South America (Fig. 12), OCO-2 fluxes are consistent for each priors used with around 0.5 PgC/yr efflux during the El Niño period (2015-2016) and neutral emissions during the recovery period. IS fluxes are also strong during the El Niño period, but remain moderately high in 2017. As observed in the Fig. 1 of the paper of Peiro et al. (2022), which used the same set of IS data, the number of IS data does not decrease
significantly, meaning that changing observational coverage is not the cause of this behavior. One possible explanation is the lag between flux in the Tropics and observation by the in situ network, which could be aliasing flux signals in time, though this hypothesis is difficult to test. For Northern Tropical Africa (Fig. 11), as already mentioned above, large difference are seen between OCO-2 and IS inversions. OCO-2 net fluxes are strong with high sources of carbon between 0.5 PgC/yr and 1.5 PgC/yr. We can see also some fire-dependent differences : FIREMo and FIRE4 drop off for 2017 but FIRE3 driven fluxes
do not. This difference in 2017 is particularly observed with OCO-2. IS, on the contrary, give strong sinks in this region, the strongest one for all Tropical regions. Examining Fig. A5, we note the dependence of the IS posterior emissions known with the priors. Northern Tropical Africa is known to have very few IS data compared to the other Tropical regions (Fig. 1). Northern Tropical Asia (Fig. 13) shows agreements between the priors and OCO-2 data constraints for 2015 and 2016, but shows significant differences between OCO-2 and IS for 2017 and 2018. The IS and OCO-2 inversions all agree together that
there is a source of carbon in 2015 but OCO-2 inversions have a smaller sink in 2016 while it is a source with IS (smaller with IS3re and IScms), which seems to show an impact of the El Niño event and impact of the fire priors with IS, while for 2017 and 2018, IS particularly seems to show a stronger recovery in the region than observed by OCO-2. The sparse coverage of in-situ data over this region could explain the difference with OCO-2, though not specifically for 2017 alone, and hence further investigations are needed for this region.

Very large differences between the IS and OCO-2 inversions appears for Southern Tropical South America (Fig. 12) as well. Interestingly, the OCO-2 posteriors emissions seem to be closer to the priors than the IS are. One explanation for that has been mentioned previously in Peiro et al. (2022), where the cloud coverage above the moist Amazon decreases the amount of OCO-2 retrievals, while IS data are located more inside the moist Amazon. This difference in posterior flux could then come from different area of observation. In opposition to the other Southern Tropical regions, the ENSO signal appears in 2016 instead
of 2015 for OCO-2. This region follows the inter-seasonal variations of the Northern Tropical regions, which also see highest emissions in 2016.





**Figure 12.** Same as Fig.10 but for North and Tropical America regions.





**Figure 13.** Same as Fig.10 but for North Asia and Tropical Asia regions.





Southern Tropical Africa is almost neutral in the OCO-2 posterior fluxes (see Fig. A6). We can see the large balance between the intense fires and the respiration, which are anti-correlated in their variability. GFED4re and MOre have larger sources than GFED3re and CMS. With the IS, there is large variation across the priors where IS4re and ISMOre both constrain a source of carbon for the whole period, while ISCMS and IS3re have smaller source of carbon and even a sink in 2016 and 2017. These differences seem to suggest that both fires and respiration are especially important when observational coverage is limited.

Finally, Southern Tropical Asia (Fig. 13) shows difference among the prior ensemble for both OCO-2 and IS inversions with particularly differences in the fires for 2015. The inversions adjusted the sinks for the MOre and GFED4re inversions to be larger than the two other inversions in order to accommodate the smaller fires observed with FIREMo and FIRE4. This is not observed however for the IS inversions which just show sources of carbon for both ISMOre and IS4re while the inversions constrained with the GFED3 fires (IS3re and IScms) give sinks of carbon showing smaller net carbon. The impact of the fires over this region seems to have a strong impact with both data constraint. The IS inversions using FIREMo and FIRE4 show then large sources in 2015, which are not observed with FIRE. If we compare the posteriors with the priors, we can in fact see that the IS tends to be closer to the priors than the OCO-2 inversions. This suggest that for this region as well, the few amount of IS data might explain this result and the larger amount of OCO-2 seems to better constrain the posterior fluxes.

In summary, we observe consistent differences between the Northern latitudinal regions and the Tropics, where the respiration tends to be higher over the Northern latitudes than the Tropics (which has larger sources), and IS has higher NEE than OCO-2. Differences in the priors used show impact particularly over the Tropics where both GFED4 and MOPITT observe large fires emissions. The MOPITT results look like the GFED4.1s results for the Southern Tropical regions, while the GFED3 and CMS results look alike, suggesting the sensitivity in these regions to the fire prior, not only for IS but also for OCO-2 data constraint.

### 3.2.3 Validation against TCCON data

As mentioned previously, most of the differences observed between in situ and OCO-2 inversions could be due to their respective coverage. in situ measurements have less data over the Tropics and Southern Hemisphere than OCO-2 retrievals. However, besides the spatial coverage, satellite retrievals might be affected, particularly over the Tropics, by the consistently cloudy region known as the Inter-Tropical Convergence Zone (ITCZ) as well as aerosols from biomass burning or dust (such as over and near the Sahara). It is then important to validate the OCO-2 and in situ posterior simulated mixing ratios against independent data. In this section, in order to explore the accuracy in the posterior fluxes, we evaluated the posterior fluxes by sampling the resultant concentrations for comparison with TCCON measurements. All posterior mixing ratios have been sampled around TCCON retrieval locations and times using the appropriate averaging kernels.

For evaluation of our CO posteriors and priors emissions with TCCON (not shown here), we found that biases between MOPITT CO posterior simulated mixing ratio and TCCON were lower than biases with the CO priors with on average a $\sim 5$ ppb reduction each year. Additionally, for the 2015-2018 period, the posterior biases were $\sim 7$ ppb underestimated TCCON values while the priors were $\sim 13$ ppb overestimated TCCON values.



**Figure 14.** CO$_2$ mixing ratio (in ppmv) of TCCON data (black) and the prior (a, b and c) and posterior (g, h and i) using MOre (red), GFED4re (green), GFED3re (blue) and CMS (gray) over three regions: Northern Hemisphere Extra-Tropical (left plots), Tropics (middle plots), and Southern Hemisphere Extra-Tropical (right plots). Differences against TCCON are shown in plots d, e, and f for the priors and j, k and l for the posteriors. For the second and fourth rows, OCO-2 posterior simulated mixing ratio are in plain lines and IS posterior simulated mixing ratio are in dashed lines. There are 16 TCCON sites for the northern hemisphere, 3 sites for the tropics and 4 sites for the southern hemisphere.



Figure 14 shows biases between prior and posteriors simulated mixing ratio ($XCO_2$) of the different $CO_2$ inversions against TCCON data by latitudinal bands : Northern Hemisphere Extra-Tropics (NH), Tropics (T), and Southern Hemisphere Extra-Tropics (SH). The number of sites by latitude used for the validation are referenced in the figure caption. The priors used for GFED4re, GFED3re and MOre inversions have similar errors and have particularly less biases over Southern Hemisphere than over the Northern latitudinal (see table 6). CMS has the largest biases with large $XCO_2$ overestimation of TCCON values (two

times more than biases observed with the three other priors) with biases of 4.82 ppmv in NH and 4.28 ppmv in the Tropics. CMS prior has in addition, the largest standard deviation values compared to the other priors and the lowest coefficient of correlation (table 6). Improvements of biases and standard deviation with the GFED3re prior compared to CMS which also use FIRE3 as fire prior, are likely due to the re-balanced respiration that match the NOAA growth rate. This re-balanced respiration and growth rate matches have also been used for GFED4re and MOre priors. However, all priors $XCO_2$ seems to have a positive

trend with increase of biases over the time, particularly pronounced for CMS $XCO_2$.

    The three mixing ratio of the re-balanced priors are relatively similar, with the MOre prior showing more biases than GFED for the Northern Hemisphere and less biases than GFED4 but more than GFED3 in the Southern Extra-Tropical hemisphere, and almost similar biases than GFED4 in the tropics. However, the tropics have only 3 TCCON sites for validation. Validation over this latitudinal band needs to be viewed with this in mind. In Southern Hemisphere, MOre prior has smaller biases than

GFED4re. This prior biases improvement might result from the optimized CO fire emissions in the MOre prior, which, has already mentioned, the CO posterior emissions were higher than the CO prior emissions (temperate South Africa and temperate Australia). However the larger biases present in the $CO_2$ priors with MOPITT fire compared to the GFED priors could come form the underestimation of CO emissions observed with the CO posterior emissions over Boreal forests (CO biases are ∼4ppb lower with XCO posterior than prior for Eureka site but ∼5ppb higher for Ny-Alesund and Sodankyla, not shown here).

We observed in the results section that posterior fluxes had similarity across the priors for each data constraint for SH Ext (see Fig. 10) but 2016 is adjusted downward significantly in the OCO-2 fluxes. We observe, in Fig. 14.l, a larger negative bias for OCO-2 than for IS particularly in 2016. For NH Ext, we observed previously (see Fig. 10 for North America and Europe mainly), a strong sink for OCO-2 over the period compared to IS, which observed stronger year-to-year variability. When evaluating with TCCON data (Fig. 14.j), we can see that OCO-2 has lower biases in 2015-2016 but higher biases for the

2015-2018 period and underestimates the concentration for almost the whole period compared to IS.

    The posterior $XCO_2$ are in better agreement with TCCON measurements than the priors. Additionally, all standard deviation and coefficient of correlation are similar between all inversions with slightly larger standard deviation for the IS inversions than for the OCO-2 inversions. We can also see that all posteriors match the variability of TCCON compared to the priors. Biases observed with CMS have been greatly reduced through the inversion, showing biases of the same order than the other

inversions.





**Table 6.** Bias in ppmv, standard deviation and coefficient of correlation between prior and posterior simulated mixing ratio, and TCCON by latitudinal bands such as NH (Northern Extra-Tropics), SH (Southern Extra-Tropics) and T (Tropics) over the period 2015-2018.

| | | priors | | | | | | | |
|---|---|---|---|---|---|---|---|---|---|
| | Latitudinal bands | GFED3 | GFED4 | MOPITT | CMS | | | | |
| Bias | NH | 1.57 | 1.29 | 1.85 | 4.82 | | | | |
| | T | 1.49 | 2.05 | 2.07 | 4.28 | | | | |
| | SH | 0.84 | 1.45 | 1.41 | 3.27 | | | | |
| Std | NH | 3.66 | 3.67 | 3.79 | 5.05 | | | | |
| | T | 4.10 | 3.96 | 4.18 | 4.28 | | | | |
| | SH | 3.53 | 3.57 | 3.68 | 4.86 | | | | |
| Corr | NH | 0.982 | 0.981 | 0.978 | 0.948 | | | | |
| | T | 0.991 | 0.988 | 0.990 | 0.985 | | | | |
| | SH | 0.991 | 0.993 | 0.992 | 0.987 | | | | |
| | | posterior OCO-2 | | | | posterior IS | | | |
| | Latitudinal bands | GFED3re | GFED4re | MOre | CMS | IS3re | IS4re | ISMOre | ISCMS |
| Bias | NH | -0.26 | -0.27 | -0.27 | -0.27 | 0.12 | 0.12 | 0.12 | 0.12 |
| | T | 0.20 | 0.21 | 0.21 | 0.21 | 0.35 | 0.40 | 0.38 | 0.37 |
| | SH | -0.44 | -0.44 | -0.44 | -0.44 | -0.20 | -0.22 | -0.21 | -0.20 |
| Std | NH | 3.40 | 3.40 | 3.40 | 3.40 | 3.53 | 3.52 | 3.53 | 3.54 |
| | T | 3.33 | 3.34 | 3.34 | 3.34 | 3.43 | 3.41 | 3.42 | 3.46 |
| | SH | 2.84 | 2.83 | 2.84 | 2.83 | 2.93 | 2.93 | 2.93 | 2.92 |
| Corr | NH | 0.995 | 0.995 | 0.995 | 0.995 | 0.995 | 0.995 | 0.995 | 0.995 |
| | T | 0.994 | 0.995 | 0.995 | 0.994 | 0.994 | 0.995 | 0.995 | 0.994 |
| | SH | 0.994 | 0.995 | 0.995 | 0.994 | 0.996 | 0.996 | 0.996 | 0.996 |

## 4   Discussion

In this study, we have presented an optimized $CO_2$ fire prior flux based on emission ratio between $CO_2$ and CO that comes from optimized CO fire emissions using MOPITT CO retrievals. In addition, as fire emissions and plant respiration (and hence net fluxes) are difficult to disentangle a priori, we re-balanced the respiration with each fire prior and with the annual NOAA

growth rate. We then explored a range of NEE emissions based on different fire emissions including a $CO_2$ fire estimate calculated from CO fire emissions information in order to better constrain biospheric emissions. We focused our study for the period 2015-2018 to observe the impact of the El Niño event in 2015 and the recovery period which followed it.

Globally, and for most regions, we find that the dependence of the inversion results on prior emissions is of secondary importance when compared with the data constraint, in the sense that variations in posterior flux are much smaller across

different prior mean fluxes (and the different uncertainties that come from scaling the prior mean flux) as compared with



differences resulting from assimilating OCO-2 versus in situ data. There are exceptions, most notably in the Northern and Southern Tropics, where the in situ constraint is especially limited and the corresponding posterior annual fluxes vary by as much as 0.5 PgC, which is a large fraction of the expected total El Niño signal. This suggests that in situ constrained flux estimates in the Tropics are more sensitive to the assumed prior flux, of which fires are a significant component, and should

be assigned the appropriate amount of uncertainty in accordance with this finding. It also implies that while residual biases in satellite retrievals remain a key focus of the top-down inversion community, further work is needed to improve prior fluxes in Tropical regions as well as deploy more in situ measurements. Current efforts by multiple organizations should assist in that effort on a short time basis, but more investments in long term monitoring are needed (communication from Kathryn McKain). OCO-2 inversions are also sensitive to the prior assumption in Northern Africa, though to a lesser extent, as well as

in Tropical Asia. Tropical Asia have been particularly well studied in the past where Nechita-banda et al. (2018) and van der Werf et al. (2017) have shown the underestimation of GFED inventories over peat fires compared to space-based instruments such as IASI and MOPITT. This reinforces the need for better measurements and bottom up estimates of biospheric and fire fluxes in these Tropical regions. In 2015, during the onset of the El Niño event which caused intense fires over Indonesia, the fire estimated from MOPITT CO emissions are stronger than with GFED4 emissions but lower than GFED3 emissions.

As mentioned previously, we know that GFED4.1s has information of small fires compared to GFED3 which allow better accuracy particularly over the Tropics where peat fires are important. Over Southern Tropical and Northern Tropical Asia, the combination of the spatio-temporal variability of MOPITT CO fire and the GFED4.1s emissions information included in the prior fire emissions of the CO inversion might bring additional information in the emission ratio and hence in the fire prior used in $CO_2$ inversions. Indeed, fires over peat lands spread more during the El Niño event due to intense drought conditions

(Nechita-Banda et al., 2018). Consequently, they emit two to four times more CO than forest fires (Akagi et al., 2011) and contribute significantly to the exchange between terrestrial carbon stocks and the atmosphere by decreasing the uptake of atmospheric $CO_2$ by the biosphere. This is particularly shown for the IS inversions where IS4re and ISMore have sources of carbons compared to the IS constrained with the GFED3 fires, showing then higher net sources with GFED4 and MOPITT than with GFED3 fires. Moreover, the $CO_2$ posterior emissions using MOPITT CO information were able to catch the seasonality

of fires over Southern tropical Asia during the El Niño event as discussed in Nechita-Banda et al. (2018) and van der Laan-Luijkx et al. (2015) that the other priors using GFED inventory were not able to capture. It is thus important to include CO fire emissions over this region to improve estimates and constrain $CO_2$ NEE and Fire emission with both OCO-2 and IS data constraints. But uncertainty in our emission ratio remains when converting CO to $CO_2$ emissions in our prior. GFED vegetation partition only account for six different types of vegetation which might not be finer enough to represent all different types of

fuels. Additionally, the emission factors used in the emission ratio are characteristic of vegetation type but are not dependent of spatial or temporal scales. We know, for instance, that African savanna fires can go from flaming to smoldering, changing the combustion efficiency and then the gases emitted (Zheng et al., 2018). This could explain the differences observed over some regions of the Tropics between the priors using CO fire emissions and the other prior fire emissions. Further works are needed to improve emission ratios and particularly emission factors over different spatial and temporal scales.





The data used to constrain the inversions are very important as we were able to see the differences between OCO-2 and IS inversions over this tropical region where the source is about 0.4 PgC/yr higher with IS than with OCO-2. This bring us to the importance of data assimilated in the inversions but also about the priors used in the inversions regarding the different sectors (fire and terrestrial emissions).

The difference in partitioning of fluxes in latitude and longitude for the different data constraints is not a new observation,
and fits the findings of the v7 OCO-2 MIP Crowell et al. (2019) and previous studies comparing GOSAT and in situ data (Reuter et al., 2014; Houweling et al., 2015)) as well as of the v9 OCO-2 MIP, an extension of the v7 OCO-2 MIP (Peiro et al., 2022). More specifically, the OCO-2 data constrain a stronger Northern Extra-Tropical sink in concert with a strong tropical source, while the in situ data generally constrain a weaker Northern sink and neutral Tropical flux, or even a sink. While the Northern Extra-Tropics are relatively densely sampled by the in situ network, Schuh et al. (2019) found a strong sensitivity of
flux estimates to model transport, particularly in the vertical and meridional transport of $CO_2$. Though we utilized only TM5 in these experiments, the findings here are consistent with those found in their study.

Within each zonal band, there are disparities in how the flux is partitioned, but these are again driven more strongly by data constraint than by prior. In the Northern Extra-Tropics, the partitioning of the sink across continents is more robust in the OCO-2 posterior results than IS, with both North America and North Asia accounting for a sink of 1-1.5 PgC and Europe a bit
less. The IS results suggest a very strong sink in North Asia ( < -1.5 PgC) relative to North America (-0.6 PgC with around 0.5 PgC of source in 2015) and Europe (-0.5 PgC to -0.1 PgC). When we look at the North Tropics, all inversions see a source in North Tropical South America, and this persists longer in the IS measurements into 2017 while ending sooner in the OCO-2 inversions. The largest difference is in North Tropical Africa, where OCO-2 sees a persistent source with varying magnitude, while IS fluxes depict a sink in 2015 and 2018 and near-neutral flux in 2016-2017. Given our current understanding of Africa's
response to the El Niño from previous studies measurements (Gloor et al., 2018), strong sinks in Northern Africa are unlikely. Interestingly, IS fluxes imply a post El Niño recovery in the North Tropical Asia with net sources in 2015-2016 and sink of more than 0.4 PgC per year in 2017-2018, while OCO-2 sees a progressive transition and smaller response from -0.2 PgC/yr in 2016 through -0.4 PgC/yr in 2018. The Southern Tropics show most of the source activity from OCO-2 (southern tropical South America and southern tropical Asia), while again the IS data constrain a sink, except in Southern Tropical Asia.

Returning to the question of importance of the prior, it would seem that the simulation experiments in Philip et al. (2019) hold for our experiments as well, i.e. that OCO-2 inversions are relatively insensitive to the prior in most regions.

A generally accepted (though not documented) assertion is that a minimal amount of data is required to constrain the global growth rate, and yet we see here that OCO-2 and the global in situ network do not see the same global annual flux, even assuming the same transport and prior flux that matches the NOAA AGR. Certainly some of this mismatch is due to sampling
differences, as most of the in situ measurements assimilated here are taken in the atmospheric boundary layer in the Northern Extra-Tropics, whereas OCO-2 measurements are globally distributed, but seasonally varying coverage. Persistent transport biases as well as satellite retrieval errors likely play a factor in this global offset, though further investigation is necessary to assess the relative importance of each.



## 5 Conclusions

In this study, we have explored the potential of using $CO/CO_2$ emission ratio to add CO fire information in $CO_2$ inversions in order to better estimate and constrain $CO_2$ biospheric emissions. Fires have the potential to influence inter-annual variability and long-term trends in atmospheric $CO_2$ concentrations and particularly alter the seasonal cycle of net biome production. CO measurements are available with high precision from space and bring more accuracy in CO fire emission estimates. Including more accurate fire emissions in $CO_2$ inversions could improve the estimates of $CO_2$ land fluxes relative to a $CO_2$ inversion

without the added information of CO. In this paper, we showed how we added on global scale $CO/CO_2$ emission ratio and its respective re-balanced respiration with fire and NEE with annual NOAA growth rate, and its value for $CO_2$ inversions.

We performed several $CO_2$ transport inversions assimilating separately OCO-2 data and in situ measurements from 2015 through 2018. We found that OCO-2 and in situ net fluxes have, even if with a difference, a better agreement at global scale as observations are dense enough to constrain the fluxes than at latitudinal and regional scale. Differences in net fluxes are

particularly important over the Tropics not only between OCO-2 and in situ inversions but also between the different priors used. Discrepancies between in situ and OCO-2 inversions occurred over Northern Tropical Africa where OCO-2 inversions have shown net sources while in situ inversions have shown sinks. However, over Southern Tropical regions, discrepancies appear between the different set of priors, with higher net sources observed with the inversion using the $CO/CO_2$ emission ratio (MOre inversion) for OCO-2 inversion over Southern Tropical South America and with IS inversion over Southern Tropical

Asia, compared to the IS inversions using GFED3 fires. For tropical Asia, the constrain of priors seems to be more important than the data assimilated. Additionally, over this region, seasonality from $CO_2$ inversions using MOPITT fires seems to be better representative of the large Indonesian fires that occurred during the 2015 El Niño event.

TCCON evaluation suggested that the prior using the FIREMo ($CO_2$ fire prior emissions computing using $CO/CO_2$ emission ratio) gives accuracy in $CO_2$ mixing ratio comparable to GFED4 but with slightly larger biases over the Northern Hemisphere

and biases of the priors with the re-balanced respiration are smaller than the CMS prior. However, biases for the posterior simulated mixing ratio are in the same order. Evaluation mainly showed that biases have been decreased and variability matches better those of TCCON for the re-balanced posterior simulated mixing ratio suggesting the importance of the accuracy in fire priors and the re-balanced of terrestrial emission with fires for $CO_2$ posteriors emissions.

We illustrated the potential of using $CO/CO_2$ emission ratio, and the re-balanced respiration and NEE with fire and growth

rate, in $CO_2$ inversion for better constraint and accuracy in the $CO_2$ fire prior emissions and biospheric emissions estimates. We found that a priori $CO_2$ flux uncertainties are substantially reduced when matching the NOAA AGR as well as $CO/CO_2$ ratio but not strong enough compared to a re-balanced GFED and GFED4.1s NEE, and suggest hence for future work the development of joint $CO-CO_2$ inversions with multi-observations for stronger constraint in posterior $CO_2$ fire and biospheric emissions. Besides, the multi-species approach employing CO and $CO_2$ for instance is important for the interpretation of

upcoming satellite data such as data from the future NASA Earth Venture Mission, GeoCarb.





**Figure A1.** Time series of CO fire emissions over the OCO-2 MIP region between fire priors (dot lines) and posteriors (solid lines) emissions. Emissions are in TgCO/yr.

## 6 Appendix



**Figure A2.** Annual CO emissions for posterior and prior (hatched bars) over all OCO-2 MIP regions. Emissions are in TgCO/yr.



**Figure A3.** Prior fire flux (top row) for 2015 over Northern (left) and Southern (right) Tropical Asia of GFED4re, GFED3re and MOre by vegetation types. The percentage of vegetation over these two regions is also represented in the bottom row. The colors represent the vegetation type : agriculture (gray), peat (red), savanna (brown), deforestation (yellow), and temperate forest (blue).



**Figure A4.** Prior fire flux from 2015 through 2018 over Northern (top) and Southern (bottom) Tropical Asia of GFED4re, GFED3re and MOre by vegetation types. The colors represent the vegetation type : agriculture (gray), peat(red), savanna (brown), deforestation (yellow), and temperate forest (blue).







**Figure A5.** Same as Fig. 7 but for all OCO-2 MIP regions.



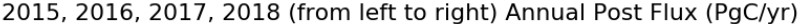

**Figure A6.** Same as Fig. 10 but for all OCO-2 MIP regions (from left to right, top to bottom) : Boreal North America, Temperate North America, North Tropical South America, South Tropical South America, Temperate South America, Temperate North Africa, North Tropical Africa, South Tropical Africa, Temperate South Africa, Boreal Eurasia, Temperate Eurasia, North Tropical Asia, South Tropical Asia, Tropical Australia, Temperate Australia, and Europe.




*Author contributions.* H.Peiro generated the CO products, MOPITT CO$_2$ fires and re-balanced priors, produced the figures and wrote the manuscript. S.Crowell generated the CO$_2$ products, provided comments and feedback on the manuscript. B.Moore provided feedback on the manuscript as well.

*Competing interests.* The authors declare that they have no conflict of interest.

*Acknowledgements.* We are thankful to Debra Wunch who helped us reviewing the TCCON references and acknowledgements, and to both Debra Wunch and Coleen Roehl who contacted the TCCON PIs. The TCCON data were obtained from the TCCON Data Archive hosted by CaltechDATA at https://tccondata.org. We thanks TCCON PIs for the TCCON measurements at Eureka, Ny-Ålesund, Sodankylä, Białystok, Bremen, Karlsruhe, Paris, Orléans, Garmisch, Park Falls, Rikubetsu, Lamont, Anmeyondo, Tsukuba, Edwards, Caltech, Saga,
Izaña, Ascension Island, Darwin, Réunion Island, Wollongong, Lauder. Eureka measurements are made by the Canadian Network for the Detection of Atmospheric Change (CANDAC) and in part by the Canadian Arctic ACE Validation Campaigns. They are supported by the Atlantic Innovation Fund/Nova Scotia Research Innovation Trust, Canada Foundation for Innovation, Canadian Foundation for Climate and Atmospheric Sciences, Canadian Space Agency, Environment Canada, Government of Canada International Polar Year funding, Natural Sciences and Engineering Research Council, Northern Scientific Training Program, Ontario Innovation Trust, Ontario Research Fund and
Polar Continental Shelf Program. Observations for Białystok are funded byt the European Union (EU) projects InGOS and ICOS-INWIRE, and bu the Senate of Bremen. Local support for Bremen and Ny-Ålesund are provided by the EU projects InGOS and ICOS-INWIRE (26188, 36677, 284274, 313169 and 640276), and by the Senate of Bremen. Orléans observations are supported by the EU projects InGOS and ICOS-INWIRE, by the Senate of Bremen and by the RAMCES team at LSCE. The Réunion Island TCCON site is operated by the Royal Belgian Institute for Space Aeronomy with financial support since 2014 by the EU project ICOS-Inwire and the ministerial decree for ICOS
(FR/35/IC1 to FR/35/IC5) and local activities supported by LACy/UMR8105 – Université de La Réunion. The Paris TCCON site has received funding from Sorbonne Université, the French research center CNRS, the French space agency CNES, and Région Île-de-France. Garmisch funding was provided by the EC within the INGOS project. Park Falls, Lamont, Edwards and Caltech TCCON site have received funding from National Aeronautics and Space Administration (NASA) grants NNX14AI60G, NNX11AG01G, NAG5-12247, NNG05-GD07G, and NASA Orbiting Carbon Observatory Program. They are supported in part by the OCO-2 project. The TCCON station at Rikubetsu and
Tsukuba are supported in part by the GOSAT series project. Darwin and Wollongong TCCON stations are funded by NASA grants NAG5-12247 and NNG05-GD07G and supported by the Australian Research Council (ARC) grants DP140101552, DP110103118, DP0879468 and LP0562346. Lauder TCCON site has received funding from National Institute of Water and Atmospheric (NIWA) Research through New Zealand's Ministry of Business, Innovation and Employment. We also acknowledge the ObsPack data used for our IS inversions.



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
