# Peer review of "Optimizing Four Years of CO2 Biospheric Fluxes from OCO-2 and in situ data in TM5: Fire Emissions from GFED and Inferred from MOPITT CO data"

_Atmospheric Chemistry and Physics, 2022_

## Author Response (AR1)

**Reply to Reviewer #1**

The authors designed a two-step inversion approach to study the impact of fires on the inversion estimation of net carbon sources and sinks. In the first step, CO fire emissions are constrained with MOPITT CO data, and these optimized emissions are translated into CO2 emissions using biome specific emission factors. Subsequently, these optimized CO2 fire emissions are used as input in a second inversion step - alongside a rebalanced prior NEE that fits the global atmospheric growth rate - to constrain regional NEE with OCO-2 and in situ CO2 data. Despite much information is provided and carefully investigated, I do have a number of concerns about the results presented. The main text also requires additional editing and a thorough proofread to improve the general readability. In particular, I would like to see a shortened results section with a focus on the main findings and a smaller number of figures. The lengthy descriptions in each section distracts from the main points of this original work. Perhaps some of detailed descriptions, figures and comparisons can be moved to a supplementary document. The points below must be addressed before the paper can be accepted for publication in ACP.

We thanks Reviewer #1 for taking the time to review our manuscript and for the comments. We answered below the comments with information on page and line numbers that have been changed in the manuscript when necessary. Red is for suppression or modification, while blue is for sentences added in the manuscript.

**Main concerns:**

**1. Reported optimized fire CO emissions:**

   1. After reading the first part of the paper, a number of things struck me regarding the CO emissions. Figure 5 provides an overview of the prior (GFED4s) and posterior CO emissions. I find it curious why the Northern Tropical and Southern Tropical African fires are scaled down so much. This contradicts other recent studies of African fires. Zheng et al. (2018), which you have included in the reference list, did a similar kind of inversion for the African continent with MOPITT CO data. For the period 2005-2016, they found more or less similar emissions for Northern Africa compared to GFED4s. But for southern Africa, they found that GFED4s underestimated the emissions by about 62%. Your results show the opposite; GFED4s overestimate emissions for Africa. Of course, there are differences in the time period studied between the two papers, but nevertheless it looks surprising. I think that this difference in outcome should be noted in the paper and, if possible, also explained.

In our manuscript, we indeed found that MOPITT CO data assimilated in TM5 has lower annual emissions for tropical Africa than observed with the fire prior. But posterior emissions are superior to the priors for the temperate regions of Africa. We only assimilated MOPITT CO v8 TIR-NIR data in our CO inversion using the model TM5. Our period of study is from 2015 through 2018, corresponding to 4 years of analysis. In Zheng et al., (2018) paper, they indeed looked at a different period: 2005 through 2016 corresponding to 11 years of analysis. Additionally, they did not assimilate the same data. They constrained their inversions with MOPITT CO v7 data as well as OMI CH2O (Ozone Monitoring Instrument, formaldehyde) and in-situ measurements from WDCGG (World Data Center for Greenhouse Gases) of CH4 and MCF (methyl chloroform) data. By adding these in-situ measurements in the inversions, they particularly help constrain OH, but also the photochemical reaction (source) of CO. Their chemistry is then different to the one used in TM5. They consequently have a multi-species atmospheric Bayesian inversion approach to get a well constrained global CO budget. This CO estimates has been shown to be higher than other inversions. In Zheng et al., 2019, they compared their CO biomass burning emissions with their multi-species estimation to nine previous studies using different inversion systems (Fig.10 of Zheng et al., (2019)). The use of multi-species inversion increases the annual average emissions of CO biomass burning at global scale by 20-37% compared to other inversions. Even if we used an updated version of MOPITT data compared to this study, we did not assimilate in-situ data and our chemistry is not as well performed as they used. As a reminder, the chemistry used in TM5 is based on climatology emissions (CO production based on a full TM4 chemistry of 2006). The model, transport, as well as assimilation algorithm used in Zheng et al., (2018, 2019) are also different to those used in TM5. It is then difficult to evaluate our results to their results.

Additionally, the figure below gives confidence that our posterior CO emissions fit the MOPITT data over Africa.

Finally, it is important to mention that Zheng et al., (2018, 2019), do not provide any evaluation against independent data. Therefore, there is no result proving that their results are more accurate or more trustworthy than our results.

2. Is this African signal driven purely by MOPITT CO? If so, can you demonstrate this by showing world maps with annual mean concentrations of prior CO (from GFED4s), posterior CO (from optimized fires) and MOPITT CO?

We thank the reviewer for this comment. We evaluated below the inversion's framework to the MOPITT measurement. We found, indeed, that the lower signal in our posterior is driven by low MOPITT concentrations over Africa. The figure below has been added to the supplementary information consequently and reference to this figure has been added in the section 3.1.

[Figure]

**Figure S1b.** Spatial distributions of the CO total column (XCO). Left column : distribution of annual mean XCO of MOPITTv8 retrieval. Center column : Mean annual difference between the prior simulation and MOPITT. Right column : Mean annual difference between the posterior simulation and MOPITT. From top to bottom are the annual mean from 2015 through 2018. Results are in ppb.

Figure S1b shows the spatial distribution of the annual mean XCO observed with MOPITTv8. We can observe a latitudinal gradient from north to south with high values over East Asia and central Africa. For the prior, we can see that on average it is higher than the observations, particularly over the tropics

(such as Indonesia) but also for northern America such as Alaska in 2015 or Africa. Differences between the prior and the observations over these regions are greater than 34 ppb. The optimized CO concentrations have on average lower difference with the observations, but the concentrations are between 16-20 ppb lower on average across the globe than the MOPITT measurements, except for some regions of Africa, South America in 2015 and East Asia for all years where the concentrations are higher than the MOPITT data. We can particularly observe optimized CO concentrations of 32 ppb greater than the measurements in 2015 over Indonesia and southeast Asia. Looking particularly at Africa and for both tropical south and north Africa, priors XCO are higher than the MOPITT measurements by 32 ppb. The low emissions observed with the posterior compared to the prior could come from this difference between the prior mixing ratio and the measurement and/or the chemistry in TM5 (different from other studies, such as Zheng et al., (2018) which have a multi-species inversion for a more observationally constrained chemistry).

3. And how does the seasonal cycle of these fires look like for Northern and Southern Africa? This can be shown with a figure similar to Figure A4.

The seasonal cycle of Africa is already available in Fig. A1 (now Fig. S6 in supplement information) for the regions: Temperate north Africa, northern tropical Africa, southern tropical Africa, and temperate south Africa. We can see that both prior and posterior emissions are similar for temperate north Africa (with emissions less than 0.12 TgCO/yr), probably caused by the large biases of Sahara dust. The seasonal cycle of the prior and posterior are similar for all regions. For temperate south Africa, we can see higher posterior emissions compared to the prior. For northern tropical Africa, our posterior CO emissions are lower than with the prior for the whole period, even during peaks of emissions, corresponding to annual emissions 25% lower with the posterior than with the prior. While for southern tropical Africa, the maximum peak of emissions is closer each other between the posterior and the prior, particularly in 2016 and 2018, and with a 50-70 TgCO/yr of difference in July-August of 2015 and 2017. The posterior CO emissions are 16% lower than the prior for this region.

4. And finally, is there independent data that support your results? The 'lack' of fire emissions in Africa could explain the appearance of the large compensating natural CO2 source in Northern Tropical Africa in your OCO-2 inversions (Figure 11).

We have added, in supplementary information, an evaluation of the CO inversion against TCCON data (Fig. S10, see response to Reviewer #2).

In the evaluation against TCCON, we can observe an underestimation of the posterior CO mixing ratio of ~ -12 ppb in 2015 at the Ascension Island site. However, the a priori CO mixing ratio has an overestimation of 5 ppb in 2015. A similar pattern is found for Reunion Island, with an underestimation of about -7 ppb with the posterior and an overestimation of about 2 ppb with the prior. However, the biases at the Darwin TCCON site give -3 ppb for 2015-2016 (-0.5 ppb for 2017-2018) with the posterior and 20 ppb for 2015-2016 (22 ppb for 2017-2018) with the prior. This gives the impression that our inversion is not getting the best fluxes for Ascension Island, but we can see that this is not the case for other tropical locations. Ascension Island is known to be impacted with Saharan dust and therefore the posterior simulated concentration could be biased due to aerosols.

This underestimation of CO fire with the posterior could explain the larger (smaller) NEE source for MOre emissions compared to other OCO-2 simulations in tropical north and south Africa. For IS inversions, the compensating emissions give a deeper net sink with ISMOre for tropical north Africa compared to other IS simulations. It gives for southern tropical Africa higher net sources compared to IS3re and ISCMS, but lower net sources to IS4re. Now that we have a site-by-site evaluation with TCCON (see Reviewer #2 and Figure 10 in the updated manuscript), we can see that for Ascension Island, ISMOre has the lowest biases with TCCON (-0.55ppm) compared to other posteriors. GFED4re and MOre additionally have the lowest biases with TCCON (-0.80 and -0.83ppm respectively) compared to GFED3re and CMS-GFED3. Even if we cannot conclude which inversion is doing better than the other with the evaluation, it seems that, at least for Ascension Island, ISMOre does better than the other inversions.

5. In addition, I would like to inform the authors of a recent PNAS paper by Ramo et al. (2021) which shows with new 20-m burned area data from Sentinel-2 that the GFED4s emissions for Africa are probably greatly underestimated.

Ramo et al. (2021), https://doi.org/10.1073/pnas.2011160118

We thank the Reviewer for this information. It would be interesting for future work to include these fire emission estimates as a prior and compared the resulting CO emissions with the ones estimated using GFED4.1s. We included this information in the discussion page 36, line 714: A recent study has shown the underestimation for Africa of MODIS burned area and consequently GFED4s, compared to the new Sentinel-2 burned area product (Ramo et al., (2021)). The higher fire posterior emissions observed with previous studies using GFAS as a prior compared to GFED4 (Nechita-Banda et al., (2018)) and the results of Ramo et al., (2021) seems to suggest for future work to carefully choose the CO fire prior used in a CO-CO$_2$ study. Future work will be done comparing different CO posterior emissions.

6. A similar contradiction arises in Indonesia. In Figure 5a, strangely enough, lower posterior emissions are reported for the different fire types during the intense El Nino fires of 2015. This is the opposite of what Nechita-Banda et al. (2018) found in their MOPITT CO inversion study. They found emissions up to 120 Tg CO, about 1.5 times greater than what GFED4s provided. This difference should also be addressed and explained in your paper.

The reviewer compared our MOPITT CO inversion against the inversions done in the paper of Nechita-Banda et al., 2018 mentioning that our results corresponding to the 2015 annual emissions are different to the estimation of Nechita-Banda et al., 2018. However, it is important to remark that the temporal scale, spatial scale, data assimilated and prior used are different between the two studies. First, while our study looks at annual emissions, Nechita-Banda et al., 2018 looks for the brief period of mid-August to mid-November 2015. Their study does not include the whole year 2015. Second, the regions of interest over Southeast Asia are different between our two studies. Their studies focused on the Indonesia and Central tropical Asia, while our study has two regions that divide Indonesia and Central Asia (please refer to Fig. 4 of the pre-print, now Fig. 3 of our manuscript). Finally, the data used and assimilated in both studies are not the same. In our paper, for the CO inversion, we only assimilated MOPITT CO TIR-NIR version 8 data. However, in the study of Nechita-Banda et al., 2018, they used and assimilated both MOPITT CO NIR-TIR version 7 data and the ground-based NOAA data. We can read in their paper, page 4, section (d): *"The main results described in this paper are from the 'IASI' and 'MOPITT' simulations, where IASI and, respectively, MOPITT level 2 data are used together with NOAA surface stations, to optimize CO emissions"*. NOAA ground-based data have been shown to have higher CO biomass burning emissions compared to the prior when assimilated at global scale (Hooghiemstra et al., 2012). The use of these ground base data in Nechita-Banda et al., 2018, together with MOPITT data could explain the higher emissions than with MOPITT alone. Additionally, it is

important to notify that our posterior MOPITT emissions are done using the prior GFED4.1s, while in Nechita-Banda et al., 2018 they used and mentioned "*Most simulations presented use GFAS v.1.3 as prior biomass burning emissions, except for 'IASI OH GFED', which uses GFED4s as prior*". Additionally, they have adjusted the GFED peat emissions with a higher CO/NOx ratio than typically used in GFED. But, in their Fig. 4, we can see for the total Indonesia region that the posterior emissions IASI using GFAS (called IASI OH) has higher emissions than the posterior emissions using GFED as prior (called IASI OH GFED). Additionally, in their Fig.5, we can see, for their period of study, higher CO fire emissions with prior GFAS than with prior GFED. Their posterior CO fire emissions (opt MOPITT and opt IASI) have similar seasonality than prior GFAS and consequently give higher emissions than the prior GFED. In their study, they do not present results of posterior emissions using only MOPITT data with GFED prior. Using both MOPITT-NOAA data and the prior GFAS in their inversions, could have given higher emissions than using a MOPITT-GFED inversion. We cannot, consequently, compared our CO fire estimate with Nechita-Banda et al., 2018.

In their paper, we can find a comparison with two other studies, Yin et al., 2016 and Huijnen et al., 2015, which both studies focused on similar region and assimilated MOPITT data using the prior GFAS in their inversion. In Huijnen et al., 2015 paper, we can see in Fig.4 that their posterior emissions are lower for the whole period of study than their prior (GFAS), contradiction to what Nechita-Banda et al., 2018 found. When Nechita-Banda et al., 2018 extrapolated the results of Huijnen et al, 2015 to the same period they used, they found 96 TgCO (17 TgCO lower).

Finally, it is important to mention that no evaluation against independent data are performed in their study. Therefore, there is no result proving that their results are more accurate or more trustworthy than our results.

7. Besides that, it also contradicts with what is shown in the lowest panel of Figure 9 of your paper. The blue (GFED4re) and red (MOoptre) bars are (almost) identical for 2015 (and the other 3 years). And finally, this result contradicts with the statement at line 694: "the fire estimated from MOPITT CO emissions are stronger than with GFED4 emissions". Again, according to Figure 5a, MOPITT CO emissions are actually smaller.

In Figure 5 (now Figure 4), we represented the biomass burning of MOPITT CO posterior and GFED4.1s prior emissions partitioned by vegetation types. The partition of the vegetation types is done

using the fraction of dry matter available in the GFED product. From the MOPITT CO Posterior emissions, we calculated and estimated the average CO2 fire emissions (corresponding to FIREMo in our study). However, in some CO2 inversion, biofuel is included with the biomass burning emissions. Biofuel refers here as the CASA-GFED3 (Ott, 2020) product of the anthropogenic burning of harvested wood (van der Werf et al., 2010). This product is calculated as the population density times national per capita fuel consumption estimates while being constrained by the total available coarse woody debris at each model time step. Consequently, we summed each fire and biofuel emissions to get FIRE3, FIRE4 and FIREMO used in our CO2 inversions. We have already included a sentence in the manuscript "*All CO2 FIRE priors include both biomass and biofuel burning*" referring to this inclusion. But since this seems to not be informative, we also added this information in Table 3, with corresponding FIRE emissions. It is important then to remember that biofuel emissions are used in our CO2 results for the FIRE emissions.

Regarding line 694, our sentence: "*the fire estimated from MOPITT CO emissions are stronger than with GFED4 emissions but lower than GFED3 emissions*" was referencing to CO2 fire estimated of Figures 9 and A3 of the pre-print, the only section where we compared all together fire estimates from MOPITT, GFED4 and GFED3. We are sorry for the confusion, we should have mentioned this sentence by "In 2015, during the onset of the El Nino event which caused intense fires over Indonesia, FIREMo are stronger than with FIRE4 emissions but lower than FIRE3 emissions." In the comments below, we reply and calculate the conversion from CO to CO2 for FIREMo, which is consistent with our results. We remind that we did only the conversion for the FIREMo. We did not convert our CO prior emissions into CO2 fire emissions but used the GFED4.1s and GFED3 products available from the CASA-GFED products.

8. Given these two major inconsistencies, I wonder how much we can trust the results of the fire inversions for the other regions. For example, the strong reduction in fire emissions in the boreal regions is also striking and is somewhat inconsistent with the record-breaking fires we have seen in recent years in these regions.

For the boreal regions, we indeed have lower posterior CO fire emissions than the prior. But this seems to come from the measurements. You can see Fig. S1b, that the MOPITT measurements do not see any signal from the boreal regions. The comparison PRIOR-MOPITT in this figure shows that the prior mixing ratio has higher concentrations than the measurements over Alaska and the boreal regions.

There is then an underestimation of the measurement compared to the prior for this region. This underestimation seems to emerge in the posterior mixing ratio and could explain the lower emissions with the posterior observed in the fire CO emissions. The evaluation with TCCON data (Figure S10) shows indeed an underestimation globally of 7ppb with the posterior mixing ratio, but the prior has a larger bias (of around 13ppb).

9. I also noticed that the breakdown of fire emissions in Figure 5 does not agree at all with the reported CO emissions by GFED4s. As an example, if I look at Southern Tropical Asia in panel 5a, the breakdown between GFED4s agricultural, deforestation, savanna, and peat fires is roughly: 5, 38, 25 and 22 Tg CO/yr, respectively. However, if I look at the published GFED4s emission tables for 2015 for the EQAS region (roughly similar to your STA region) I get a completely different set of emissions per fire type: 0.7, 31, 3.8 and 74.2 Tg CO/yr, respectively. So in words, GFED4s reports a much larger source of CO from peat fires and a smaller source of CO from savanna/grasslands than you. Therefore, please carefully check Figure 5 for possible errors for all the regions.

GFED4s tables: https://www.geo.vu.nl/~gwerf/GFED/GFED4/tables/GFED4.1s_CO.txt

To verify that vegetation partition was correctly applied to the CO prior, Figure R1 below shows the difference between the prior CO emissions for January and June 2015 on a global scale minus the sum of the emissions partitioned by vegetation.

[Figure]

**Figure R1.** Difference between the prior CO Fire emissions (GFED4.1s) versus the sum of the emissions partitioned by vegetation for January 2015 (left) and June 2015 (right). Values are in TgCO/yr (x10⁻⁷).

We can see on the figure that the differences are very small (in the order of 3x10-7 TgCO/yr) and so quasi in-existent. The vegetation partitioned shown in our Figure 5 from the preprint is then correct.

The difference between the values reported on the GFED website and our prior could be caused by the difference in spatial scales. The GFED reported on the website are reported at the MODIS resolution and then aggregated by region, while our estimates are reported at 3x2 spatial resolution.

10. Similar issues also apply to Fig. A3. The contributions from the different fire types do not agree with the reported values in Fig. 5.

We need to remind the readers that we calculated and converted the CO2 fire emissions from CO fire emissions only for our FIREMO estimates. We did not convert the emissions for the FIRE4 (for GFED4.1s) and FIRE3 (for GFED3). The FIRE4 and FIRE3 have been taken directly from the CASA-GFED emissions inventories. Additionally, like previously mentioned, this is potentially a difference in spatial scales. The GFED emissions are calculated at the resolution of MODIS pixels, while the FIREMO ones are done at 3x2 degrees resolution. Then, like already mentioned in the manuscript, all CO2 inversions are done at 6x4 degrees resolution.

We double checked if the FIREMO reported for the CO2 results are correctly calculated from the CO fire emissions, which indeed they are. So for instance, if we take the CO fire emissions of our CO posterior emissions (MOPITT) reported in Figure 5 (now Figure 4) over southern tropical Asia in 2015,

which is approximately 15 TgCO/yr for the peat lands, we need to apply the following equation (as already explained in our manuscript in section 2.3.1.b):

$$\frac{\frac{12}{28}*CO(TgCO/yr)*10^{-3}}{ER_{peat}}=CO_2(PgC/yr) \rightarrow \quad \frac{\frac{12}{28}*15*10^{-3}}{0.194}=0.033\,PgC/yr \quad \text{which corresponds}$$

approximately to our figure A3.

If we do the same with the CO values of deforestation, agriculture and savanna which are approximately 26 TgCO/yr, 3 TgCO/yr and 23 TgCO/yr, we get respectively 0.13, 0.013 and 0.17 PgC/yr. From these values, the biofuel emissions need to be added to have exactly the values shown in Figure A3 (now Figure S8). These values are practically similar to our estimate in Figure A3 (now Figure S8).

11. To better understand the reported posterior CO fire emissions it would also be helpful if the TCCON comparisons for CO are provided to the reader (in a supplementary document).

We have already answered this comment with our reply to Referee #2. We added in the supplementary document, the TCCON evaluation of our prior and posterior CO mixing ratio. We ask the reader to refer to the comments of Referee #2 for further information.

12. Finally, I would like to reiterate Meinrat Andreae's comments that the emission factors should indeed be replaced by the updated ones reported in Andreae (2019).

We answered to this comment with our reply to Meinrat Andreae's comment. We ask the reader to refer to Andreae's comment for further information.

**2. Role of fire emissions on the optimized NEE**

13. How can we benefit from this more tedious 2-step inversion approach? The answer is somewhat lost in the lengthy description of the results. If I simply look at Figures 11, 12 and 13, the differences in NEE between the different inversion experiments (with the same data constraints) are small. The largest differences originate from using either OCO-2 or in situ CO2 data for reasons discussed in the paper (like differences in data coverage). However, with either data constraint, the optimized CO2 fire emissions seem to have a small impact on the NEE in comparison to the unoptimized CO2 fire emissions.

As described in the abstract, both MOPITT derived fires and GFED4s fires provide larger net sources in the tropics.What specifically did we learn from the MOPITT experiment? Also the comparison with independent TCCON data in Table 6 does not show significant differences in comparison to the other inversions if you use the optimized fire emissions in a NEE inversion. A more in depth discussion about the wider application of this methodology is necessary and should be mentioned in the abstract.

We thank the reviewer for this comment. We indeed shortened the manuscript as advised in the following comments and by Reviewer#2. However, we added in the evaluation section a comparison between the priors and posteriors with the OCO-2 retrievals and IS measurements to see how the inversions fit the data. We also changed the evaluation with TCCON to have a site-by-site evaluation. Regarding the abstract, we changed the sentences page 1 line 20:

"Evaluation with TCCON suggests that the re-balanced posterior simulated give biases and accuracy very close each other where biases have decreased and variability matches better the validation data than with the CASA-GFED3. Further work is needed to improve prior fluxes in Tropical regions where fires are a significant component."

as following:

"Evaluation with TCCON data shows lower biases with the three re-balanced priors than with the prior using CASA-GFED3. However, posteriors have accuracies very close each other, making difficult the conclusion of which simulation is better than the other. One major conclusion from this work is the strong constrain at global scale of the data assimilated compared to the fire prior used. But results in the tropical regions suggest sensitivity to the fire prior for both the IS and OCO-2 inversions. Further work is needed to improve prior fluxes in tropical regions where fires are a significant component. Finally, even if the inversions using the FIREMO prior did enhance the biases over some TCCON sites, it is not the case for the globe. This study consequently push forward the development of a CO-CO2 joint

inversion with multi-observations for possible stronger constraint in posterior CO2 fire and biospheric emissions. "

**3. Readability of the paper**

14. The paper would benefit from some additional editing. Mainly to correct and shorten the long (and sometimes awkward) sentences and to trim down some of the lengthy descriptions. In particular, the Results section should focus more on the main results of the paper and be presented in a more concise and logical way. Now I find it difficult to quickly get the gist of the paper in the whole list of comparisons between the inversions and between the different data constraints used. The fact that it is a two-step inversion makes that even more difficult. The posterior of the first inversion becomes the prior of the second inversion, and that blurs the distinction between 'prior' and 'posterior' labels. There is so much work in this paper that I wonder if it wouldn't be better to split the paper into two separate papers: one for the estimation of fire emissions with MOPITT CO constraints, and a follow-up study on the impact on NEE with OCO-2 and in situ data, but that's something up the authors to decide. You can perhaps choose to move some the detailed comparisons to a supplementary document where the reader can find additional information about the main results.

Thank you for this remarks. We decided to not separate the paper as the comparison among the different simulations was to see any impact or not of using CO information in CO2 inversions with two different set of assimilated data. We consequently chose to move some of information in a supplementary document to make the paper easier to read. For more clarity also, we have added some diagrams in the methodology section.

15. I also noticed at lot of inconsistencies between labels reported in the caption, in the figures and elsewhere in the tables and the main text. For example, I came across multiple names for the same inversion type: CMS-GFED3 (in Table 5), CMS (e.g. in Fig. 12) and OCOcms (in Fig. 5a). Another example: MOPITTopt (e.g. line 455), MOoptre (e.g. Fig. A4), and MOre (e.g. Table 4). I find this distracting and should be fixed in the revised manuscript.

Sometimes you interchangeably use the inversion name (e.g. MOre) when you refer to a flux and vice versa. For example at line 454 you write: "The prior categories shown are fire, NEE and net fluxes for the prior GFED4, GFED3, MOPITTopt and CMS-GFED3".

If you follow your own Table 5, this should change to: "The prior categories shown are fire and NEE for GFED4re, GFED3re, MOre and CMS-GFED3".

We corrected all of the annotation in the manuscript and particularly named the priors, and the posteriors.

16. I also believe the number of figures should be reduced (currently 14 excluding the Appendix figures). You can merge a number of figures into a single figure. For example, Figure 8 and 9 should be combined into 5 squared panels. That provides an overview of all emission estimates on a single page. Similarly, Figures 11, 12 and 13 can be combined into a single page filling figure with 8 panels.

We reduced the number of figures and instead of splitting the figures by region, we only use the figure that was containing all regions. This figure was previously available in the annex.

17. Finally, shortening of the region labels with No and So in the figure titles is often not necessary, is confusing, and it doesn't look very pretty. If there is enough space write the labels in full. For example, So Trop So America should become Southern Tropical South America.

We corrected the labels for all figures.

18. In the specific comments below I give more examples of long sentences, wrong usage of inversion names, and other specific errors.

We thank the reviewer for these specific comments.

**Line 6.** This part is not clear. In the end are the CO2 NEE and ocean fluxes optimized with OCO-2 and insitu data or only NEE? This should be stated very clearly in the abstract.

Suggestion:

These optimized CO2 fire emissions (FIREMo) are used to re-balance the Net Ecosystem Exchange (NEEmo) and respiration (Rmo) with the global CO2 growth rate. Subsequently, in a second step, these rebalanced fluxes are used as priors for an inversion to derive the NEE and ocean fluxes constrained either by the Orbiting Carbon Observatory 2 (OCO-2) v9 or by in situ CO2 data.

The NEE and ocean fluxes are optimized with OCO-2 and IS. This was mentioned page 15, line 357. We took the suggestion in consideration and changed the sentence with: These optimized CO2 fire emissions (FIREMo) are used to re-balance the CO2 Net Ecosystem Exchange (NEEmo) and respiration (Rmo) with the global CO2 growth rate. Subsequently, in a second step, these rebalanced fluxes are used as priors for a CO2 inversion to derive the NEE and ocean fluxes constrained either by the Orbiting Carbon Observatory 2 (OCO-2) v9 or by in situ CO2 data.

**Line 11.** Be consistent throughout the paper with labels. Use either CASA-GFED3 or GFED3. We corrected this through the all paper.

**Line 12.** "Results show…" Unclear what this sentence is trying to say. Does "Results" refer here to the evaluation with TCCON? Or does it refer to the flux estimates?

The way I read it is that the posterior flux estimates (whether you mean NEE or fire is also unclear) are more robust (i.e. similar) than the different prior flux estimates. For clarity, we changed the sentence to: Comparison of the flux estimates show that at global scale posterior net flux estimates are more robust than the different prior flux estimates.  However, at regional scale, we can observe differences in fire emissions among the priors, resulting in large adjustments in the Net Ecosystem Exchange (NEE) to match the fires and observations.

**Lines 16-20.** I find this short recap of the main results quite hard to read because so many geographical regions and elements of 2-step inversion approach are compressed in 2 sentences (GFED4s, MOPITT CO, OCO-2, insitu data). Please rewrite.

A suggestion:

Slightly larger net CO2 sources are derived with posterior fire emissions in the OCO-2 inversion, in particular for most Tropical regions during 2015 El Nino year. Similarly, larger net CO2 sources are also derived with posterior fire emissions in the in-situ data inversion for Tropical Asia.

We rewrite this lines with: Slightly larger net CO2 sources are derived with posterior fire emissions using either FIRE4 or FIREMo in the OCO-2 inversion, in particular for most Tropical regions during 2015 El Nino year. Similarly, larger net CO2 sources are also derived with posterior fire emissions in the in-situ data inversion for Tropical Asia.

**Line 21.** Use either: 're-balanced posterior simulation' or 're-balanced posterior simulated concentrations' . We changed this sentence regarding a comment from Reviewer #2: Evaluation with TCCON data shows lower biases with the three re-balanced priors than with the priorCMS. However, posteriors simulated concentrations give biases and accuracy very close to each other, making difficult a conclusion of which simulation is better than the other.

**Line 21.** 'very close **to** each other' Done

**Line 34.** Be aware that since 2017 GFED4s emissions are not based anymore on direct burned area datasets, but instead based on relationships between MODIS active fire detections and GFED4s emissions for the period 2003-2016. This is because the underlying burned area dataset has been upgraded in the meantime from Collection 5.1 to Collection 6, making it incompatible for usage in GFED4s. GFED4s emissions from 2017 onward are therefore called GFED4s-beta emissions.

See: https://www.geo.vu.nl/~gwerf/GFED/GFED4/Readme.pdf

We modified the sentence "The first uses total fuel consumption per product of the burned area and the fuel consumption per unit area deduced from the burned area and active fires products of the Moderate Resolution Imaging Spectroradiometer (MODIS)." by:

"The first uses, since 2017, total fuel consumption per product of the burned area and the fuel consumption per unit area deduced from the burned area and active fires products of the Moderate Resolution Imaging Spectroradiometer (MODIS). Previous years were based directly on burned area datasets."

**Line 51.** Induce Done

**Line 81.** There are many examples you write double plural "emissions sources", "emissions estimates", "emissions inventories". Although I'm not an English native speaker, I think it's grammatically better if you write it as "emission sources" or "emission estimates". Corrected

**Line 98.** "and the post-event" I suggest "and the subsequent years" Changed

**Line 106.** "The importance of these results for conclusions …". I think this is grammatically incorrect. Please rewrite. Something like: "The importance of these inversion results are discussed in Section 4." Sentence changed to: The importance of these inversion results are discussed in Section 4.

**Line 110.** Please provide a clearer breakdown of the 2-step approach. A suggestion:

Our inversions are performed in sequence: (1) we assimilate total column CO retrievals from the MOPITT v8 products to produce optimized CO fluxes, which are used to update the assumed $CO_2$ fire emissions, and then (2) we assimilate either total column $CO_2$ from OCO-2 version 9 retrievals or $CO_2$ in situ data to produce optimized $CO_2$ NEE fluxes.

We considered this suggestion and wrote: Our inversions are performed in sequence: (1) we assimilate total column CO retrievals from the MOPITT v8 products to produce optimized CO fluxes, which are used to update the assumed $CO_2$ fire emissions, and then (2) we assimilate either total column $CO_2$ from OCO-2 version 9 retrievals or $CO_2$ in situ data to produce optimized $CO_2$ NEE and ocean fluxes.

**Line 113.** validation (singular) Corrected

**Line 124.** "…allowing a well-understood of its continuity and consistency ". Please rewrite. We modified the sentence with: However, MOPITT products have been consistently validated against airborne vertical profiles and ground based measurements, allowing a well-understood product

**Line 177.** Awkward sentence. Perhaps start it like this: "Despite the known shortcomings (biases) of satellite data, several studies have preferred to use satellite data over the Tropics to take full advantage of the improved spatial coverage." Changed

**Line 182.** at smaller spatial and temporal scales  Changed

**Line 214.** suggestion: "…shows the site locations over the globe." Changed

**Line 226.** Change to "the global in situ network" Added

**Line 233.** …the corresponding satellite and in situ data. Added

**Line 236.** R covariance structure is not discussed in the paper.  We have added the sentence page 12 line 290: The errors are assumed uncorrelated leading to a diagonal observational error covariance matrix **R**.

**Line 262-264.** These 2 sentences should be merged to the same paragraph. One of the reviewer's comments concerned the length of the manuscript and recommended reducing it. To shorten the manuscript, we have therefore moved lines 264 to 277 to the appendix.

**Line 272.** Change to "fire carbon emissions with 11%" Changed

**Line 293-296.** This part needs some editing. Also, I'm not sure what you trying to say here. Do you mean perhaps if an optimized CO flux for a pixel becomes twice as large (after inversion with MOPITT CO) you scale up the fluxes of the underlying vegetation types with a factor of 2? This section needs indeed some editing and clarification. For this purpose, we added in the manuscript a flowchart (Fig. 1) about the CO inversion and the vegetation partitioning used for the FIREMo calculation.

[Figure]

Figure 1. Flowchart of the FIREMo calculation

We also changed the paragraph to:

In this section, we describe the computation of our optimized prior fire emission (FIREMo) which we will use to observe the impact of CO fire emissions in posterior CO2 biospheric fluxes. The steps of the FIREMo calculation are shown in Fig. 1. For each pixel of CO posterior fire emissions, we applied a vegetation fraction based on the dry matter product (DM) of GFED4.1s. We obtained fire emissions for each monthly vegetation type (savanna, boreal forests, peat, temperate forests, deforestation and agriculture waste). Figure S1 shows GFED DM vegetation type for each year over land, where each pixel represents one or more vegetation types.

**Line 296-297.** Suggestion: "Figure 3 shows for instance the GFED vegetation type for each year, where each pixel represents one or several vegetation types." We modified the sentence by: Figure S1 shows GFED DM vegetation type for each year over land, where each pixel represents one or more vegetation types.

**Line 299.** As mentioned Andreae. Use the new updated EFs published in Andreae (2019). We already replied to this comment in Andreae's comment. The CASA-GFED3 product does not use the new updated EF published in Andreae (2019) and consequently, for consistency among the fire product used, we did not use the new EF product in our FIREMo product.

**Line 309.** "type per grid box". This is at 3x2 resolution? If so, state that explicitly. Changed

**Line 311.** "in balance with fire estimate". You mean in balance with the atmospheric CO2 trend, or not? We changed the sentence by: We used this FIREMo as a fire prior emissions in CO2 inversions along with a re-balanced respiration and NEE (in balance with fire estimate), using the parameterization described in the following section 2.3.1.c.

**Line 318.** Unclear. Are the ocean fluxes also optimized or not? We modified the sentence by: Ocean fluxes are taken from Takahashi et al. (2009). They are assumed to have an uncertainty variance of 50%. Both biospheric and oceanic emissions are optimized in the CO2 inversions.

**Line 323.** You mean CASA-GFED3 We meant GEOS-Carb CASA-GFED3 project.

**Line 326.** "gross ecosystem exchange". Is this the correct terminology? Should it not be the Net Primary Production (NPP)? NPP is equal to the sum of gross primary production (GPP) and autotrophic (maintenance) respiration (Ra). See below.

NEE = GPP+Rh+Ra=Rh+NPP

NPP = GPP+Ra

So, we can read in our paper that the net ecosystem exchange (NEE) is expressed as the sum of heterotrophic respiration (Rh) and gross ecosystem exchange (GEE) : NEE= Rh + GEE

Which corresponds with what you wrote : NEE= GPP + Ra + Rh = GEE + Rh. Our equation is right, and the Rh is the respiration we have balanced with the fire, approach used in GEOS-Carb CASA-GFED3 (Ott , 2020).

eq. 2. What is the meaning of number 3? Because it is based on CASA-GFED3? Yes it is referencing to CASA-GFED3. The text above the equation already mentioned the meaning of elements used in eq.2. We modified Geos-Carb into CASA-GFED3.

**Line 328.** What is the meaning of MOPITTOpt and RMo? First time these parameters are mentioned. We corrected that with FIREMo and RhMo (respiration linked to FIREMo)

**Table 3:** Something is off with the calculated values of FIRE4 (GFED4s). When I calculate the emissions myself from the official GFED4s tables I calculate for 2017, 2018 and 2019 very different emissions than what is reported in Table 3 (highlighted here in bold). I take the global emissions from https://www.geo.vu.nl/~gwerf/GFED/GFED4/tables/GFED4.1s_CO2.txt and subsequently convert them to carbon emissions with 12/44 ratio.

FIRE4 1.88 2.09 1.73 **1.78 1.69 2.13**

We corrected the values as indeed the values were not correctly written. We corrected both GFED4 and FIRE4.

It would also be helpful to see the estimates of NEEre3, NEEre4 and NEEreMO, and CMS-Carb in Table 3.

We added the estimates in Table 3. We also decided to remove 2014 and 2019 values are the full years were not used in the inversion (spin up period).

There is also another issue with FIRE4 and FIRE3 estimates in Table 3. They do not match with the blue and green bars in Figure 7. All estimates in Fig. 7a exceed 2 PgC/yr, which is not the case in Table 3.

Indeed, as we mentioned page 15 line 351 "All CO2 FIRE priors include both biomass and biofuel burning. " The biofuel of table 3 is additionned to the fire emissions to get the FIRE emissions used in the inversions. When we addition the fire and biofuels emissions of table 3, we find the values shown in Figure 7. We changed the table3 by adding GFED3, GFED4, FIRE3 and FIRE4 values.

Line 380-423. Very difficult to read. A lot of inconsistencies in region labels, and redundant information in the text. I suggest to keep it simple and concise by briefly discussing the results in separate short paragraphs for each continent. Just keep it simple. Highlight that the optimized fires are generally smaller than prior fires.

This section also omits important large differences between GFED4s and the posterior fires (see my earlier concerns). Especially the differences for the boreal regions, Africa, and Indonesia are striking and inconsistent with the recent literature. This should be addressed.

As discussed in the comment earlier, the studies of Zheng et al., (2018) and Nechita-Banda et al., (2019) did not use the same data in their inversions or assimilated additional data. Zheng et al., (2019) shows that their inversions set-up gives higher emissions than other inversions. Moreover, no evaluation is available to determine which results are the most reliable. We just added a few sentences in our manuscript about the differences between our study and their results. We rewrote the paragraph as:

[revised manuscript text omitted]

Typo in legend: moppit should be MOPITT

In the legend make MOPITT and GFED4s transparant white, not grey (as this color is already used for agri).

I suggest to include a bar graph in each panel that shows the global emissions for the different land types.

We modified the figure, following comments from Reviewer #2, with only a bar plot figures. We modified the legend as well as suggesting here. The figure can be find with the comments of Reviewer #2. A figure with the global emissions for each regions was already present in the manuscript in Annex. We moved this figure in supplementary information (Fig. S7).

**Line 428.** You mean "The first one" Changed

**Line 438.** temperate without capital Changed

**Figure 7:** Please put the experiment labels in the middle of each grouping of bars. In addition, label the 4 panels with a, b, c and d. Similar comments apply to the other bar plots. We changed the label in all bar plots. However, regarding the space available in each panel. We did not modified the legend.

**Section 3.2.1.** Overall I believe this section can be trimmed down and made more concise. Done

**Figure 8 and 9** should be combined into 5 squared panels. Done

**Line 474.** "global fires emissions" => "global fire emissions" Done

"We can also observed" => "We can also observe" Done

**Line 477.** "FIREMo observes less emissions" => e.g. "FIREMo yields less emissions" Done

**Line 485.** Example of a somewhat tedious number of sentences that should be condensed to one sentence.

"The larger emissions with FIREMo compared to FIRE4 over tropical Asia comes mainly from some specific vegetation. The main vegetation type in this region is savanna and we can observe that for the $CO_2$ prior emissions, FIREMo has the higher flux for Northern tropical Asia (Southern tropical Asia) compared to FIRE3 and FIRE4 (FIRE4 respectively) for savanna but also for agriculture and deforestation (see Fig. A3)."

Besides the sentence structure, I disagree with what it says. Figure 9c shows FIREmo emissions are of similar magnitude as FIRE3 and FIRE4. We modified the paragraph. However, Fig A3 shows higher savanna emissions with FIREMO than with FIRE4. We then kept this sentence and moved Fig. A3 in supplementary information:

The larger emissions with FIREMo compared to FIRE4 over tropical Asia comes mainly from savanna (the main vegetation type in this region, see Fig. S8).

**Line 493.** capture => captured Done

**Line 495 and several other lines.** "the GFED" => "GFED" Done

**Line 498.** Smokes => smoke  Done

**Line 510.** "…between OCO-2 and IS inversions the larger ensemble…" => "…between OCO-2 and IS inversions detailed in Crowell et al. (2019) and Peiro et al. (2022)"  Done

**Line 517.** "…for the Tropics with larger sources for the OCO-2 inversions…". Is this not simply because FIREmo is much smaller in tropics than FIRE4, and thus we see a compensating effect by increasing the NEE source? This sentence does not refer to a specific simulation but compares the OCO-2 inversions with the IS inversions. The sentence was specifically "The net sinks observed with the in situ inversions are weaker than OCO-2 for 2016, 2017, and 2018, and the year-to-year variations are significantly larger than the OCO-2 results. Similar behavior is observed for the SH Ext and opposite behavior for the Tropics with larger sources for the OCO-2 inversions, which could be related to cross-talk between the zonal bands given the sparse coverage of in situ data in the Tropics. " Fires are the same between the IS and OCO-2 inversions. So the difference in sink or source between the IS and OCO-2 inversions are mainly driven by the assimilated data. As we mentioned, the differences in behavior could be related to the sparse coverage of in situ data in the tropics.

**Line 517.** "…opposite behavior for the Tropics" This part should be the beginning of a new sentence. We changed by: Similar behavior is observed for the SH Ext. However, opposite behavior is observed for the Tropics, with larger sources for the OCO-2 inversions. The differences between both inversions could be related to cross-talk between the zonal bands given the sparse coverage of in situ data in the Tropics.

**Line 525.** "we can again observe a consistency in OCO-2 across the priors "

You do perhaps mean " consistency between OCO-2 and the priors" ? We did mean across the priors. Similarly to line 520. We changed the OCO-2 part of the sentence however by: We can again observe a consistency across the priors of the OCO-2 inversions.

**Line 526.** "(with sources for OCO-2 inversion)". I don't understand what this means. **We reformulated this sentence:** MOre and ISMOre have a smaller sink in 2015 (with sources for OCO-2 inversion) compared to the other inversions in order to balance the 0.5 PgC/yr smaller fires that FIREMo gives.

By:

More has a smaller sinks in 2017 and 2018, but has a source in 2015 (larger in 2016) compared to the other inversions, in order to balance the 0.5 PgC/yr smaller fires that FIREMo gives.

**Line 530.** "This could then explain why we observe stronger sinks with in situ than OCO-2 posterior NEE emissions. " Which figure can we observe this? **We can observe this in Figure 10 of the pre-print (now Fig. 7).**

**Line 532.** "different with a Tropical sink in all years except in 2015 and 2016 ". Where can I see this? In figure 10? **Indeed, in Figure 10.c, we can observe that the net fluxes of IS give a net sink in 2017 and 2018, but a net sources for 2015 and 2016 (with the exception of ISCMS which has a net sink very close to 0).**

**Line 522-544.** This part needs to be rewritten. I feel that the key message is somewhat lost in this extensive summary of differences between inversions and regions. Make sure the following items are expressed in a concise manner:

- Sinks of OCOcms and IScms are generally weaker than the other inversions. Suggest sensitivity from the imposed AGR.

- Global sinks are larger with in situ data, which is largely driven by larger sinks in the tropics. Possible culprit: sparse data coverage.

- While OCO2 inversions show larger sources over the tropics and larger sinks over SH Ext. Could be a compensating effect for the scaled down fire emissions in the tropics. **Done**

**Line 531.** At least from Figure 10 I conclude that the in situ inversions yield for all 4 years a tropical sink, not only 2017 and 2018 as you write. I also see sinks up to 2.5 PgC/yr. How does this relate to the reported sink of -0.5 PgC/yr? This sentence was referring to the net fluxes (see line 532), which indeed the net sinks of IS inversions are around -0.5 PgC/yr. But, as expressed to the previous comment, we rewrote this paragraph.

**Line 548.** "…we can see that the OCO-2 inversions have deeper net sinks over the Boreal regions than with OCO-2…" Perhaps you mean OCO-2 inversions have deeper net sinks in comparison to IS? Indeed, it was changed.

**Figure A6.** Panel titles do not agree with the caption labels. E.g. first panel shows North America, but caption says Boreal North America, and the second panel shows North Trop South America but the caption says temperate North America. Modified

**Line 554.** Clarify what you mean with "drop off sinks" Modified with: For instance, it seems that the sink decreased for 2018

**Line 555.** This is an example where you can be more concise and to the point: "…is balanced by the Tropical Asia (North and South) where net fluxes go from sources to sinks."

You can write this as: "…is balanced by sinks in Tropical Asia (North and South) " Changed

**Line 555.** An example of a sentence with inconsequent use of tense: "2015 was a large net sources of carbon (due to intense fires) while 2016, 2017 and 2018 are deeper sinks with IS over Northern Tropical Asia and sinks with OCO-2 over Southern Tropical Asia. "

I would write it like: In 2015 there were large net carbon sources (due to intense fires), while in the other years there were larger sinks over Northern Tropical Asia (with IS) and Southern Tropical Asia (with OCO-2) Changed

**Line 557.** An example of unnecessary sentence. "At the same time, posterior fluxes in Europe are anti-correlated with posterior fluxes over Northern Tropical Africa." The sentence that follows already makes the point of anticorrelations between Europe and Northern Tropical Africa. **Deleted**

**Line 559.** "where the post ENSO period has smaller sources in Northern Tropical Africa linked with smaller sinks in Europe (Fig. 11). "

I don't think this is true. The deepest sink for Europe appears in 2017, not in 2015. **We removed this sentence.**

**Line 560.** "more in line with…" more in line than what? IS inversions? Be clear please. **Done** even though the years of study were before 2015.

**Line 561.** Remove "carbon per year". It is redundant **Done**

**Line 563.** "no estimate that can be refuted at present. " Do you mean: there is no reliable benchmark for comparison? **Yes, we modified it.**

**Line 565.** by Houweling et al. (2015) **Done**

**Line 568.** "We can see that our inversions here are within the estimates observed in the study of Peiro et al. (2022)." You can skip this sentence. You already say something similar in lines 566-567. **Done**

**Line 570.** Another example of a some text that is very difficult to read

"Our re-balanced priors give the deepest sink in 2017 (in 2016 for CMS prior) which is observed as well in the posteriors net fluxes using OCO-2 and it is in opposition of the OCO-2 inversions of Peiro et al. (2022) which have deeper sinks in 2016. This is due to stronger fire emissions in 2017 compared

to the other years balanced with the respiration, and the differences between the two studies could be due to the re-balanced respiration. "

Can you not write something like:

A major difference between this study and Peiro et al. (2022) is that the rebalanced priors and posterior fluxes provide the largest sink in 2017, as opposed to 2016. This is likely a consequence of the larger fires and the subsequent rebalanced respiration that was derived in this study. **Changed**

**Line 575.** "but an agreement across priors within each observational constraint." What agreement across priors are you referring to? Both types of constraints (OCO-2 and IS) use the same set of prior emissions. So I don't understand this sentence.

Maybe if you write it like this it becomes clearer what you try to say:

Between all inversions the largest differences in fluxes appear between IS and OCO-2 constraints. However, across the different fire emissions we observe a split; on one hand inversions using FIREmo are similar to FIRE4, while inversions using CMS are more similar to FIRE3. That means fires have a larger impact on the posterior solution than the rebalancing of prior NEE to match the global AGR. **Changed**

**Line 579.** "are balanced with higher sources for the other regions that have net sources, regions mainly over the Tropics" please rephrase.

"are balanced with larger sources in other regions, mainly over the Tropics." **Changed**

**Line 586.** "lag between flux in the Tropics and observation by the in situ network" Why is there such a lag in the tropics and not in the extra-tropics? Is that because the distances are longer between the measurements sites and the major source/sinks regions? Please explain in the main text.

**We added the sentence:** The number of in situ observation is particularly low in the tropics compared to the extra-tropical southern and northern hemispheres (Fig. 2 of Peiro et al., (2022)). One possible explanation is the lag between flux in the Tropics and observation coverage by the in situ network, which could be aliasing flux signals in time, though this hypothesis is difficult to test.

**Line 589.** "FIREMo and FIRE4 drop off for 2017 but FIRE3 driven fluxes do not."

Please be clear. Do the fire emissions become smaller or the inferred NEE fluxes become smaller? Which region?

Overall I think section b needs to be edited. The main points should be addressed in a more clearer and concise way. The current summary of results is very extensive and long which makes it difficult for the reader to extract the key points.

The key points from section b that needs to be highlighted in a more concise manner:

- Between IS and OCO-2 inversions there are persistent differences in posterior NEE

- Some of these differences are caused by differences in data coverage, lag between flux and observation, cloud fraction, etc.

- Larger sinks with OCO-2 in North America and Europe, while larger sinks with IS in Asia.

- Independent of observational constraints: the sinks in the tropics are generally smaller while there are larger net sinks in the NH Ext.

- Independent of observational constraints: Generally smaller sinks during El Nino in the tropics.

**We rewrote the section for better clarification and higlighted the key points at the end of the section.**

**Line 638.** "Additionally, for the 2015-2018 period, the posterior biases were ∼ 7 ppb underestimated TCCON values while the priors were ∼ 13 ppb overestimated TCCON values." Change to: "In comparison to TCCON, for the 2015-2018 period, the posterior biases were underestimated by 7 ppb, while the priors were overestimated by 13 ppb" **Done**

**Line 644.** "over the Northern latitudinal" change to: "over the Northern hemisphere" **We modified this paragraph with a new figure evaluating the mixing ratio for each TCCON site. See comments Reviewer #2.**

**Line 654.** "In Southern Hemisphere, MOre prior has smaller biases than GFED4re." Are these differences significant at all? All lines seem to be on top of each other in Figure 14. Are the differences significant in comparison to the measurement precision of TCCON CO2? Please elaborate on this. We have a new figure for the TCCON evaluation which give an evaluation against each TCCON sites. This new evaluation allows a better visualization of the differences between the inversions. As mentioned in the manuscript line 210 "The global monthly means of the total column CO 2 measurements have accuracy and precision better than 0.25% (less than 1ppm) relative to validation with aircraft measurements". Wunch et al., (2010) have shown that any differences with magnitudes less than 0.4 ppm could be attributable to TCCON station site-to-site biases. The evaluation at the Ascension Island site, for instance, biases for ISCMS, IS3re, IS4re and ISMOre are respectively of -0.61, -0.63, -0.56 and -0.55 ppm. The differences and biases cannot be attributed to the TCCON measurement precision. For this tropical site, a bias reduction of 0.8 ppm (0.6 ppm) with ISMOre is obtained compared to IS3re (ISCMS).

**Table 6.** I suggest to rename the labels to FIRE3, FIRE4 and FIREMo We removed the table as we instead annotated the biases directly with each TCCON site evaluation to reduce paper length and for more clarity.

**Line 694.** Please discuss the discrepancy between your inversion study and Nechita-Banda et al. regarding the 2015 Indonesian emissions. They found emissions of 0.5 PgC, which is not only more than GFED4, but also more than the GFED3 estimate reported in Fig. 9 of your study.

What is exactly mentioned in the paper of Nechita-Banda et al., 2018 is "Our estimates of CO emissions can be used to quantify the release of gaseous total carbon emissions to the atmos- phere (which includes CO2, CO, CH4 and NMVOC). For this conversion, we need to use biomass burning emission factors, which are quite uncertain. Based on our range of results and a range of emission factors available in the literature [14,15,39], we find that a range of 0.35–0.60 Pg C was emitted from the 2015 fires in Indonesia and Papua. ". They did not find emissions in 2015 of 0.5 PgC but a range of 0.35-0.60 PgC. Our CO2 fire emissions for the southern tropical south Asia give exactly 0.37 PgC for FIRE3, 0.33 PgC for FIRE4 and 0.35 PgC for FIREMo (which can be seen in Fig. 9 of our pre-print). We calculated the emissions for 2015 over the same Indonesia and Papua region of Nechita-Banda, and we found fire emissions of 0.41 PgC for FIRE3, 0.37 PgC for FIRE4 and 0.39 PgC for FIREMo. These

fire estimations are included in the range found by Nechita-Banda et al. There is, consequently, an agreement with our CO2 fire estimates and those found in Nechita-Banda et al., (2018). We consequently added: Nechita-Banda et al., (2018) converted their CO fire emissions in CO$_2$ emissions using emission factors and estimated that a range of 0.35-0.60 PgC was emitted in Indonesia and Papua from the 2015 fires. We calculated our fire CO$_2$ emissions over the same region and found 0.41 PgC, 0.37 PgC and 0.39 PgC for FIRE3, FIRE4 and FIREMo respectively. Our fire CO$_2$ estimates are hence in agreement with those found by Nechita-Banda et al., (2018).

**Line 695.** If GFED4s is able better capture small fires then please explain why GFED3 predicted larger emissions for Indonesia in 2015. Is this related to higher fuel loads in the older model? We found lower fire emissions for southern tropical south Asia with GFED4 compared to GFED3 in 2015. Even though our study period is different, this low GFED4 fire emissions compared to GFED3 were also found in the study of Shi et al., 2015. A possible explanation could indeed come from the CASA biogeochemical model predicting higher biomass densities than with the new version. Additionally, fuel loads in GFED4 for savanna and grassland have been found lower than measured in the field. We added in the manuscript: As mentioned previously, we know that GFED4.1s has information of small fires compared to GFED3 which allow better accuracy particularly over the Tropics where peat fires are important. However, we can see lower FIRE4 emissions than FIRE3 for southern tropical south Asia, similarly to what Shi et al., (2015) have found for the 2002-2012 period. A possible explanation could be that the CASA biogeochemical model of GFED3 predicts higher biomass densities than with the new version used in GFED4. Validation against fuel loads measured in savanna and grassland field have been found higher than with GFED4 (Randerson et al., (2012), Giglio et al., (2013)).

**Line 702.** "where IS4re and ISMore have sources of carbons compared to the IS constrained with the GFED3 fire, showing then higher net sources with GFED4 and MOPITT than with GFED3 fires"

change to "where IS4re and ISMore derive carbon sources in contrast to IScms that derives a carbon sink with GFED3 fires." We have changed to: This is particularly shown for the IS inversions where IS4re and ISMore have higher net sources of carbons compared to the IS constrained with GFED3 fires.

**Line 704.** "the CO2 posterior emissions using MOPITT CO information were able to catch the seasonality"

Please clearly state that you referring here to the CO2 fire emissions and not NEE. The GFED emissions also show seasonality. Is there independent proof MOPITT derived fires show a better seasonality? I would think the smoke could have hampered the MOPITT observations just as much as the MODIS observations. Please elaborate on this. We changed with: Moreover, FIREMo was able to catch the seasonality of fires over southern tropical Asia during the El Nino event, compared to the other priors using GFED inventory. As discussed in NechitaBanda et al., (2018) and van der Laan-Luijkx et al., (2015), GFED4 does not capture fire seasonality due to the use of burned area, compared to GFAS. We already mentioned this line 416 of the pre-print. The burned are may be more sensitive to the initial stages of the fire than the continued burning. GFAS based on the active fires product of MODIS seems to capture fire seasonality compared to GFED (NechitaBanda et al., (2018) and van der Laan-Luijkx et al., (2015)). Figure A4 of the pre-print shows a similarity in seasonality between our FIREMo emissions and GFAS from van der Laan-Luijkx et al., (2015), with a fire peak later than GFED4. However, in both GFED and GFAS method (and similarly for MOPITT), the detection of fires underneath clouds and below the canopy is difficult. But, FIREMO emissions, compared to FIRE3 and FIRE4, has the advantage of combining optimized fire emissions with local observations.

**Line 706.** "It is thus important to include CO fire emissions over this region to improve estimates and constrain CO2 NEE and Fire emission with both OCO-2 and IS data constraints "

I don't think this sentence covers your methodology correctly. I suggest to rephrase it differently: "It is thus important to use CO observations to constrain estimates of CO2 fire emissions, and subsequently constrain NEE with OCO-2 and IS observations" Changed

**Line 709.** "finer enough" to "fine enough" Done

**Line 710.** "Additionally, the emission factors used in the emission ratio are characteristic of vegetation type but are not dependent of spatial or temporal scales. "

I think you try to say that emission factors lack spatial and temporal variability to account for the full dynamics range of combustion characteristics. That is different than saying "not dependent of". We changed the sentence with this suggestion

**Line 713.** "between the priors using CO fire emissions and the other prior fire emissions "

I don't understand this comparison. All your experiments use fire emissions. Do you mean comparing optimized fire emissions (FIREmo) with non-optimized fire emissions (FIRE4)? Changed with: This could explain the differences observed over some regions of the Tropics between FIREMo and the other prior fire CO2 emissions.

**Line 713.** "Further works are needed " to "Further work is required" Done

**Line 716.** "this tropical region". This is a new paragraph, so which region are you talking about now? We changed: The data used to constrain inversions is very important. We could see up to 0.4 PgC/yr differences between OCO-2 and IS inversions in tropical regions. This bring us to the importance of the data assimilated in the inversions but also about the priors used in the inversions concerning the different sectors (fire and terrestrial emissions).

**Line 730.** "The IS results suggest a very strong sink in North Asia " Do you think this is mostly an inversion artefact due to low data coverage here? As mentioned by Reviewer #2, this paragraph is more a result than a discussion. We consequently removed it. The answer to the question is yes. The disagreement between the OCO-2 and in situ inversions might be driven by the differences in the amount of data assimilated since both inversions have the same transport model and inverse setup. We know that there are fewer in situ than OCO-2 observations above northern Asia, and particularly above the boreal forest of Eurasia, which is an important area for sources and sinks of atmospheric CO2 (Houghton et al., 2007; Siewert et al., 2015).

**Line 746.** "OCO-2 measurements are globally distributed, but seasonally varying coverage.", Another difference is that OCO-2 represents a column density as opposed to a concentration in the

lower boundary layer. Indeed, and this difference was already mentioned. But for clarity, we changed the sentence: Certainly some of this mismatch is due to sampling differences, as most of the in situ measurements assimilated here are taken in the atmospheric boundary layer in the Northern Extra-Tropics, whereas OCO-2 measurements are globally distributed, but seasonally varying coverage.

To:

Part of this discrepancy is certainly due to: (i) most of the in situ measurements assimilated here are taken in the atmospheric boundary layer while OCO-2 represents a column density; and (ii) most of the in situ measurements are in the northern extra-tropics, whereas OCO-2 measurements are globally distributed, but with seasonally varying coverage.

**Line 771.** Looking at the posterior fluxes and the TCCON comparisons in Table 6, I hardly see any differences in performance between MOre and the other inversions (GFED3re, GFED4re and CMS). So the added value of optimizing fire emissions before optimizing NEE is not very apparent. On the contrary, your results seem to be very insensitive to the optimized fire emissions. This outcome should be presented much clearer in your discussion and conclusions. We already have modified the sentence:

Regarding the question of the importance of the prior and the question of which prior could do better than the other, we have seen through the results and the evaluation, than no simulation is better than the other on average. Even if the biases seem to have been reduced with FIREMO for certain sites (such as Ascension island for instance), they are in the same order as the other a priori biases for other site. On average and and overall, the added value of optimizing fire emissions before optimizing NEE is not very apparent. Our results seem, overall, to be very insensitive to optimized fire emissions. Philip et al., (2019) performed simulation experiments with different NEE priors, and concluded that posterior NEE estimates are insensitive to prior flux values. But they found large spread among posterior NEE estimates in regions with limited OCO-2 observations. Our results suggesting that OCO-2 inversions are relatively insensitive to prior in most regions, are consistent with Philip et al., (2019), and not only for OCO-2 inversions but also for IS inversions.

We also added in the conclusions: The added value of fire emission for NEE optimization is not apparent. Our results seem hence to be very insensitive to optimized fire emissions.

**Line 776.** "We found that a priori CO2 flux uncertainties are substantially reduced when matching the NOAA AGR as well as CO/CO2 ratio but not strong enough compared to a re-balanced GFED and GFED4.1s NEE, and suggest hence for future work the development of joint CO-CO2 inversions with multi-observations for stronger constraint in posterior CO2 fire and biospheric emissions. "

This is a key sentence as it wraps up your paper. However, even after reading the paper I have difficulty to fully understand it.

I tried to rephrase it in three separate sentences. Is my interpretation correct?

"We found that CO2 fluxes are more robust if the NEE and fire emissions are rebalanced in order to match the NOAA AGR as well as the satellite-based CO constraints. However, a more reliable NEE is obtained if we utilize in situ and satellite-based CO2 constraints. This opens new avenues for future research for the development of a joint CO-CO2 inversion framework that uses multiple streams of data to improve the fire and biosphere emissions."

We changed the sentence with: We found that CO2 fluxes are more robust if the NEE and fire emissions are rebalanced in order to match the NOAA AGR. However, a more reliable NEE is obtained with the assimilated data, using either in situ or satellite-based CO2 constraints. This opens new avenues for future research for the development of a joint CO-CO2 inversion framework that uses multiple streams of data to improve the fire and biosphere emissions.

   Firstly, I think that the posterior CO fields from the MOPITT flux inversion should be evaluated. I would recommend comparing the posterior fields to TCCON XCO in the same way that the posterior CO2 fields were evaluated.

We indeed acknowledge that an evaluation of the CO fluxes should be present in the paper. However to not create an overly long paper, we have added this evaluation in supplement information.

We performed an evaluation of the priors and posteriors mixing ratio against each TCCON site as shown in the Fig. S10 below.

[Figure]

**Figure S10.** Annual mean difference in ppb between the prior (in green) and posterior (in blue) against each TCCON site. Mean and coefficient of correlation are specified for each year. From top left through bottom right are the annual years 2015, 2016, 2017 and 2018.

We corrected and added the figure number in the sentence page 32 line 636: For evaluation of our CO posteriors and priors emissions with TCCON (not shown here), we found that biases between MOPITT CO posterior simulated mixing ratio and TCCON were lower than biases with the CO priors with on average a ∼ 5 ppb reduction each year. Additionally, for the 2015-2018 period, the posterior biases were ∼ 7 ppb underestimated TCCON values while the priors were ∼ 13 ppb overestimated TCCON values. To In comparison to TCCON, for the 2015-2018 period, the CO posterior biases were underestimated by 7 ppb, while the CO priors were overestimated by 13 ppb (Fig. S10). Even if the posterior biases are lower than the prior biases, the underestimation observed in Fig. S10 against TCCON could explain the low fluxes observed of the FIREMo compared to the other fire estimates over some regions. We can observe an underestimation of the posterior CO mixing ratio of ~ -12 ppb in 2015 at the Ascension Island site, while the a priori CO mixing ratio has an overestimation of 5 ppb in 2015. However, the biases at the Darwin TCCON site give -3 ppb for 2015-2016 (-0.5 ppb for 2017-2018) with the posterior and 20 ppb for 2015-2016 (22 ppb for 2017-2018) with the prior. This gives

the impression that our inversion is not getting the best fluxes for Ascension Island, but we can see that this is not the case for other tropical locations. Ascension Island is known to be impacted with Saharan dust and therefore the posterior simulated concentration could be biased due to aerosols.

Second, it would be useful to characterize whether employing this MOPITT-based biomass burning estimate improved the inversions in any way. In particular, were the inversions able to better fit to OCO-2 and IS measurements that were strongly impacted by biomass burning emissions? You could perform this comparison by running a tagged tracer experiment for biomass burning emissions and then look at the data-model mismatch for measurements that had a large biomass burning signal.

We performed a tagged tracer experiment, comparing the data-model mismatch to OCO-2 and IS retrievals for all simulations. As these results allow to evaluate how the priors and posteriors fit the data assimilated, the results have been added to the Validation section of the paper with the figure and discussion as noted below. We also modified the evaluation plots in the paper to have a site-by-site TCCON evaluation instead of an evaluation by latitudes (see Fig. 10).

3.2.3 Evaluation of the simulation

3.2.3.a Evaluation of the inversions to fit the OCO-2 retrievals and IS data

The global distributions of OCO-2 retrievals over the 2015-2018 period (Figure 9.a) shows a latitudinal gradients from north to south with higher XCO2 concentrations in the tropics and the northern hemisphere. High land values (no higher than 409 ppm) are observed over east Asia, north west Africa, north tropical south America. Figure 9.b shows the global distributions of IS data with higher number of observations in the northern hemisphere than the tropics or the southern hemisphere. High XCO2 concentrations (higher than 409 ppm) can be observed for temperate north America and near the coast of east Asia. The regional mean differences between the prior or posterior and the OCO-2 retrievals (IS data) are summarized in Table S1.

[Figure]

**Figure 9.** Spatial distributions of the CO2 total column (XCO2). Mean distribution of OCO-2 retrieval (a) and In-Situ data (b) over the 2015-2018 period. Annual difference between the prior of each simulation (CMS (2nd row), prior3 (3rd row), prior4 (4th row) and priorMO (5th row)) and OCO-2 in the 1st column (IS in the 3rd column). Annual difference between the posterior simulation of each simulation (row similar to the priors) and OCO-2 in the 2nd column (IS in the 4th column). Results are in ppm.

The prior have larger differences with the OCO-2 retrievals than the posteriors. The prior3 (using both FIRE3 and NEEre3, see Fig. 2) better fit the OCO-2 measurements than the other priors for the southern hemisphere and the tropics (Fig.9 and Table.S1). The priorCMS however does not fit the OCO-2 measurements with high bias between 3 and 4 ppm. The large difference is also observe with the IS measurements. For the IS inversions, the differences between priors and posteriors with the IS data are very similar. This result suggests that the inversion does not change much from the prior, but

this result can be explain due to the small number of observations available in these regions. While the optimized concentrations fit the OCO-2 retrievals quite well compared to the priors, suggesting the inversion's ability to fit the data. For the comparison among the simulations, there is no large difference between the different simulations and the data, particularly for the optimized CO2 measurements.

| Regions | OCO | | | | | | | | IS | | | | | | | |
|---|---|---|---|---|---|---|---|---|---|---|---|---|---|---|---|---|
| | Prior CMS | prior3 | prior4 | Prior MO | CMS | GFED3re | GFED4re | MOre | Prior CMS | prior3 | prior4 | Prior MO | ISCMS | IS3re | IS4re | ISMOre |
| North America | 0.64 | 0.20 | 0.16 | 0.24 | -0.071 | -0.073 | -0.074 | -0.073 | -90.94 | -91.38 | -91.42 | -01.34 | -91.61 | -91.61 | -91.61 | -91.61 |
| North T.S. America | 0.15 | 0.033 | 0.043 | 0.050 | -0.017 | -0.016 | -0.017 | -0.017 | 6.18 | 6.07 | 6.08 | 6.09 | 6.03 | 6.03 | 6.03 | 6.03 |
| South T.S. America | 0.046 | 0.0045 | 0.015 | 0.014 | -0.0078 | -0.0073 | -0.0078 | -0.0074 | 5.99 | 5.95 | 5.96 | 5.96 | 5.93 | 5.93 | 5.93 | 5.93 |
| Temp.S. America | 0.30 | 0.078 | 0.14 | 0.13 | -0.034 | -0.033 | -0.034 | -0.033 | 19.91 | 19.69 | 19.75 | 19.74 | 19.59 | 19.59 | 19.59 | 19.59 |
| Temp.N.Africa | 0.068 | 0.020 | 0.017 | 0.024 | -0.0078 | -0.0078 | -0.0079 | -0.0078 | -0.75 | -0.80 | -0.81 | -0.80 | -0.82 | -0.82 | -0.82 | -0.82 |
| North.T. Africa | 0.14 | 0.029 | 0.045 | 0.048 | -0.0097 | -0.0097 | -0.0095 | -0.010 | 5.88 | 5.77 | 5.79 | 5.79 | 5.73 | 5.73 | 5.73 | 5.73 |
| South.T. Africa | 0.15 | 0.035 | 0.068 | 0.061 | -0.0094 | -0.0095 | -0.010 | -0.0098 | 13.08 | 12.97 | 13.00 | 13.00 | 12.92 | 12.92 | 12.92 | 12.92 |
| Temp.S. Africa | 0.21 | 0.056 | 0.10 | 0.094 | -0.023 | -0.022 | -0.022 | -0.022 | 27.09 | 26.94 | 26.98 | 26.98 | 26.87 | 26.87 | 26.87 | 26.87 |
| North Asia | 0.61 | 0.20 | 0.15 | 0.23 | -0.060 | -0.060 | -0.062 | -0.061 | -13.99 | -14.40 | -14.44 | -14.37 | -14.64 | -14.63 | -14.63 | -14.63 |
| North.T. Asia | 0.24 | 0.071 | 0.079 | 0.094 | -0.011 | -0.011 | -0.012 | -0.012 | -10.67 | -10.84 | -10.83 | -10.81 | -10.91 | -10.91 | -10.91 | -10.91 |
| South.T. Asia | 0.18 | 0.051 | 0.077 | 0.078 | -0.015 | -0.015 | -0.016 | -0.016 | 10.75 | 10.62 | 10.64 | 10.65 | 10.57 | 10.57 | 10.58 | 10.58 |
| Trop Australia | 0.082 | 0.022 | 0.037 | 0.036 | -0.0073 | -0.0074 | -0.0073 | -0.0073 | 3.42 | 3.36 | 3.38 | 3.38 | 3.34 | 3.34 | 3.34 | 3.34 |
| Temp Australia | 0.38 | 0.095 | 0.17 | 0.16 | -0.048 | -0.047 | -0.047 | -0.047 | 33.68 | 33.40 | 33.47 | 33.46 | 33.27 | 33.27 | 33.27 | 33.27 |
| Europe | 0.25 | 0.083 | 0.059 | 0.094 | -0.032 | -0.032 | -0.032 | -0.032 | -10.13 | -10.30 | -10.33 | -10.29 | -10.40 | -10.40 | -10.40 | -10.40 |

**Table S1.** Summary of CO2 model (prior or posterior)-data comparison against OCO-2 retrievals and IS data for each OCO-2 MIP regions. Values are in ppm.

3.2.3.b Validation against TCCON data

[revised manuscript text omitted]

2. The CO2 flux inversion configuration is insufficiently described. Are the CO2 flux inversions optimizing ocean and NEE fluxes? And what are the prior errors applied to these quantities? And of great relevance to the results, how do the prior errors vary between the different experiments? I would expect the posterior regional NBE fluxes to be very sensitive to the prior error statistics, particularly for the IS inversion.

The $CO_2$ flux inversions are optimizing the ocean and NEE fluxes, this is and was already mentioned in the paper page 15, line 357: "*We optimized CO biomass burning emissions and $CO_2$ biospheric and oceanic emissions on a weekly basis*".

Concerning the uncertainties, we added the following sentences page 13 line 319 : The uncertainties in the prior fluxes are derived from different climatological fluxes with exponential spatio-temporal correlation assumed. For the oceanic component, the horizontal correlation is 1000 km and the

timescales is 3 weeks, while for the terrestrial component, length and timescale are 250km and 1 week. These uncertainties are applied similarly to all experiments.

3. I find much of the text to be quite awkwardly worded, which can make the manuscript hard to follow. In addition, there are a number of rather sloppy mistakes in the description of experiments, equations, and variable names. I have flagged several issues in my specific comments, but not all. I strongly recommend that the authors go through the manuscript carefully to fix these issues.

We thanks the Reviewer for this remark and have corrected the manuscript accordingly. We also corrected the manuscript relative to the specific comments below. In addition, we have reviewed the paper to make sure that all variable names were correct. We particularly specified the differences between the priors used, posteriors simulation names using either OCO-2 or IS measurements. Corrections can be found in the track corrected manuscript with red for sentences deleted and blue for sentence we have added.

**Specific comments:**

L2: "used as a tracer of CO2" to "co-emitted with CO2" Corrected

L5-6: This statement is confusing: "These CO2 fire emissions allow us, then, to estimate adjusted CO2 Net Ecosystem Exchange (NEE) and respiration which are then used as priors for CO2 inversions" **We have changed this sentence with the suggestion of Reviewer #1:** These optimized CO2 fire emissions (FIREMo) are used to re-balance the CO2 Net Ecosystem Exchange (NEEmo) and respiration (Rmo) with the global CO2 growth rate. Subsequently, in a second step, these rebalanced fluxes are used as priors for an CO2 inversion to derive the NEE and ocean fluxes constrained either by the Orbiting Carbon Observatory 2 (OCO-2) v9 or by in situ CO2data.

L20-22: This statement is confusing: "Evaluation with TCCON suggests that the re-balanced posterior simulated give biases and accuracy very close each other where biases have decreased and variability matches better the validation data than with the CASA-GFED3." **We have changed the sentence with :** Evaluation with TCCON data shows lower biases with the three re-balanced priors than with the prior using CASA-GFED3. However, posteriors have average bias and scatter very close each other, making it difficult to conclude which simulation is better than the other.

L48: "atmospheric measurements" to "atmospheric measurements of CO2" changed

L68: Define "terrestrial biosphere fluxes", to some this could include biomass burning. We changed the sentence by : "CO2 emissions are separated into four categories: anthropogenic sources, ocean fluxes, terrestrial biosphere fluxes (meaning the sum of the photosynthesis and respiration) and fires. "

L234: "optimal estimation"? I would assume TM5-4DVar uses 4DVar, correct? Yes, TM5-4DVar uses 4DVar, but this sentence used in the introduction was applied in a general context of the assimilation system. For clarity, we have changed this to "assimilation system".

Figure 2 caption: "Localisation" should be "Location" Changed

L286: A Gaussian correlation length of 1000 km is also applied to CO? Hard to think of a physical reason for this? As mentioned in Meirink et al., (2008), where they performed sensitivity experiments for inversion with TM5, "The background error covariance matrix B is split into spatial and temporal error correlation matrices [..]. Spatial correlations are modeled as Gaussian functions of the distance between grid cells […]. Information on spatial correlations of emission errors is generally lacking. Therefore we specify spatial error correlations simply by Gaussian functions of distance." They performed an experiments using 1000 km instead of 500 km for the prior error correlation. The experiment performed with larger a priori error correlation lengths shows larger uncertainty reduction. As explained, the region of influence of the observations is larger in this experiment, so that they effectively constrain the emissions in more grid cells.

We changed this sentence by : Spatially, a Gaussian correlation length scale of 1000 km is used, as justified in Meirink et al., (2008), while we assume the prior errors have a temporal correlation scale of 4 days.

Figure 3: Maybe try a different colorbar, it is hard to see the different regions.

We have changed this plot with this new colorbar:

[Figure]

L320: "a priori" to "from CO2 data alone" Changed

Equation 3: The notation "max(FIRE3 – FIREx,0)" is not typical notation. This appears to indicate that the max is taken down the zeroth dimension of the array, but the dimensions of the array have not been defined in the text. Please revise.

In order to give more indication of this notation, we added: This equation means that the difference between FIRE3 and FIREx is cut off at 0 when the difference is negative. With this equation we only consider the positive difference (when we have lower FIREx emissions than FIRE3).

L354-358: This is a methods section not a results section. We moved it at the end of the methodology section.

L380-385: It is unclear what is meant by "bias satellite data due to cloud coverage". Please explain exactly how cloud coverage biases satellite data. Biases can happen at regional scale due to poorly modeled scattering by clouds and aerosols in the trace gas retrievals (Wunch et al,. (2017)), which can have an impact on the flux estimation (Crowell et al., 2019, Chevallier et al., 2007).

L380-391: This whole section is quite unclear. Consider re-writing. We changed the paragraph :

Over Temperate Northern Africa, this can be a result of bias satellite data due to cloud coverage, giving the CO posterior emissions closer to the prior GFED4.1s. However, for Northern Temperate America, the prior might be well enough constrained and validated over this region, to give similar CO fire emissions than the posterior CO. For Temperate North Africa, MOPITT posterior fires remain close to

the prior GFED4.1s estimates, meaning that the inferred emissions are consistent with GFED4.1s. This region is known to have a lot of Saharan dust transported across the Atlantic ocean and towards Europe most of the year and to be largely cloudy during the wet season of the African monsoon (from May to August), which could explain the posterior emissions being close to the prior. This is also the case for Northern Tropical Africa, however, MOPITT posterior fires has lower emissions than the prior GFED4.1s estimate. But, we still need further investigation over Northern Tropical Africa to understand why GFED4.1s and MOPITT are different each other. Tropical South America is also known to have a cloudy coverage limiting satellite observations, but over this region we only observe similar emissions between the prior and the posterior for Northern Tropical America, even if MOPITT has slightly higher emissions, while for Southern Tropical America, differences between the prior and the posterior are strong.

To :

For North Temperate America, posterior emissions remain close to the prior estimates, suggesting that the inferred emissions are consistent with GFED4.1s. Comparable results are also observed for Temperate North Africa. However, this region is known to have a lot of Saharan dust transported across the Atlantic Ocean and towards Europe most of the year, which could explain the posterior emissions being close to the prior as those MOPITT soundings have largely been removed by pre-screeners . North Tropical Africa is not only affected by dust, but it is also largely affected by clouds during the wet season of the African monsoon (from May to August), which could lead to errors in retrievals that pass the pre-screeners. The combination of clouds and dust could explain the MOPITT posterior fires having lower emissions than the prior GFED4.1s estimate. But further investigation into North Tropical Africa is needed. Even though the prior is higher than the posterior for tropical Africa, in opposition to the previous multi-species study of Zheng et al., (2018), the posterior emissions better fit MOPITT measurement than the prior (Fig. S4). Tropical south America (including North Tropical South America and South Tropical South America) is also known to have cloud coverage limiting satellite observations. We however observe similar emissions between the prior and the posterior for the northern region, with slightly higher emissions for MOPITT. For the southern region, differences between the prior and the posterior are strong. The cloud coverage might explain this behavior, but further investigation are needed for these two regions.

L389-390; Fig 12 cap: There are not regions defined as "Tropical America". I think these should be Tropical South America. We have removed this figure to have only one figure grouping all regions, as expressed by Referee #1.

I found figure 5 very hard to look at. The subset bar plots are far too small. I would recommend re-plotting similar to Figure A2. We took this comment in consideration and made this following plot instead:

[Figure]

**Figure 4.** Annual CO fire emissions by vegetation type over the OCO-2 MIP regions between fire priors (hatch bars) and fire posterior from 2015 through 2018. Vegetation types are representing by

colors : agriculture in gray, deforestation in yellow, savanna in dark-red, temperate forest in blue, peat land in red and boreal forests in green. Emissions are annually in TgCO/yr.

L443: "meridional", should this be "zonal"? Indeed, we changed it.

Figure 8 and others: "OCOcms" – I did not see this defined anywhere. Changed

L554: "the 2018 drop off sink". This is unclear. We changed it by : For instance, it seems that the sink decreased for 2018

L560: "Emissions estimated observed with OCO-2"? Changed in "Emissions estimated with OCO-2".

L673: "fire emissions and plant respiration (and hence net fluxes)". These two quantities alone do not combine to the net flux. In this sentence the net fluxes refer only to the plant respiration. But for clarity, we changed by: In addition, as fire emissions and plant respiration (terms included in the net fluxes) are difficult to disentangle, we re-balanced ...

L696-699: I do not understand the sentence "Over Southern Tropical and Northern Tropical Asia, the combination of the spatio-temporal variability of MOPITT CO fire and the GFED4.1s emissions information included in the prior fire emissions of the CO inversion might bring additional information in the emission ratio and hence in the fire prior used in CO2 inversions." We decided to remove this sentence as it is detailed later lines 703-705.

L727-739: This paragraph seems to be a description of the results rather than a discussion topic. The relevance to the main findings of the study also seem unclear, I would suggest removing. We removed this paragraph. And we added the last part of this paragraph in the results section page 32 line 620.

L740: What findings? The sentence line 740 is : *Returning to the question of importance of the prior, it would seem that the simulation experiments in Philip et al. (2019) hold for our experiments as well, i.e. that OCO-2 inversions are relatively insensitive to the prior in most regions.* We reformulated this sentence by : Regarding the question of the importance of the prior and the question of which prior could do better than the others, we have seen through the results and the evaluation, than no simulation is better than the other on average. Even if the biases seem to have been reduced with FIREMO for certain sites (such as Ascension island for instance), they are in the same order as the other a priori biases for other site. On average and and overall, the added value of optimizing fire emissions before optimizing NEE is not very apparent. Our results seem, overall, to be very insensitive to optimized fire emissions. Philip et al., (2019) performed simulation experiments with different NEE priors, and concluded that posterior NEE estimates are insensitive to prior flux values. But they found large spread

among posterior NEE estimates in regions with limited OCO-2 observations. Our results suggesting that OCO-2 inversions are relatively insensitive to prior in most regions, are consistent with Philip et al., (2019), and not only for OCO-2 inversions but also for IS inversions.

L742-748: This is not an assertion, but a direct result of mass balance. Over a single year, there can be differences in the growth rate due to sampling. However, if averaging over a few years, the signal will be well mixed. It is unclear what is being referred to with these statements. Is it referring to the "Global" fluxes shown in Figure 10? If so, this is only showing the land flux, right? The difference between the IS and OCO-2 "Global" land fluxes should be compensated for by differences in the ocean fluxes. Please clarify. We indeed referred to mass balance. This is referring for instance to the mass balance between Europe and north Africa. According to Feng et al. (2016) the large sink over Europe inferred from GOSAT data was caused by large biases outside of the region, which for mass balance, the inversions was removing larger CO2 over Europe, in agreement with Reuter et al. (2014) and Reuter et al. (2017).

**Citation**: https://doi.org/10.5194/acp-2022-120-RC1

Meirink, J. F., et al. (2008), Four-dimensional variational data assimilation for inverse modeling of atmospheric methane emissions: Analysis of SCIAMACHY observations, J. Geophys. Res., 113, D17301, doi:10.1029/2007JD009740.

Wunch, D., Wennberg, P. O., Osterman, G., Fisher, B., Naylor, B., Roehl, C. M., O'Dell, C., Mandrake, L., Viatte, C., Kiel, M., Griffith, D. W. T., Deutscher, N. M., Velazco, V. A., Notholt, J., Warneke, T., Petri, C., De Maziere, M., Sha, M. K., Sussmann, R., Rettinger, M., Pollard, D., Robinson, J., Morino, I., Uchino, O., Hase, F., Blumenstock, T., Feist, D. G., Arnold, S. G., Strong, K., Mendonca, J., Kivi, R., Heikkinen, P., Iraci, L., Podolske, J., Hillyard, P. W., Kawakami, S., Dubey, M. K., Parker, H. A., Sepulveda, E., García, O. E., Te, Y., Jeseck, P., Gunson, M. R., Crisp, D., and Eldering, A.: Comparisons of the Orbiting Carbon Observatory-2 (OCO-2) XCO2 measurements with TCCON, Atmos. Meas. Tech., 10, 2209–2238, https://doi.org/10.5194/amt- 10-2209-2017.

Chevallier, F., F.-M. Bre´on, and P. J. Rayner (2007), Contribution of the Orbiting Carbon Observatory to the estimation of CO2 sources and sinks: Theoretical study in a variational data assimilation framework, J. Geophys. Res., 112, D09307, doi:10.1029/2006JD007375.

**Reply to Meinrat O. Andreae comments**

I have some concern regarding the emission factors used in section 2.3.1b, page 12f. The authors base their values for the emission factors of CO and $CO_2$ on GFED4.1s, which in turn are based on a blend of Andreae and Merlet (2001) and Akagi et al. (2011). Newer estimates for these emission factors are available in Andreae (2019). These newer estimates, which are based on a much more comprehensive data base than the previous estimates, differ from the ones used here by as much as 30% in some cases. I wonder how much difference it would make if the updated emission factors would be used in the authors' calculations.

We thank Meinrat O. Andreae for the comments and remarks posted.

First, we would like to mention that all estimates (CMS-GFED3, GFED3, GFED4.1s and FIREMo) are based on the emission factors based on Andreae and Merlet (2001) and Akagi et al (2011). The CMS-GFED3 $CO_2$ emissions are based on this calculation, we consequently thought more judicious and correct to have all $CO_2$ emissions with the similar emissions factors for better comparison. In order to be more specific on that, we added the sentence page 12 line 304 : "For better comparison and as the CMS-GFED3 product (we will introduce later) used the emission factor of Andreae and Merlet (2001) and Akagi et al. (2011), we applied the same emission factors and consequently did not use the new estimate established by Andreae (2019)".

However, to answer the question if the use of a different emission factor would have give different conclusions, we calculated the FIREMo product with the emission factor of Andreae (2019) and compared it with the one used in our paper (FIREMo calculated with Akagi et al., 2011).

The figure below show the difference for the OCO-2 MIP regions between the two FIREMo estimates. We also added the NEE and net fluxes estimates based on the respective FIRE emissions. We can observe that for all regions over the globe, the difference between the two fires estimates are negligible (in the order of 15 TgC/yr) with higher estimates using Akagi et al., (2011) than with Andreae (2019). In our study, since the respiration are calculated using the fires estimates and re-balancing to match the global NOAA growth rate, the differences between both estimates in the net fluxes are completely negligible.

The small differences could have been also assumed, when looking at the Fig. 2 of Andreae (2019) paper, we can see that the ratio of MCE (Mole Combustion Efficiency calculated using CO and $CO_2$ emission factor) between this study and Akagi et al., (2011) has a ratio very close to 1. Additionally, the ratio of emission factor for $CO_2$ between both studies is also very close to 1. There is then no large differences between the two estimates for all vegetation types.

**2015, 2016, 2017, 2018 (from left to right) Annual Prior Flux (PgC/yr)**

[Figure]

*Figure 1. Differences between the prior emissions calculated with CO posterior fire emissions using Andreae (2019) versus Akagi et al., (2011) for the OCO-2 MIP regions. Fire emissions are in dark red bar, NEE emissions are in hatched bar and net flux are the dot/lines. Flux are in PgC/yr.*

One should also keep in mind, that in particular the EF for $CO_2$ and consequently the emission ratio $CO/CO_2$ are quite difficult to determine accurately in the field for a number of reasons. These include the difficulty of distinguishing the often relatively small fire inputs of $CO_2$ from large biospheric variability, the issue of variable background concentrations, and the problem of accounting for residual smoldering emissions that do not get lofted into the smoke plumes (Guyon et al., 2005; Burling et al., 2011; Yokelson et al., 2013). This introduces systematic errors in the $EF(CO_2)$ values that may well exceed 10%. While this problem obviously cannot be mitigated here, it should be at least pointed out to the reader as a significant source of uncertainty and possibly explored by a sensitivity study.

This is a good point that we acknowledge and indeed need to be mention in the paper. As one of the reviewer mentioned that the paper was too long in length ("*The lengthy descriptions in each section distracts from the main points of this original work. Perhaps some of detailed descriptions, figures and comparisons can be moved to a supplementary document.*"), we did not add a sensitivity study in the paper (we will consider that for future work), but we have added this sentence page 36 line 714: "However, the estimation of EF and consequently the emission ratio $CO/CO_2$ cannot be determined accurately in the field and can introduce systematic errors in the $EF(CO_2)$ values that may well exceed 10%. One challenge is separation of the information between small fire inputs of $CO_2$ (and hence their detection) from large biospheric variability. Other difficulties come from the issue of variable background concentrations and from smoldering emissions that are not projected into the smoke plumes (Guyon et al., 2005; Burling et al., 2011; Yokelson et al., 2013).".

Two minor issues:

In the caption of Table 2, van der Werf et al. (2017) should be cited explicitly (if the authors prefer to keep these emission factors).

We added the reference in the caption of Table 2.

I don't understand what is meant by the sentence: "Finally, the emission ratio for each vegetation type was divided to the posterior CO fire partitioned as used in Christian et al. (2003) and Basu et al. (2014)." (line 307f).

In this section, we explained how we calculated our FIREMo ($CO_2$ fire prior estimates based on CO posterior emissions from MOPITT). We break down our CO emission estimates within the 3x2 regions according to vegetation type using the GFED4.1s partitioning to get CO emission from each vegetation type for each grid box. Then we used the emission ratios measured by Van der Werf et al., (2017) to convert those into $CO_2$ emission per vegetation per grid box. For this conversion, the posterior CO fire partitioned are divided by the corresponding emission ratio of each vegetation type, such as the equation:

$$CO_{2i} = \frac{CO_i}{ER_{(CO/CO2)_i}}$$ with $i$ corresponding to the vegetation type (sava, borf, temf, peat, agri, defo).

To make sure this sentence is understood by the reader, we added page 13, line 308: "Finally, the emission ratio for each vegetation type was divided into the posterior CO fire partitioned for each

vegetation type (annotated i in the equation) as used in Basu et al., (2014) following the equation :

$$CO_{2i} = \frac{CO_i}{ER_{(CO/CO2)_i}} \quad ."$$
* * *
Akagi, S. K., Yokelson, R. J., Wiedinmyer, C., Alvarado, M. J., Reid, J. S., Karl, T., Crounse, J. D., and Wennberg, P. O., Emission factors for open and domestic biomass burning for use in atmospheric models: Atmos. Chem. Phys., 11, 4039-4072, doi:10.5194/acpd-10-27523-2010, 2011.

Andreae, M. O., and Merlet, P., Emission of trace gases and aerosols from biomass burning: Global Biogeochemical Cycles, 15, 955-966, 2001.

Andreae, M. O., Emission of trace gases and aerosols from biomass burning – an updated assessment: Atmos. Chem. Phys., 19, 8523-8546, doi:10.5194/acp-19-8523-2019, 2019.

Burling, I. R., Yokelson, R. J., Akagi, S. K., Urbanski, S. P., Wold, C. E., Griffith, D. W. T., Johnson, T. J., Reardon, J., and Weise, D. R., Airborne and ground-based measurements of the trace gases and particles emitted by prescribed fires in the United States: Atmos. Chem. Phys., 11, 12,197–12,216, doi:10.5194/acp-11-12197-2011, 2011.

Guyon, P., Frank, G. P., Welling, M., Chand, D., Artaxo, P., Rizzo, L., Nishioka, G., Kolle, O., Fritsch, H., Silva Dias, M. A. F., Gatti, L. V., Cordova, A. M., and Andreae, M. O., Airborne measurements of trace gases and aerosol particle emissions from biomass burning in Amazonia: Atmos. Chem. Phys., 5, 2989–3002, 2005.

Yokelson, R. J., Andreae, M. O., and Akagi, S. K., Pitfalls with the use of enhancement ratios or normalized excess mixing ratios measured in plumes to characterize pollution sources and aging: Atmos. Meas. Tech., 6, 2155-2158, doi:10.5194/amt-6-2155-2013, 2013.

---

## Author Response (AR2)

Reviews #1

We thanks the reviewer for the helpful comments. The feedback has helped us improve the paper. We addressed all comments below. The reviewer comments are in grey, our responses in black, sentences from the manuscript are in *italic*, and sentences in the revised manuscript are in blue.

Specific comments:

- Line 6: You can leave out "(FIREMo)". It is already introduced in the previous sentence.

We removed it.

- Line 16: "match the fires and observations". I don't think you match fires. You only match CO2 observations from OCO-2 and IS. NEE is the buffer, compensating for any missing or abundant fires.

The matching mentioned in this sentence was mainly referring to the modification applied in the net flux in balance with each fire emission estimates. The difference among the NEE prior emissions are a consequence of the balance between the respiration and the fires emissions (see equation 3).

For better clarity, we replace the following sentence:

"*However, at regional scale, we can observe differences in fire emissions among the priors, resulting in large adjustments in the NEE to match the fires and observations.*"

by:

"However, at regional scale, we can observe differences in fire emissions among the priors, resulting in differences among the NEE prior emissions. The derived NEE prior emissions are re-balanced in concert with the fires. Consequently, the differences observed in the NEE posterior emissions are a result of the balanced with fires and the match of $CO_2$ observations."

- Line 22: "One major conclusion from this work is the strong constrain at global scale of the data assimilated compared to the fire prior used." This is vague and unclear. Please rewrite.

We re-write the sentence by :

"A major result of this work, that we can observe at global scale, is the strong constraint and influence of the $CO_2$ assimilated data among the inversions, on the net fluxes. Inversions using OCO-2 (or IS) data have closer emissions each other and so are more influenced by observations, compared to the fire prior used which has minor constraint."

- Line 25: FIREMO => FIREMo **Done**

- Line 26: "it is not the case for the majority of TCCON sites" **Done**

- Line 37: "The first uses, since 2017..." You need to write this differently. It implies that all fire models based on burned area use the modification that only GFED4s uses. First explain there are two methods of emission estimation, i.e. burned area approach and FRP approach, and then make a remark that GFED4s works since 2017 somewhat differently because MODIS burned area algorithm has been updated from Collection 5 to Collection 6. That means GFED4s fluxes are not based anymore from the burned area product directly but on the relationships between climatological GFED4s emissions between 2003–2016 and MODIS active fire detections and its FRP. This explanation should actually be included in the Appendix.

Thank you for this comment, we write line 37 with "The first approach uses burned area products." and we have added line 740, the sentence "The GFED4.1s version have encountered some changes since 2017 because MODIS burned area algorithm has been updated from Collection 5 to Collection 6. Consequently, GFED4s fluxes are not based anymore from the burned area product directly but on the relationship between climatological GFED4s emissions between 2003-2016 and active fire detection and its FRP product."

- Line 44: "Two emission inventories use this approach..." Are you sure these are the only ones available? I would write "Two examples of emission inventories that use this approach are..."

We took this comment in consideration, and we replace the sentence with the reviewer's suggestion.

- Line 46: "emissions inventories" => "emission inventories" **Done**

- Line 51: "trace gases emissions" => "trace gas emissions" **Done**

- Line 53: "...complicates the inference a great deal " Not sure what you are trying to say here.

We changed this sentence with "Moving from global annual fluxes to finer scales in space and time greatly complicates the emission estimation".

- Line 69: were examined **Done**

- Line 72: "Rather, it is assumed that fire emissions have much lower uncertainty (generally believed to be less than 10%)" But at line 51 you claim the errors are orders of magnitude for the emissions. Is the 10% uncertainty referring to fossil fuel emissions perhaps? Please check and correct.

This sentence refers to the previous one line 72 " Most inversion models do not explicitly constrain fire emissions with $CO_2$ observations". Most inversion models do not constrain fire emissions with CO2 observations as they assumed low uncertainty of fire emissions. Which is why we precised line 77 "This inference is problematic, not least due to the aforementioned fire emissions uncertainties in time and space, which could alias into inferred biospheric fluxes at continental or regional scales." It is believed that fire uncertainty are less than 10% however uncertainties can still lead to errors up to an order of magnitude for the total trace gases emissions. This sentence is correct but should be place in the context of other sentences that follow or precede the sentence.

- Line 76: "This inference is problematic" => "This assumption is problematic" **Done**

- Line 76: "emissions inventories" => "emission inventories" **Done**

- Line 92: "was caused not by rising of biomass burning emissions " Please rewrite

We rewrite this by "was not caused by increased biomass burning emissions".

- Line 96: "emissions inventories" => "emission inventories" **Done**

- Line 104: 2016, 2017 and 2018.

It is in fact not correct to say the 2015 El Nino event. The 2015 El Niño started in 2015 and ended in 2016. We changed the sentence with "the 2015-2016 El Niño event", as already mentioned line 186.

- Line 108: "Finally, these updated fire emissions are imposed in an atmospheric CO2 inversion that constrains CO2 fluxes, using either OCO-2 XCO2 retrievals or in situ data, with different assumed fire emissions and appropriately rebalanced prior biogenic fluxes."
  I suggest to write this differently.
  A suggestion:
  "Finally, these updated fire emissions and appropriately rebalanced prior biogenic fluxes are imposed in an atmospheric CO2 inversion to constrain the net land and ocean CO2 fluxes using

either OCO-2 XCO2 retrievals or in situ data. To evaluate these new emissions, an alternative set of fire emissions and rebalanced prior biogenic fluxes have also been used in this CO2 inversion framework."

We thanks the reviewer for this suggestion that we took in consideration.

- Line 113: You can mention the Appendix as well here.

We added the sentence "Description of the different GFED versions are presented in Appendix A."

- Line 229: Are MOPITT CO and OCO-2 CO2 observations also aggregated to 3x2 and 6x2 resolution in the inversions (e.g. in the observation operator)? This needs to be explained clearly in the paper how you treated the observations in regard to the simulation resolutions.

The observations are not aggregated at the model resolution. Fluxes and measured concentrations are linked through the transport and the observation operator. The observation operator samples the model fields at the location and time of the observations. We added in our manuscript line 246 "Fluxes and measured concentrations are linked through the transport and the observation operator. The observations are not aggregated at the model resolution." and line 249 "Due to some information gaps in the observational coverage, there is not enough information for the state vector. Therefore, the prior fluxes are used as the foundation to which we make corrections with information from the observations. These corrections are determined by the relative strengths of the prior uncertainty and the model-data mismatch statistics.".

- Because fire emissions are often very local (say within ~10km) and you use very detailed CO observations at 22km, how can you justify using such a coarse resolution of 3x2 (~300x200km) for transport simulation? You need to elaborate in the methodology or discussion how this can affect your inversion results on CO emissions.

Even if fires are very local, the column observations have a large footprint in space and time. It is hence not possible to work at the fire's local resolution. There is an increasing demand for inversion at higher spatial resolution. However, it calls for new development in the inverse models to reduce the calculation cost and ask a shift from global to regional inversions. The development of regional inversions allows in theory an efficient usage of high-resolution data while preserving a reasonable computational cost. Additionally, in comparison to numerical weather prediction model, global tracer transport models have generally used lower resolution due to the inclusion of chemistry and long

window data assimilation which cannot afford such a computational expense at higher resolution. The length scale of emissions is much shorter than the length scale of its atmospheric signature, especially in the column. Consequently, this large length scale needs to be considered for the inversion. Additionally, the observation operator must be stored as a large Jacobian matrix before the computation necessary for the inversion. For the inversion, the computation of the observation operator is consequently problematic at higher resolution when assimilating large time series of satellite observations. In order to produce our CO or CO2 fluxes in atmospheric inversion systems, the observations are used in the 4D-Var (data assimilation system) based on the tracer transport model TM5. It is important also to consider that the transport error grows significantly as we refine our model resolution. We added this paragraph in the manuscript line 246 "If TM5 cannot represent the synoptic variability accurately, then the resulting errors when comparing the model with observations will prevent these observations from being used effectively in the 4D-Var. The mismatch between the model and the observation due to the differences in the resolution of the tracer transport model (including both the resolution of the meteorological ERA-Int fields and the resolution of the fluxes on the model grid) and the resolution of the observation footprint is also known as representativeness error (observational error). If the observational error in data assimilation is not correctly accounted, there will be errors in the optimized parameters (surface fluxes). For more information on the calculation of observational error in TM5, see Bergamaschi et al., (2010). However, it has been shown in previous studies that going from coarse resolution of the global tracer transport models to higher resolution does not provide improvement with respect to observations (Lin et al., 2018, Remaud et al., 2018)."

- I can imagine that the simulation of pyroconvective plumes and the transport of CO high in the troposphere and stratosphere might be very challenging at very coarse resolutions. Is captured adequately, and if not how would it affect your optimized fire emissions?

Since CTMs have at global scale coarse resolution, they could not capture the vertical transport induced by the heat released from landscapes scale fires or the pyronconvective plumes. Therefore, the plume dynamics and its associated vertical transport are based on parameterizations called "Injection Height" (Colarco et al., 2004). This "Injection Height" represents the height and vertical layer of the CTM where smoke emissions are no longer influenced by the plume dynamics but simply released in the atmosphere. As already mentioned in our paper line 252 *"Injection heights, in the CO inversion, are computed using IS4FIRES (Integrated System for Wild-Land Fires, http://is4fires.fmi.fi/, Sofiev et al. (2013)). This emission database is driven by re-analysis FRP obtained from MODIS (Giglio et al.,*

*2006)) instrument on board Aqua and Terra satellites.*". The injection height already used in our optimized fire emissions and mentioned in our manuscript, we did not write additional information in the manuscript.

- I assume IS data is sampled inside TM5 at the correct elevation and location, but please explain on how you obtain a representative sample of CO2 when you simulate at a very coarse resolution of 6x4 (~600x400km). This needs to be discussed in the manuscript as well.

For the IS inversion, TM5 is sampled at grid cells containing in situ observation, at the hours corresponding to the observations, and at appropriate vertical level and horizontal location. The model profile is interpolated to a level corresponding, on average, to the altitude above sea level of the observation site. Information on the observational error has been previously provided.

- Line 225: "CO inversion :" remove space before :

We modified that line 255.

- Lines 270-278: I think is more appropriate here to explain how GFED4s has been modified from 2017 onward.

Since this information is already available in the Appendix A, we mentioned "Further description in GFED versions can be found in Appendix A" and we added in the appendix

"The active fire data comes from Tropical Rainfall Measuring Mission (TRMM), the Visible and Infrared Scanner (VIS), and the Along-Track Scanning Radiometer (ATSR), three other instruments on board with MODIS. MODIS has a 500 m horizontal resolution. ".

- Line 279-285. Is a length scale of 1000km not too long for local fire events? Please explain. Would a smaller length scale affected your emission results? The Meirink reference is not about fire emissions but about methane emissions. So why is this a valid choice?

We have used a horizontal length scale of 1000km which has been determined based on previous data assimilation and comparison of emission inventories. This choice is made in order to represent the best realistic spread from errors in emissions. We used Meirink et al., (2008) as reference for choosing a length scale of 1000km because the TM5 CO is based on the methane development described in Meirink et al., (2008). The CO inversions described in Hooghiemstra et al., 2012 or Krol et al., (2013) were based on the 4DVAR version of the TM5 model developed and described by Krol et al., (2005,

2008); and Meirink et al., (2008). More recently, the CO inversions developed in Nechita-Banda et al., (2018) were also based on the TM5 model of Krol et al., (2015, 2008) and Meirink et al., (2008). All these previous studies used a horizontal correlation length scale of 1000km for the global prior emissions, length justified in Meirink et al., (2008).

Additionally, a length scale of 1000km is not the largest horizontal scale used for CO inversions. Larger spatial correlation lengths were previously used. For instance, Barre et al., (2015) used 2000km with the CESM model. All CO inversions with TM5 used a length of 1000km while for LMDz they used 500km (Zheng et al., 2019, Chevalier et al., 2009).

- Line 284: "The errors are assumed ..." This sentence needs to start differently because you switch topic from matrix B to matrix R.
  suggestion:
  In the observation covariance matrix R we only assume uncorrelated errors, meaning we only have errors along the diagonal.
  Also good to explain why or why not this is a realistic assumption.

Thank you for this suggestion that we took in consideration. The errors are assumed uncorrelated allowing the use of a diagonal matrix for the observational error covariance matrix R. This can be assumed since the observation error can be in general easily quantifiable by careful calibration of the instruments.

We added line 284: "In the observation covariance matrix R, we only assume uncorrelated errors, meaning we only have errors along the diagonal. This can be assumed since observation error is in general easily quantifiable by careful calibration of instruments."

- Line 289: "For each pixel..." The pixel size is 3x2 right? Good to add that here. "For each pixel (3x2 degree resolution)" Done

- Figure 1: Because biofuel was added to all CO2 FIRE estimates, should biofuel burning not be included somewhere in the schematic overview?

Indeed, biofuel emissions was added to all CO2 FIRE estimates, including FIREMo. We hence added Biofuel in the schematic overview of Fig. 1.

- Line 297: g.mol−1 I don't think you need to type the dot between g and mol. Same applies for other units elsewhere in the text. **Done**

- Line 307: Add here that you also included biofuel emissions to obtain FIREMo.

We changed the sentence with "Finally, we sum up these emissions across all surface types and also include $CO_2$ biofuel emissions (see table 3) in order to get monthly total optimized prior $CO_2$ biomass burning emissions that we called "FIREMo" (see Fig. 1). ".

- Line 322: The Ott reference links to the dataset of GEOS-Carb CASA-GFED3, but it does not clarify the methodology of respiration modification. Is there another reference available?

There is no other reference available for the GEOS-Carb CASA-GFED3 product than the one we already provided. For more details on the product, it is advised to contact the PI, in this case Lesley Ott. We specified in our manuscript that we took a similar approach than the one use in GEOS-Carb CASA-GFED3, meaning "resembling without being identical". In GEOS-Carb CASA-GFED3 they use monthly fire emission in their estimation of the net $CO_2$ fluxes, while most of other inversions have chosen to only report net land fluxes by holding fire emissions fixed. This is already mentioned in the manuscript both in the introduction and in the methodology (line 335).

To avoid confusion, we decided to rewrite the following sentence "*Terrestrial biosphere fluxes and fire emissions are difficult to disentangle from $CO_2$ data alone, and some inverse modeling studies (e.g. Crowell et al. (2019)) choose instead to report the net land fluxes. Likewise, some global land flux estimates such as GEOS-Carb CASA-GFED3 project (Ott, 2020) use fire estimates to revise the terrestrial biosphere flux estimates through modification of ecosystem respiration. We take a similar approach, starting with the gross primary production and respiration estimates from the CASA-GFED3 3-hourly 0.5∘×0.625∘ (Ott, 2020). *" by "Terrestrial biosphere fluxes and fire emissions are difficult to disentangle from $CO_2$ data alone, and some inverse modeling studies (e.g. Crowell et al. (2019)) choose instead to report the net land fluxes. Likewise, some global land flux estimates such as GEOS-Carb CASA-GFED3 project (Ott, 2020) use fire estimates with ecosystem respiration to revise the terrestrial biosphere flux estimates. We take a similar (but not identical) approach, using emissions of fire and respiration to estimate the terrestrial biosphere flux. We start with the gross primary production and respiration estimates from the CASA-GFED3 3-hourly 0.5∘×0.625∘ (Ott, 2020). ".

- Line 331: which is in balance with **Done**

- Table 3. If I take for 2017 GFED4s emissions (as an example), and add up all the other flux components (biofuel, ocean, NEEre4, and fossil), the sum is not equal the AGR_noaa. Is there something I'm missing in this logic? 4.9 = 0.486 + 10.07 + 1.78 - 4.83 - 2.6

  However, AGR_noaa for 2017 is 6.06. Please explain.

We thanks the reviewer for this comment and question. It is an error from our part. The AGR$_{NOAA}$ mentioned in the table are the corresponding values from 2014 through 2017 and not 2015 through 2018. Consequently, the 6.06 PgC/yr is in fact corresponding to 2016. We have corrected that in the table 3. Additionally, we realized that the NEE values were not the prior values and so were not correct. Since this table only include prior information, we corrected the table with only prior values (for verification, we can see that the new values are well corresponding to Fig. 5). For further information in the AGR calculated with the re-balanced respiration, we have added their respective values in the table. As we can observe, the AGR calculated are close to the AGR$_{NOAA}$. We have added the new values of the following table in Table 3 of our manuscript.

| | 2015 | 2016 | 2017 | 2018 |
|---|---|---|---|---|
| OCEAN | -2.6 | -2.6 | -2.6 | -2.6 |
| BIOFUEL | 0.479 | 0.476 | 0.486 | 0.486 |
| FF | 9.89 | 9.91 | 10.07 | 10.28 |
| GFED3 | 2.03 | 1.63 | 1.97 | 1.97 |
| GFED4 | 2.09 | 1.73 | 1.78 | 1.69 |
| FIRE3 | 2.51 | 2.11 | 2.46 | 2.46 |
| FIRE4 | 2.57 | 2.21 | 2.27 | 2.18 |
| FIREMo | 1.82 | 1.47 | 1.58 | 1.56 |
| NEECMS | -1.93 | -1.71 | -1.58 | -1.55 |
| NEEre3 | -3.42 | -3.41 | -5.40 | -5.10 |
| NEEre4 | -3.40 | -3.50 | -5.11 | -4.73 |
| NEEreMo | -2.43 | -2.51 | -4.25 | -3.90 |
| | | | | |
| AGR$_{CMS}$ | 7.87 | 7.71 | 8.35 | 8.59 |
| AGR$_3$ | 6.38 | 6.01 | 4.53 | 5.04 |
| AGR$_4$ | 6.46 | 6.02 | 4.63 | 5.13 |
| AGR$_{Mo}$ | 6.68 | 6.27 | 4.8 | 5.34 |
| | | | | |
| AGR$_{NOAA}$ | 6.3 | 6.06 | 4.54 | 5.05 |

- Figure 2 and Table 4: Suggestion: It makes more sense I think when you name OCO-2 inversions differently. Instead of CMS-GFED3, GFED3re, GFED4re and MOre, you could name them OCOCMS, OCO3re, OCO4re and OCOMOre. That way they are more akin to the IS inversion names and more easily recognisable as OCO-2 inversions.

Thank you for this suggestion, we changed the manuscript using these names.

- In the final column of boxes with inversion names you could give them the same color as used for the bar graphs (e.g. Fig. 5). So, black for CMS, blue for GFED4re, green GFED3re and red for More.

Thank you for this suggestion which we used.

- Line 367: Typo Coparison **Done**

- Line 376: "involves" => "produces" or "yields" **Done**

- Line 387: "south America" => "South America" **Done**

- Line 390: "are strong" => "are large" **Done**

- Line 390: "are needed" => "is needed" **Done**

- Line 397: "seem to observe as mush as the prior GFED4.1s." I would phrase it differently: "a characteristic that is shared between the MOPITT constrained fire emissions and GFED4s"

We considered this suggestion

- Figure 4:
- Include in the legend the colored fire emission categories **Done**
- Please provide in the final version of the manuscript a higher resolution of the figure image. The current image is somewhat blurry and hard to read. **Done**
Place 'year' tick markers and labels in the middle of each grouping of bars **Done**
In caption remove space in front of " :" **Done**

- Line 401: "Previous studies have shown that peat area and depth, producing large amount of carbon ( 0.60 PgC/yr which represents 26% of the total carbon fire emissions, Nechita-Banda et al. (2018)), were found to have significant uncertainties in Indonesia in the emissions inventories"

  Suggestion:

  "Previous studies have shown that the parameterization of peat (surface area and layer thickness) resulted in significant uncertainties in emission inventories. This is especially true for Indonesia where combustion of peat can produce significant amount of carbon."

We considered this suggestion to write this new sentences: "Previous studies have shown that the parameterization of peat (surface area and layer thickness) resulted in significant uncertainties in emission inventories. This is especially true for Indonesia (Lohberger et al., 2017; Hooijer and Vernimmen, 2013) where combustion of peat can produce significant amount of carbon (Nechita-Banda et al., (2018)).".

- Line 403: "emissions inventories" Done

- Line 404: "Our posterior have lower emissions than the prior for..." => "Our posterior fire emissions are lower than the prior fire emissions" Done

- Line 404: "southern tropical Asia" => "Southern Tropical Asia" Done

- Line 405: "However, Nechita-banda et al. (2018) assimilated MOPITT and NOAA observations and used GFAS as fire priors, an inversions set-up different to what we used."
  "However, Nechita-banda et al. (2018) assimilated MOPITT and NOAA observations and used GFAS as prior for fire emissions. Also, their inversion set-up was different to what we used."

Done

- Line 406: "Additionally, no evaluation against independent data have been performed in their study to determine if their results are more trustworthy than our results. "
  "Additionally, no evaluation against independent data have been performed in their study, so there is no reason to believe their results are more trustworthy than ours" Done

- Line 410: "that GFED4.1s have a fire peak earlier than MOPITT" => "that GFED4.1s fire emissions have a fire peak earlier than MOPITT constrained emissions" Done

- Line 422: "for the prior4" => "for prior4" **Done**

- Line 425: "decreasing from 2015 through 2018." The net sink is actually increasing.

Indeed, the net sink is increasing from 2015 through 2018. We changed it.

- Figure 5

Place tick markers and labels in the middle of each grouping of bars. **Done**

"Annual prior CO2 emissions, " add units => "Annual prior CO2 emissions (PgC/yr)" **Done**

"are represented" => "are shown" **Done**

"between GFED4.1s (blue), GFED3 (green), MOPITTopt (red) and CMS (black)."

Shouldn't these not be called either GFED4re, GFED3re, MOre, and CMS-GFED3 according to Table 4, or prior4, prior3, priorMo and priorCMS? Indeed, we corrected the notations.

- Line 432: "FIRE4 and FIREMo " labeling is still somewhat confusing. Is it not better to call them prior4 and prior3?

We decided not to follow this decision. The notation prior4 in this case represents the net fluxes and so both FIRE4 and NEEre4.

- Line 437: "consistent with the fact that GFFED4.1s CO was observing higher CO fire emissions than MOPITT " I don't see the added value of this part of the sentence. The first part already makes that point.

We removed this part.

- Line 439: "observed by " => "estimated by" **Done**

- Line 441: "Northern" **Done**

- Figure 6: "North" **Done**

- Line 449 and elsewhere: "GFED" write it in full "GFED4.1s".

GFED in this paragraph corresponds to both GFED3 and GFED4.1s inventories (or FIRE3 and FIRE4).

We corrected this paragraph with:

*"For savanna, agriculture and peat lands, FIREMo observed a peak in fire seasonality after the peaks observed with  both FIRE3 and FIRE4 (Fig.  S8). This is particularly true for the 2015 El Nino fires but less for the fires that occurred in 2017 and 2018. In this period, FIREMo does not observed as much fire emissions as  FIRE3 and FIRE4 with a similar seasonality. The difference in seasonality for 2015 could be a result of the large and intense fires during the El Nino event burning larger regions and releasing more smoke which could have impacted the MODIS burned area data used in  GFED3 and GFED4.1s inventories but probably also the MOPITT retrievals. Further investigation are then needed to study this region."*

- Line 451: But how sure are we that the MOPITT constrained emission seasonality is "better" (as you write at line 445) if smoke could also impacted the MOPITT observations? Is there evidence in TCCON data? Please elaborate in the paper.

The paragraph referred here is "*As already observed with the CO emissions (Fig. S6) and discussed in van der Laan-Luijkx et al. (2015) and Nechita-banda et al. (2018), the seasonality of fires over tropical Asia is better captured with MOPITT than with the emission inventories for peat lands. However, this is not only true for peat but also for other vegetation types. For savanna, agriculture and peat lands, FIREMo observed a peak in fire seasonality after the peaks observed with GFED (Fig. S9). This is particularly true for the 2015 El Niño fires but less for the fires that occurred in 2017 and 2018. In this period, FIREMo does not observed as much fire emissions as GFED with a similar seasonality. The difference in seasonality for 2015 could be a result of the large and intense fires during the El Niño event burning larger regions and releasing more smoke which could have impacted the MODIS burned area data used in GFED but probably also the MOPITT retrievals. Further investigation are then needed to study this region.*" This was developed later in the manuscript line 644 "*As discussed in Nechita-Banda et al. (2018) and van der Laan-Luijkx et al. (2015) for Equatorial Asia and tropical south America, GFED4 does not capture fire seasonality due to the use of burned area, compared to GFAS. In both GFED and GFAS method (and similarly for MOPITT), the detection of fires underneath clouds and below the canopy is difficult. But, FIREMo emissions, compared to FIRE3 and FIRE4, has the advantage of combining optimized fire emissions with local observations.*". Assumption of the seasonality difference between GFED and MOPITT observed with previous studies was mentioned line 410 "*van der Laan-Luijkx et al. (2015) and Nechita-banda et al. (2018) hypothesized that GFED4.1s might not capture the timing of emissions over area with peat fires due to the use of burned area, which may be more sensitive to the initial stages of the fire than to the continued burning.*". So the difference of seasonality is more linked to the different phase of burning than the smoke biases. However, it is

important to also acknowledge the existence of data gaps in both MODIS and MOPITT due to clouds and smokes. For better clarity between all of these elements, we rewrote paragraph line 445 "As already observed with the CO emissions (Fig. S5) and discussed in van der Laan-Luijkx et al. (2015) and Nechita-banda et al. (2018), the seasonality of fires over Tropical Asia seems to be better captured with MOPITT than with the CO emission inventories for peat lands. However, this is not only true for peat but also for other vegetation types and can also be observed for CO2 emissions. For savanna, agriculture and peat lands, FIREMo has a peak in fire seasonality after the peaks observed with both FIRE3 and FIRE4 (Fig. S9). This is particularly true for the 2015 El Nino fires but less for the fires that occurred in 2017 and 2018. In this period, FIREMo does not observe as much fire emissions as FIRE3 and FIRE4 with a similar seasonality. The large difference in seasonality for 2015 could be particularly marked due to the large and intense fires of the El Nino event burning larger regions and releasing more smoke. However, it is important to acknowledge the existence of data gaps due to clouds and smokes in both MODIS burned area products (used in GFED3 and GFED4.1s inventories) and probably MOPITT retrievals. Further investigations are then needed to study this region.".

- Line 452: "emissions : " again remove space after : Done

- Line 452: For simplicity sake, I suggest to title section 3.2.2 in a similar fashion as 3.2.1: e.g. 3.2.2 Posterior NEE and fire CO2 fluxes

We considered this suggestion and changed the title of 3.2.2.

- Line 462: See previous comment about Figure 2. I think it is much clearer if you name GFED3-CMS as OCOCMS. The distinction between OCO-2 and IS inversions is much easier that way.

We apply this change in all the manuscript and in the figures.

- Line 464 and elsewhere: "different priors " better to refer to "different inversions". Now the use of "prior" labels in this particular section is confusing because we are actually looking at "posterior" fluxes. Please change this throughout the section.

Done for the sentence where priors should have been inversions. However, in some other sentences, priors was correctly used.

- Line 466: "across the priors" => "across the inversions" Done

- Line 470: "across the priors" => "across the inversions" **Done**

- Line 470: "adjusted downward" meaning larger sink? If so, be clear in the text.

We changed by "However, the 2016 sink is larger for the OCO-2 fluxes (between -0.4 PgC/yr and -0.6 PgC/yr) than the in situ fluxes".

- Line 473: "than the other regions " => "than in the other regions " **Done**

- Line 475: "but we know that there are a few in situ data present in 475 the SH Ext and so they have a different data constraint " => "but we know that there are a few in situ sites present in the SH Ext resulting in a limited constrain on emissions as well" **Done**

- Line 476: "across the priors" => "across the inversions" **Done**

- Line 478: "sinks with in situ NEE emissions." => "sinks with in situ observations." **Done**

- Line 479: "across the priors" => "across the inversions" **Done**

- Line 481: "The net fluxes of ISMOre and IS4re look similar for the Tropical regions, while the net fluxes of IS3re and ISCMS look alike, suggesting the sensitivity in these regions to the fire prior, not only for IS but also for OCO-2 data constraint." This sentence needs to be rewritten because it is hard to follow.

We rewrote this sentence with "For the Tropical regions, ISMOre and IS4re net fluxes look similar. Similarly, the inversions constrained with FIRE3 look alike such as IS3re and ISCMS. This suggests the sensitivity of inversions to the fire prior in these regions.".

- Line 485: "we can see difference in the carbon balance" => "we can see a number of differences in the inferred carbon balance" **Done**

- Line 486: I don't agree with this sentence. If I look at Fig. 7, IS data gives the largest net sink for NH ext.

Indeed, the sentence *"If we first look the Northern Extra-Tropical regions (North America, North Asia and Europe), we can see that the OCO-2 inversions have deeper net sinks than IS (see Fig. 8)."* has

been modified to "If we first look the Northern Extra-Tropical regions, we can see that the IS inversions have deeper net sinks than OCO-2 (see Fig. 7).".

- Line 490: "with few data (north Asia regions)." => "where data is sparse (North Asia regions)." Done

- Line 492: "It is interesting also to see balance " => "It is also interesting to see the balance " Done

- Line 493: "sink decreased " => "sink reduction " Done

- Line 496: "uptake of around " => "with an uptake of around " uptake where? In Europe?

We changed the sentence with "Reuter et al., 2014 found, using GOSAT data, a similar mass balance between Europe and Northern Africa with an uptake of around 1 PgC/yr in Europe".

- Line 497: Would CarbonTracker Europe not provide a detailed European estimate?

https://carbontracker.eu

CarbonTracker Europe provides CO2 emissions estimation for the Transcom regions combining measurements (such as the ObsPack products) and modeling system. They used a 1x1 degree zoom of the TM5 transport model and 20 sites over Europe. For the prior biosphere, they used SiBCASA-GFED and for the prior fire they used GFED4. They mentioned also on their website that the biological fluxes have not been verified by observations. The uncertainties associated with these fluxes is not known. While recent estimation is available online, there is no publication providing CO2 emission estimate over Europe with CarbonTracker Europe covering 2015-2018 and associated validation, and so there is a lack of carbon budget information over Europe for our period of interest.

CarbonTracker Europe net flux (calculated by extracting the fossil fuel emissions from total flux in their table https://carbontracker.eu/fluxtimeseries.php#imagetable) gives:

| | 2015 | 2016 | 2017 | 2018 |
|---|---|---|---|---|
| CarbonTracker Europe net fluxes | -0.1 | -0.22 | -0.51 | -0.15 |

Except for 2017, the OCO-2 and IS inversions have different net fluxes than with CarbonTracker Europe.

- Line 501: "in 2017, as opposed to 2016 " refer to figure 6.

We added "(see Fig. 6 and 8)".

- Line 503: Again somewhat confusing sentence: "MOre and GFED4re inversions (respectively ISMOre and IS4re) are similar, while GFED3re and CMS-GFED3 (respectively IS3re and ISCMS) are more similar."
  Why not:
  "ISMOre and IS4re inversions providing similar results (both based on either optimized GFED4.1s and default GFED4.1s emissions), while the same is true for IS3re and ISCMS inversions (both based on GFED3 emissions)."

Thank you for this suggestion. We changed the sentence by: "ISMOre and IS4re inversions provide similar results (both based on either optimized GFED4.1s and default GFED4.1s emissions), while the same is true for IS3re and ISCMS inversions (both based on GFED3 emissions). Same is true for OCO-2 inversions as well where OCOMOre and OCO4re have similar results while OCO3re and OCOCMS are similar.".

- Line 508: "for each priors" you mean "for each inversion"?

This has been already changed with a previous comment.

- Line 512: "observation" => "observations" Done

- Line 513: "southern and northern hemispheres " => "Southern and Northern Hemispheres" Done

- Line 516: "than the IS are " => "than the IS posterior emissions are" Done

- Line 519: "then come from different area of observation " I don't understand this.

The sentence " This difference in posterior flux could then come from different area of observation" refers to both lines 516 and 517. The posterior emissions from OCO-2 being closer to the prior is explained by the few amount of OCO-2 measurements above the moist Amazon, a area/region affected

by cloud. We are realizing that the sentence line 519 does not bring additional information. We hence decided to remove it.

- Line 522: "OCO-2 net fluxes " OCO-2 doesn't measure net fluxes. "net fluxes derived with OCO-2" Thanks for this correction. We changed the sentence.

- Line 522: "high sources " => "large sources" Done

- Line 523: "FIREMo and FIRE4 sinks decrease" Again, these labels don't fit well in the sentence. According to the definition of FIREMo and FIRE4 these are fire emission + biofuel burning. They cannot be sinks.
  If you write it like this it is easier to understand:
  Posterior net sinks derived with FIREMo and FIRE4 emissions decrease for 2017, however, the posterior net sinks derived with FIRE3 do not.
  Similar issues also appear elsewhere in your text. Please correct this.

Thank you for this suggestion, we changed the sentence and corrected the rest of the manuscript accordingly.

- Line 525: "dependence " => "known prior dependency of the IS posterior emissions" Done

- Line 526: Again "southern" => "Southern " Done

- Line 527: Where do we see this? Respiration is not plotted

We can see indirectly the respiration through the NEE emissions. Respiration has been balanced with fires, and this same respiration has been used to calculate NEE emissions. We changed this sentence *"For southern Tropical Africa, we can see the large balance between the fires and the respiration, which are anti-correlated in their variability."* by "For Southern Tropical Africa, we can see the large balance between the fires and the NEE emissions (indirectly the balance between the fires and the respiration), which are anti-correlated in their variability.".

- Line 526-531: This section is in particular hard to follow. Need to be rewritten.

We rewrote the sentence *"For Southern Tropical Africa, we can see the large balance between the fires and the respiration, which are anti-correlated in their variability. GFED4re and MOre have larger sources than GFED3re and CMS-GFED3. With the IS, there is large variation across the inversions where IS4re and ISMOre both constrain a source of carbon for the whole period, while ISCMS and IS3re have smaller source of carbon and even a sink in 2016 and 2017. These differences seem to suggest that both fires and respiration are especially important when observational coverage is limited."* with "For Southern Tropical Africa, we can see the large balance between the fires and the NEE emissions (indirectly the balance between the fires and the respiration), which are anti-correlated in their variability. Additionally, OCO-2 inversions derived with FIREMo and FIRE4 emissions (OCOMORe and OCOG4re) have larger sources than inversions derived with FIRE3 (OCOCMS and OCOG3re). With the IS inversions, there is large variation across the inversions where IS4re and ISMOre both constrain a source of carbon for the whole period, while ISCMS and IS3re have smaller source of carbon and even a sink in 2016 and 2017. These differences between inversions derived with FIREMo or FIRE4 and FIRE3 seem to suggest that fires (and so NEE re-balanced with fires) are especially important when observational coverage is limited.".

- Line 532: "between the priors" but these are posterior emissions. You mean perhaps "between the inversions"

Indeed, this has been changed

- Line 532-535: Also this part is hard to follow.

We rewrote the sentence *"Northern Tropical Asia (Fig. 8) shows agreements between the priors and OCO-2 data constraints for 2015 and 2016, but shows significant differences between OCO-2 and IS for 2017 and 2018. The sparse coverage of in-situ data over this region could explain the difference with OCO-2, though not specifically for 2017 alone, and hence further investigations are needed for this region."* with "Northern Tropical Asia (Fig. 8) shows agreements between OCO-2 and IS inversions, but shows significant differences in 2016. The sparse coverage of in-situ data over this region could explain the difference with OCO-2, but not specifically for 2016 alone, and hence further investigations are needed for this region.".

- Line 535: "Finally, for southern Tropical Asia, the inversions adjusted the NEE sinks for MOre and GFED4re to be larger than the two other inversions in order to accommodate the smaller fires observed with FIREMo and FIRE4. "

  Can be simplified to

  "Finally, for Southern Tropical Asia, a larger sink was derived with OCOMOre and OCO4re than with OCO3re and OCOCMS, to balance the smaller fires derived with FIREMo and FIRE4."

  Still I don't understand what you trying to say here. Would you not expect a larger sink when fire emissions are larger?

Indeed, we expect a larger sink when fire emissions are larger, which it is observed and was not correctly mentioned in the sentence. We rewrote the sentence with: "Finally, for Southern Tropical Asia, a smaller sink was derived with OCOMOre and OCO4re than with OCO3re and OCOCMS, to balance the smaller fires derived with FIREMo and FIRE4.".

- Line 485-546: This results section needs some additional editing as some parts are still hard to follow. The final summary paragraph was very informative. I suggest to use this as the main anchor for 3.2.2b. Each sentence of the summary text (L541-L546) can be used as a beginning for a paragraph in 3.2.2.b.

We changed the following sub-section 3.2.2.b:

[revised manuscript text omitted]

- Line 549: "shows a latitudinal gradients " => "show latitudinal gradients " Done

- Line 550: "northern hemisphere " => "Northern Hemisphere " check this throughout the manuscript Done

- Line 550: "High land values " => "High concentrations over land" Done

- Line 551: "north tropical south America " => "North Tropical South America " Done

- Line 554: "the OCO-2 retrievals (IS data) " => "the OCO-2 retrievals and IS data "

We changed with "with either the OCO-2 retrievals or IS data".

- Line 559: "This result suggests that the inversion does not change much from the prior, but this result can be explain due to the small number of observations available in these regions. " Please rewrite this.

We rewrote this: "For the IS inversions, the differences between priors and posteriors with the IS data are very similar, suggesting that the inversion does not change much from the prior. The small number of observations available in these regions could explain this result.".

- Line 561: "For the comparison among the simulations, there is no large difference between the different simulations and the data, particularly for the optimized CO2 measurements. "
  suggestion:
  "Among the different simulations, in particular, the posterior concentrations vary little in comparison to OCO-2 and IS data." **Done**

- Line 586: remove "matches" **Done**

- Line 587: "priorMO bias are slightly lower or smaller than prior4 " => "priorMO biases are slightly smaller than prior4" **Done**

- Line 590: "reduced through the inversion " => "reduced by the inversion " **Done**

- Line 593: "ISMOre can be better at some tropical sites than the other simulation " => "ISMOre provides a better match at some tropical sites than the other simulations" **Done**

- Line 595: "all standard deviation " => "all standard deviations " **Done**

- Line 597: "across the priors" => "across the inversions" **Done**

- Line 609: "with each fire prior" => "with each fire emission estimate" **Done**

- Line 612: "the recovery period which followed it " => "the recovery period that followed "
  Done

- Line 613-616: Sentence is too long and difficult to understand. Rephrase please.

We change the sentence "*Globally, and for most regions, we find that the dependence of the inversion results on prior emissions is of secondary importance when compared with the data constraint, in the sense that variations in posterior flux are much smaller across different prior mean fluxes (and the different uncertainties that come from scaling the prior mean flux) as compared with differences resulting from assimilating OCO-2 versus in situ data.*" by "Globally and for most regions, we find that the inversion results have a greater dependence on data constraint than on prior emissions. The variations in posterior flux are much smaller across different prior fluxes (and the different uncertainties that come from scaling the prior flux) as compared with differences resulting from assimilating OCO-2 versus in situ data.".

- Line 623: "on a short time basis " => "short-term" Done

- Line 626: "over peat fires " => "of peat fires" Done

- Line 636: "have been found higher than with GFED4 " => "have been found higher in GFED3 than with GFED4.1s" Done

- Line 638: 'FIREMo emissions are stronger than with FIRE4 emissions but lower than FIRE3 emissions " => "net emissions derived with FIREMo are stronger than net emissions derived with FIRE4 but lower than net emissions derived with FIRE3 "

This sentence correctly refers to fire emissions and not net emissions. We did not change line 638 consequently.

- Line 642: "IS4re and ISMOre have higher net sources of carbons " = > "IS4re and ISMOre provide higher net carbon sources" Done

- Line 649: "and IS observations But" place period at end of sentence. Done

- Line 652: "the full dynamics" = >"the full dynamic" Done

- Line 653: "changing the combustion efficiency and then the gases emitted " => "changing the combustion efficiency and the ratio CO and CO2 are emitted "

We changed the sentence by "changing the combustion efficiency (Zheng et al., 2018b) and hence the CO/CO$_2$ emission ratio.".

- Line 655: "Further works is " => "More work is " Done

- Line 656: "A recent study has shown the underestimation for Africa of MODIS burned area and consequently GFED4s, compared to the new Sentinel-2 burned area product (Ramo et al., 2021). " =>
  "A recent study has shown that MODIS product most likely underestimate burned area for Africa (Ramo et al., 2021)" Done

- Line 658: "observed with previous studies " => "The higher fire emissions estimated in previous studies " Done

- Line 659: "seems to suggest for future work to carefully choose the CO fire prior used in a CO-CO2 study " Please rephrase

We changed the sentence *"The higher fire posterior emissions observed with previous studies using GFAS as a prior compared to GFED4 (Nechita-banda et al., 2018) and the results of Ramo et al. (2021) seems to suggest for future work to carefully choose the CO fire prior used in a CO-CO2 study.* " by "The higher fire posterior emissions observed with previous studies using GFAS as a prior compared to GFED4 (Nechita-banda et al., 2018) and the results of Ramo et al. (2021) seems to suggest for future work to carefully choose the CO fire prior for the inversions.".

- Line 702: "even if with a difference " Please rephrase.

We decided to remove this part as it does not bring additional information. We changed the sentence *"We found that OCO-2 and in situ net fluxes have, even if with a difference, a better agreement at global scale as observations are dense enough to constrain the fluxes than at latitudinal and regional scale.*" to "We found that OCO-2 and in situ net fluxes have a better agreement at global scale as observations are dense enough to constrain the fluxes than at latitudinal and regional scale.".

- Line 705: "Discrepancies between in situ and OCO-2 inversions occurred over Northern Tropical Africa where OCO-2 inversions have shown net sources while in situ inversions have shown sinks "

  => "Discrepancies occurred over Northern Tropical Africa where OCO-2 inversions derived net sources while in situ inversions derived sinks " **Done**

- Line 706: "However, over Southern Tropical regions, discrepancies appear between the different set of priors, with higher net sources observed with the inversion using the CO/CO2 emission ratio (MOre inversion) for OCO-2 inversion over Southern Tropical South America and with IS inversion over Southern Tropical Asia, compared to the IS inversions using GFED3 fires. "

  => "However, over Southern Tropical regions, discrepancies appear between the different priors, with larger net sources derived with the OCO-2 inversion using the optimized fire emissions (MOre) over Southern Tropical South America and with IS inversion over Southern Tropical Asia." **Done**

- Line 709: "the constrain of priors seems" => "the priors seem" **Done**

- Line 710: "seems to be better representative of the" => ""seems to be better represent the"

We changed with "seems to better represent the".

- Lines 715-718. This section needs to be rephrased.

The following section "*However, biases for the posterior simulated mixing ratio are in the same order. Evaluation mainly showed that biases have been decreased and variability matches better those of TCCON for the re-balanced posterior simulated mixing ratio suggesting the importance of the accuracy in fire priors and the re-balanced of terrestrial emission with fires for $CO_2$ posteriors emissions. The added value of fire emission for NEE optimization is not apparent. Our results seem hence to be very insensitive to optimized fire emissions.*" has been changed to "Evaluation against TCCON shows smaller biases for all the re-balanced posterior simulated mixing ratios in comparison to the CMS posterior simulated mixing ratio. Additionally, variability of all the re-balanced mixing ratio better matches those of TCCON. This suggests the importance of the accuracy in fire priors and the re-balanced of terrestrial emission with fires, for the estimation of $CO_2$ posteriors emissions. However, the

added value of CO fire emissions for NEE optimization is not significant in term of biases reduction on average.".

- Line 719: "We illustrated the potential of using CO/CO2 emission ratio, and the re-balanced respiration and NEE with fire and growth rate, in CO2 inversion for better constraint and accuracy in the CO2 fire prior emissions and biospheric emission estimates. " This sentence needs to be rephrased.

We rephrased this sentence as followed: "We illustrated the potential of using CO/CO$_2$ emission ratio, and the re-balanced respiration with fire, in order to match the atmospheric growth rate, in CO$_2$ inversions. This was performed for better constraint and accuracy in the CO$_2$ fire prior emissions and biospheric emission estimates.".

- Line 725: "from the future " => "from the upcoming" **Done**

- Line 742: Use PgC/yr, as you used throughout the paper. **Done**

- Supplement document: Fig. S5 is missing. Please renumber the Figs from S1 to S9 **Done**

We want to remind the reviewer that the difference between NBE and NEE has been already mentioned in our introduction. NBE being the non-fossil land fluxes while NEE being assumed to be the residual between the posterior total net land flux and the assumed fire and fossil fuel emissions. Through the

rest of the manuscript, we only employed NEE as this is the estimation used in our study. The acronym BB (for biomass burning) not being used in our manuscript, we preferred not to use it, and we kept the annotations CO fire and $CO_2$ fire emissions. However, as mentioned by reviewer #1, we make sure the labels FIREMO or FIRE4 were correctly used to define $CO_2$ fire emissions and not net fluxes, which was not correctly wrote for some sentences. We have already corrected this.

Specific comments (revised manuscript):

L16: "Tropical flux estimates" - Which flux? NEE or NBE? We changed it by "Tropical net flux estimates". As a remind, net fluxes are the sum of fire and NEE (Net Ecosystem Exchange) fluxes.

L20: "TCCON data shows lower biases" - biases against model CO or CO2? Since the rest of the sentence refers to the three re-balanced priors and the one using CASA-GFED3, this evaluation is referring to the $CO_2$ TCCON data. For clarity, we modified by "[…] $CO_2$ TCCON data shows lower biases [...]".

L22-23: "One major conclusion from this work is the strong constrain at global scale of the data assimilated compared to the fire prior used" - Unclear. Is this assimlated CO data or assimilated CO2 data? Constraint on what quantity, NBE?
For clarity, we modified by "A major result of this work, that we can observe at global scale, is the strong constraint and influence of the $CO_2$ assimilated data among the inversions, on the net fluxes.".

L23: "tropical regions suggest sensitivity to the fire prior" - Sensitivity of what to the fire prior? NBE or NEE?
For clarity, we modified by "But results in the tropical regions suggest net flux sensitivity to the fire prior for both the IS and OCO-2 inversions.".

---

## Author Response (AR3)

**Reply to Reviewer #1**

We thanks the reviewer for the helpful comments and addressed all comments below. The reviewer comments are in grey, our responses in black.

The authors have addressed my final major concerns with satisfaction. I also appreciate the improved figures provided with the new manuscript. I recommend the paper to be accepted after the final remaining minor issues are resolved (see below). I concur with the other reviewer that some parts are hard to follow. In particular for someone who is reading it for the very first time. I suggest the authors and perhaps the Copernicus typesetters have a final careful look at some of the sentence structures.

Line numbers refer to the acp-2022-120-ATC2.pdf difference document

- Line 19: "balanced" => balancing Corrected Line 17

- Line 19: "the match of CO2 observations" => the constraints provided by CO observations. The sentence only refers to CO2 inversions and not CO, so we corrected the sentence Line 17 with "Consequently, the differences observed in the NEE posterior emissions are a result of the balancing with fires and the constraints provided by CO2 observations."

- Line 25: "is better" => performs better Changed Line 23

- Line 26: "A major result of this work, that we can observe at global scale, is the strong constraint and influence of the CO2 assimilated data among the inversions, on the net fluxes. "

  => We observe the assimilated CO2 data has strong influence on the global net fluxes among the different inversions. Changed

- Line 26: "have closer emissions each other and so are more influenced by observations " => have more similar emissions, mostly as a result of the observational constraints, and to a lesser extent because of the fire prior used. We changed the sentence with "Inversions using OCO-2 (or IS) data have similar emissions, mostly as a result of the observational constraints, and to a lesser extent because of the fire prior used.

- Line 38: "and historically different efforts" Start as a new sentence. Changed Line 35

- Line 255: "representativeness" => representation error. We prefer to keep representativeness error as this term has been used in several previous studies (Bergamashi et al., 2010, Hooghiemstra et al., 2012, Boersma et al., 2016) using chemistry transport model (such as TM5 and GEOS-Chem).

- Line 266: "the relative strengths of " Leave this out. Changed

- Line 293: "in GFED versions" => of the GFED versions Changed

- Line 305: "in general easily quantifiable by careful calibration of instruments. " This sounds like it only includes instrument errors. Do you take into account atmospheric transport errors? I can imagine that those errors can have significant correlations among the different observations.

How large are the errors assumed in matrix R? Please elaborate in the paper. This sentence line 295 is included in the section 2.3.1.a "CO parameterizations", there is only one set of observations here which is MOPITT measurements. The **R** matrix does not include only instrument errors. It includes also transport model errors. We changed the sentences "In the observation covariance matrix **R**, we only assume uncorrelated errors, meaning we only have errors along the diagonal. The **R** matrix includes two errors: instrument errors and transport model errors. This can be assumed since observation error is in general easily quantifiable by careful calibration of instruments." by "The observation covariance matrix **R** includes two errors: instrument errors and transport model errors. In this matrix **R**, we only assume uncorrelated errors, meaning we only have errors along the diagonal. This can be assumed since observation error is in general easily quantifiable by careful calibration of instruments.".

- Line 320: "For better comparison and as the OCOCMS product (we will introduce later) used the emission factor of Andreae and Merlet (2001) and Akagi et al. (2011), "
  => For better comparison we applied the same emission factors used by OCOCMS product (based on Andreae and Merlet, 2001, and Akagi et al. 2011), and not the more recent emission factors provided by Andreae (2019). **Done**

- Line 489: "Further investigations are then needed to study this region" => Further investigations are therefore needed for this region to make more conclusive remarks. **Done line 464**

- Line 525-645. For readability sake this section needs to be broken up into smaller paragraphs. **Done**

- Line 590: "Focusing on the Tropical…" start new paragraph here **Done line 503**

- Line 598: "Very large differences between the…" start new paragraph here **Done**

- Line 602: "For Northern Tropical Africa, net…" start new paragraph here **Done**

- Line 606: "Examining Fig. 6, we …" start new paragraph here Etc. etc. etc. **Done**

- Line 624: "north" => North **Done line 562**

- Line 636: "data constraint" => data constraints **Done line 611**

- Line 661: "with the IS data " you can leave this out. **Done line 573**

- Line 831: "biases of the priors " => start new sentence **Done line 725**

- Line 873: "GtC" => PgC **Done line 762**